# Reversible silencing of lumbar spinal interneurons unmasks a task-specific network for securing hindlimb alternation

Amanda M. Pocratsky[1,2], Darlene A. Burke[2,3], Johnny R. Morehouse[2,3], Jason E. Beare [2,4], Amberly S. Riegler[2,3], Pantelis Tsoulfas[5], Gregory J.R. States[1,2], Scott R. Whittemore[1,2,3] & David S.K. Magnuson [1,2,3]

Neural circuitry in the lumbar spinal cord governs two principal features of locomotion, rhythm and pattern, which reflect intra- and interlimb movement. These features are functionally organized into a hierarchy that precisely controls stepping in a stereotypic, speed-dependent fashion. Here, we show that a specific component of the locomotor pattern can be independently manipulated. Silencing spinal L2 interneurons that project to L5 selectively disrupts hindlimb alternation allowing a continuum of walking to hopping to emerge from the otherwise intact network. This perturbation, which is independent of speed and occurs spontaneously with each step, does not disrupt multi-joint movements or forelimb alternation, nor does it translate to a non-weight-bearing locomotor activity. Both the underlying rhythm and the usual relationship between speed and spatiotemporal characteristics of stepping persist. These data illustrate that hindlimb alternation can be manipulated independently from other core features of stepping, revealing a striking freedom in an otherwise precisely controlled system.

[1] Department of Anatomical Sciences and Neurobiology, University of Louisville, Louisville, KY 40292, USA. [2] Kentucky Spinal Cord Injury Research Center, University of Louisville, Louisville, KY 40292, USA. [3] Department of Neurological Surgery, University of Louisville, Louisville, KY 40292, USA. [4] Cardiovascular Innovation Institute, Department of Physiology & Biophysics, University of Louisville, Louisville, KY 40292, USA. [5] Miami Project to Cure Paralysis, Department of Neurological Surgery, University of Miami Miller School of Medicine, Miami, FL 33136, USA. Correspondence and requests for materials should be addressed to S.R.W. (email: srwhit02@louisville.edu) or to D.S.K.M. (email: dsmagn01@louisville.edu)

Locomotion is a behavior that reflects the interaction between supraspinal, spinal, and sensory systems[1]. While supraspinal structures control its initiation, the spinal cord coordinates the activity of muscles distributed throughout the body into regular patterns of stepping[2]. This complex behavior is based on two principles: rhythm and pattern[3]. Together, these features are functionally organized into a hierarchical network that governs locomotion[1]. Most central to movement is rhythm, which sets the step cycle period and its two defining components: swing and stance duration[4]. Within this rhythm, patterned movements must be precisely controlled to secure effective stepping[3]. Specifically, flexion and extension must be exquisitely timed to allow limb segments to shift around joints (intralimb coordination)[3] while movements between limb pairs must be coordinated (interlimb coordination)[5]. These sequences of interlimb movement are the defining features of gait[6]. As a function of speed, each gait is characterized by a distinct set of stepping rhythms and patterns[7]. Therefore, not only are these principle features precisely controlled, but they are also adaptable to the speed. The spinal networks that collectively produce this behavior are called central pattern generators, with cervical and lumbar spinal enlargements serving as hubs for the forelimbs and hindlimbs, respectively[3]. Understanding how locomotion is governed through this hierarchical network involves various levels of depth and complexity. At the systems level, emphasis is placed on describing the overall behavior of the animal during locomotion[8]. A more in-depth approach to determine how specific pathways functionally integrate into the system occurs at the network level[8]. Finally, the intrinsic and dynamic properties of individual neurons and synapses is investigated at the cellular level[8]. In this study, we explored the functional consequences of silencing an anatomically defined spinal pathway in an otherwise intact system in the freely behaving adult rat.

Using a dual-virus Tet[On] system originally developed by Isa and colleagues[9], we targeted L2 descending interneurons that project ipsi- or contralaterally to L5 in the adult rat spinal cord. A potential analog of the commissural pathway silenced here has been previously studied in the isolated neonatal rodent spinal cord[10–12]. Using electrophysiological techniques, contralateral L2–L5 interneurons were shown to be rhythmically active throughout all phases of the locomotor cycle, leading the investigators to suggest that this pathway likely coordinates the actions of various muscles required for multi-joint movements during stepping[10, 13–15]. Therefore, we hypothesized that conditionally silencing ipsi- and contralaterally projecting L2–L5 interneurons would affect flexor–extensor coordination across the joints, disrupting hindlimb kinematics during locomotion. Here, we show that reversible silencing of L2–L5 interneurons selectively disrupts left–right hindlimb alternation during overground locomotion while preserving normal intralimb movements. This perturbation, which occurs spontaneously on a step-by-step basis independent of both locomotor speed and step frequency, does not translate to non-weight-bearing locomotion. Together, these data reveal that interlimb coordination can be selectively manipulated independently from the core features of stepping, unmasking incredible flexibility in an otherwise precisely controlled system.

## Results

### Intralimb coordination persists during overground locomotion.
We performed bilateral injections at the L2 and L5 spinal cord segments to silence both the ipsilateral and contralateral projections (Fig. 1a). In double-infected neurons that constitutively express rtTAV16, doxycycline (DOX) induces enhanced tetanus neurotoxin (eTeNT) expression (Fig. 1b). eTeNT is then transported to the terminal field where it prevents

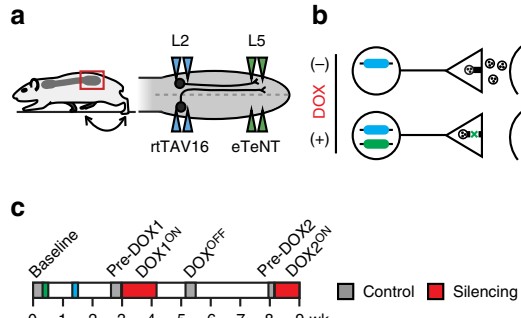

**Fig. 1** Experimental design to conditionally silence L2–L5 interneurons in the freely behaving adult rat. **a** In the lumbar spinal cord, L2 neurons with ipsilateral or contralateral projections to L5 were targeted for conditional silencing. Bilateral injections of AAV2-CMV-rtTAV16 (blue triangles) and HiRet-TRE-EGFP.eTeNT (green) were performed at L2 and L5, respectively. **b** In the presence of doxycycline (DOX), only double-infected neurons conditionally express eTeNT (adapted from Kinoshita et al.[9]). eTeNT is transported to the terminal field where it prevents synaptic vesicle release thereby silencing neurotransmission. **c** Behavioral assessments were performed at four control time points (Baseline, BL; Pre-DOX1, PD1; DOX[OFF]; Pre-DOX2, PD2) and during two rounds of DOX[ON] silencing separated by a 1-month washout (DOX1[ON] days 3, 5, 8 and DOX2[ON] days 3 and 5). Illustrations by A. Pocratsky

exocytosis of synaptic vesicles, thereby silencing neurotransmission. Removing DOX from the drinking water restores neurotransmission, allowing acute and reversible silencing of this anatomically defined pathway in the otherwise intact adult rat (Fig. 1c).

To determine the functional consequences of silencing L2–L5 interneurons on intralimb coordination, we marked the skin overlying the iliac crest, hip, ankle, and toe (Fig. 2a) to describe the limb using a three-segment, two-angle model (Fig. 2)[16]. At control time points, animals displayed stereotypic coordination with normal excursions of the limb segments (Fig. 2b), and the proximal and distal angles (Fig. 2e–f; Supplementary Figure 1a, d). Unexpectedly, when we silenced L2–L5 interneurons we saw a disruption in left–right hindlimb alternation during stepping (Supplementary Movie 1). The severity of this disruption ranged from mild changes in alternation to hindlimb "hopping" where the hindlimbs moved synchronously (Fig. 2c, d). Despite the silencing-induced effects on left–right alternation, intralimb coordination persisted as seen by the characteristic excursion pattern of proximal and distal limb segments (Fig. 2g, h, Supplementary Figure 1b, c, e, f) and the temporal coordination between the proximal and distal angles (Fig. 2i). Collectively, these data suggest that L2–L5 interneurons are likely not involved in intralimb, flexor–extensor coordination during overground locomotion.

### Silencing alters the overall locomotor stepping pattern.
In light of the overt changes to left–right hindlimb movements, we determined if silencing L2–L5 interneurons affected the overall locomotor stepping pattern. To do this, we analyzed the step sequence pattern[17], which is an analysis of the footfall order without taking into account more discrete temporal indices such as stance and swing durations (see Methods for details). The primary pattern commonly used by rodents is called alternate, which is characterized by the following footfall order: (step 1) right forelimb, (2) left hindlimb, (3) left forelimb, and (4) right hindlimb (Fig. 3a). Prior to silencing, the alternate step pattern predominated (Fig. 3d). When we silenced L2–L5 interneurons,

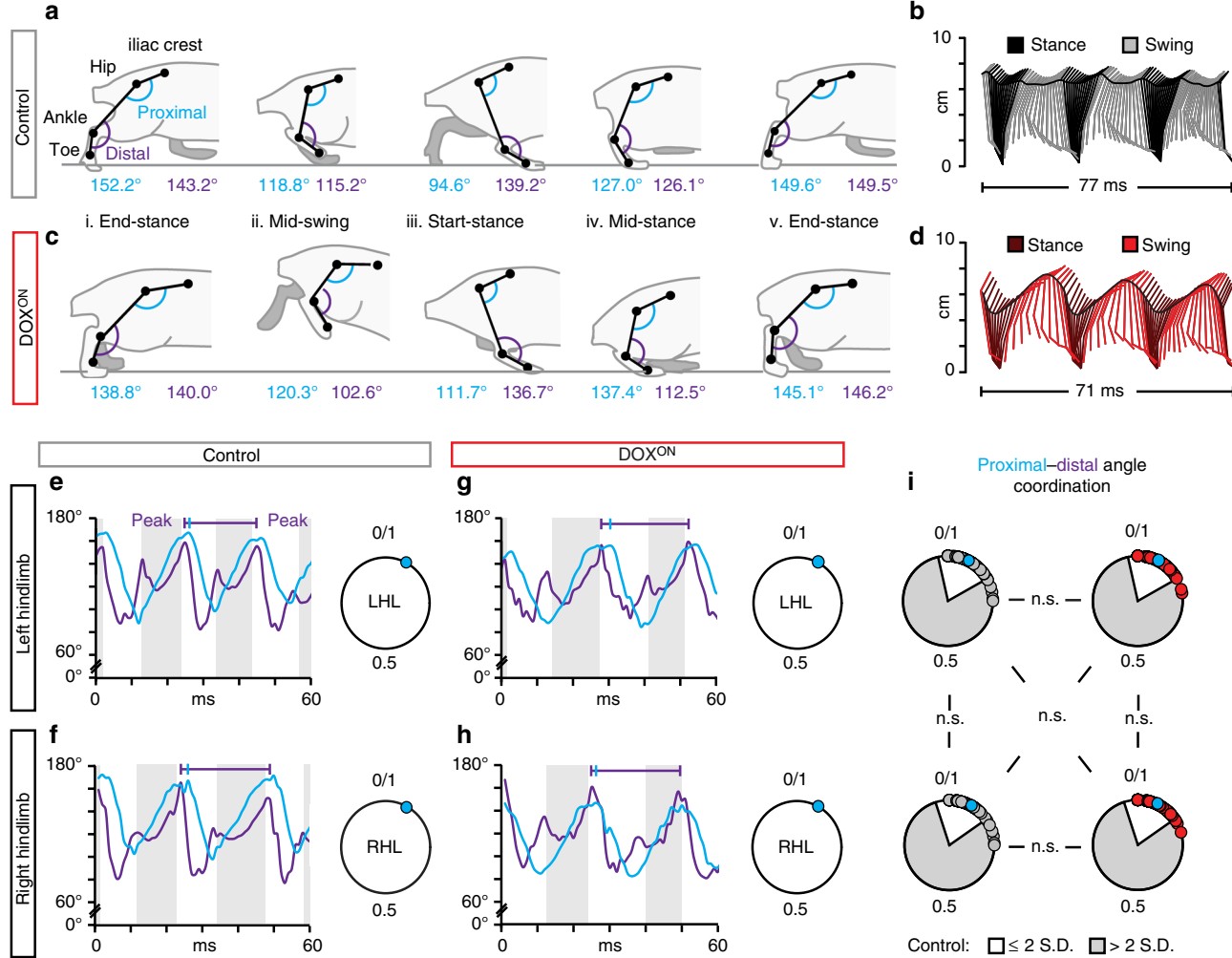

**Fig. 2** L2–L5 interneurons are dispensable for intralimb coordination during locomotion. Representative illustrations of intralimb movements during **a** control and **c** DOX[ON] stepping, as described by three hindlimb segments (iliac crest–hip, hip–ankle, and ankle–toe) and two hindlimb angles (proximal (blue): iliac crest–hip–ankle angle, distal (purple): hip–ankle–toe angle). Proximal and distal angles are shown for five select instances throughout one step cycle. **b**, **d** Two-dimensional stick figures of hindlimb stepping at Control and DOX[ON] time points, respectively (from the same stepping passes shown in (**a**, **c**); sampling rate = 100 Hz). Silencing L2–L5 interneurons increased vertical movements in the hip (**d**, black horizontal trace) while preserving normal hindlimb range of motion (Supplementary Figure 2). Onset times of peak angular excursions (**e**–**h**, blue and purple hashmarks) were analyzed to assess the temporal coordination between proximal and distal angles (see Methods for detail) (traces = excursions of proximal and distal angles; shaded region = stance phase; 0/1 in circular plot denotes normal, in phase coordinated movements; individual circles = intralimb coordination value for one step cycle; LHL left hindlimb; RHL right hindlimb). **i** Proximal-to-distal intralimb movements of the left and right hindlimb (as well as between limb pairs) remained unaffected during silencing (n = 130 and 150 intralimb cycles for collapsed Control and DOX[ON] time points, respectively; white inset = normal variability at control time points). All comparisons in (**i**) were p > 0.5 (Watson's nonparametric two-sample $U^2$ test; LHL: Control vs. DOX[ON] $U^2$ = 0.04232; RHL: Control vs. DOX[ON] $U^2$ = −0.13874; Control: LHL vs. RHL $U^2$ = −0.00653; DOX[ON]: LHL vs. RHL $U^2$ = 0.01623; Control LHL vs. DOX[ON] RHL: $U^2$ = 0.04225; Control RHL vs. DOX[ON] LHL: $U^2$ = 0.04232). Illustrations by A. Pocratsky

animals significantly increased their use of the cruciate step pattern (Fig. 3e), which reflects the sequential movements of the homologous limb pairs as opposed to alternation between the shoulder and pelvic girdles (Fig. 3b, c, cruciate: forelimb–forelimb–hindlimb–hindlimb footfall order). Removing DOX from the drinking water reversed this pattern shift (Fig. 3d, e) and silencing 1 month later reproduced the effects (Fig. 3f, g). These data suggest that silencing L2–L5 interneurons produces a quadrupedal stepping behavior that is primarily forelimb-leading and hindlimb-trailing as opposed to the stereotypic alternation between the two girdles.

**Silencing disrupts hindlimb alternation during stepping.** The salient observation from silencing L2–L5 interneurons is a change

in hindlimb alternation during stepping (Supplementary Movie 1). However, quadrupedal mammals will naturally express various patterns of interlimb coordination. These distinct interlimb coupling patterns are defining features of the classic gaits[6]. Therefore, we determined if the interlimb coordination expressed during silencing reflected these stereotypic gait patterns.

Walking and trotting are slower gaits wherein the hindlimbs alternate (Fig. 4a, lower panel). This temporal relationship can be expressed as a coordination (phase) value by dividing the initial contact time of the left hindlimb by the right hindlimb stride time (stance+swing). These phase values, ranging from 0 to 1, are plotted on a circular graph to illustrate interlimb coordination (Fig. 4a, lower right). For walking and trotting, the phase value is close to 0.5 (180°), indicating left–right alternation with out-of-phase hindlimb movements[7]. With increasing speed, the gait

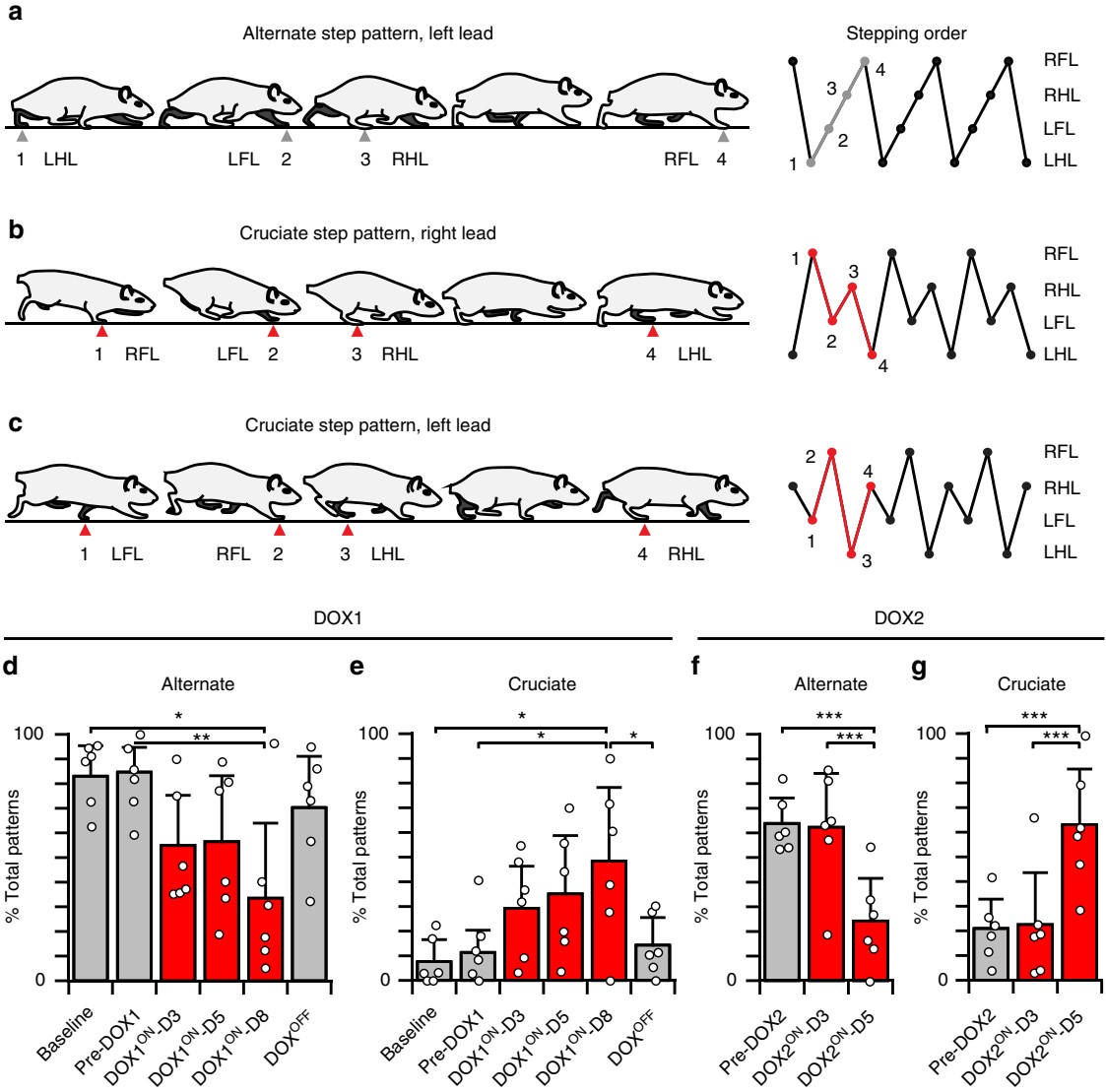

**Fig. 3** Silencing L2–L5 interneurons altered the locomotor step sequence pattern. **a–c** Schematics of alternate and cruciate step sequence patterns (SSPs) as defined by footfall order (right panels, SSP illustrated as a series of circles interconnected with lines). SSPs are defined by the footfall order (does not measure the duration of swing–stance phases). **d** The alternate pattern predominates at control time points (Baseline: 83.0 ± 13.5%; Pre-DOX1: 82.3 ± 14.8%). **d, e** Silencing L2–L5 interneurons changed the pattern from alternate to cruciate (DOX1ON-D8: 33.5 ± 30.4% alternate, 48.4 ± 29.8% cruciate as compared to 7.7 ± 8.9% and 11.4 ± 9.0% at Baseline and Pre-DOX1, respectively). **f, g** This was reversed by DOX removal and replicated 1 month later (circles = individual means; bars = group mean ± S.D.; $n = 47$–66 step sequence patterns/time point; data shown as percentage of total patterns observed) (*$p \leq 0.05$; **$p \leq 0.01$; ***$p \leq 0.001$, repeated measures analysis of variance (ANOVA) and Tukey's honest significant difference (HSD) post hoc $t$-tests). LHL left hindlimb, RHL right hindlimb, LFL left forelimb, RFL right forelimb. **d** Baseline vs. DOX1ON-D8: $p = 0.014$, critical $t = 0.41$, df = 20; Pre-DOX1 vs. DOX1ON-D8: $p = 0.006$; critical $t = 0.51$, df = 20. **e** Baseline vs. DOX1ON-D8: $p = 0.016$; Pre-DOX1 vs. DOX1ON-D8: $p = 0.032$; DOX1ON-D8 vs. DOXOFF: $p = 0.048$; each comparison with a critical $t = 0.33$, df = 25. **f** Pre-DOX2 & DOX2ON-D3 vs. DOX2ON-D5: $p < 0.001$; each with critical $t = 0.2$, df = 10. **g** Pre-DOX2 & DOX2ON-D3 vs. DOX2ON-D5: $p < 0.001$; each with critical $t = 0.3$, df = 10. Illustrations by A. Pocratsky

switches from walk–trot to gallop where there is a phase shift with increased overlap between left and right stance (or swing) phases (Fig. 4b, phase ≈0.25/0.75)[7]. At even higher speeds, some animals will switch their gait to bounding where the hindlimb movements are in-phase (Fig. 4c, phase ≈0/1). These gait-specific coordination changes also occur in the forelimbs.

To control for interanimal variability in the lead limb (illustrated in Fig. 3)[18, 19], we transformed hindlimb phase values of <0.5 to the reciprocal >0.5 and found the mean phase of all control time points (Fig. 4d). Any value >2 S.D. from this mean is beyond control variability and was considered "irregular". Prior to silencing, left–right hindlimb alternation was the overt stepping pattern (Fig. 4e). Silencing L2–L5 interneurons significantly

disrupted this alternation, but the changes observed were not clustered at the coordination values reflective of the traditional gaits. Instead, we saw the emergence of a coordination continuum from hindlimb walking to hopping (Fig. 4e). Notably, the silencing-induced effects were not all-or-none as seen by the preponderance of phase values within the normal range. Removing DOX restored hindlimb alternation while resilencing 1 month later once again significantly disrupted it. When examining the interanimal variability in the maximal deficits observed during silencing, we saw a range from approximately 20% (modest disruption) to 70% (overt disruption) of the hindlimb steps taken affected (see Methods for detail). Moreover, we saw variability in the time point of peak disruption to

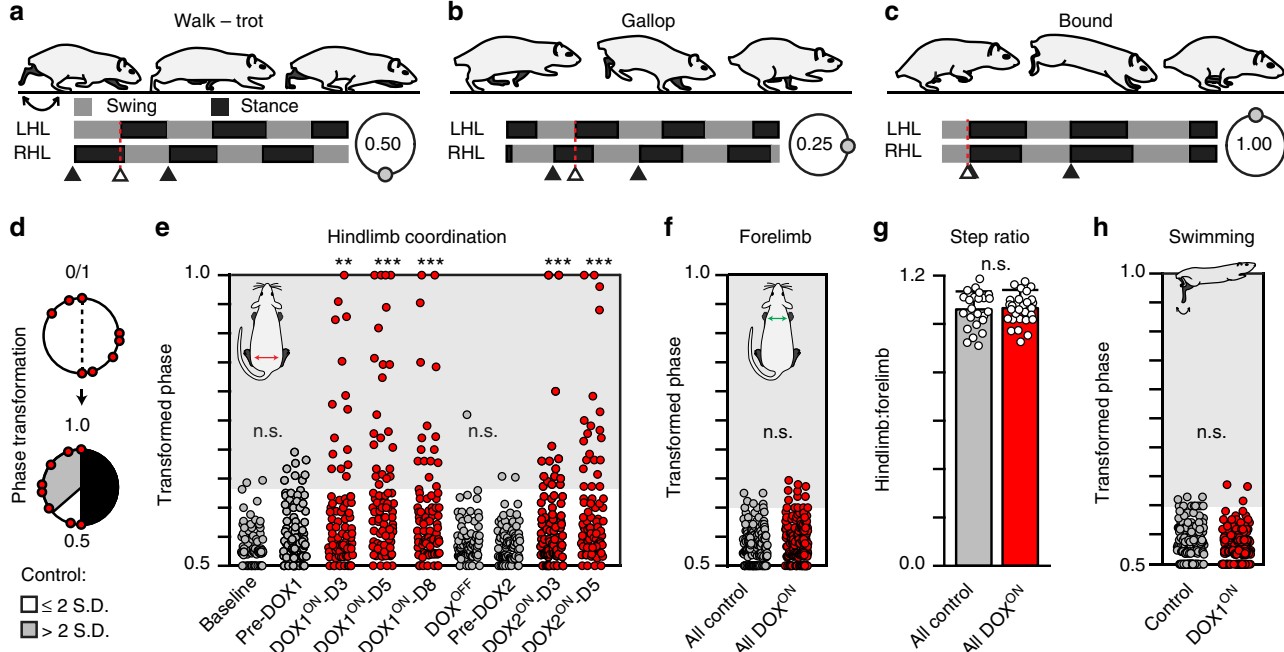

**Fig. 4** Silencing L2–L5 interneurons selectively disrupts left–right hindlimb alternation during overground stepping. **a–c** Stereotypic locomotor gaits with representative swing–stance graphs and characteristic left–right hindlimb coordination values (shown in circular plots, see Methods for detail). **d** Schematic illustrating phase transformation (see Methods for detail). Shaded area denotes any coordination value beyond normal variability observed (>2 S.D.) at control time points. Each circle represents one step cycle ($n = 84$/time point; $n = 10$–14/animal). **e** Silencing L2–L5 interneurons significantly increased the proportion of steps that deviated beyond normal variability observed at control time points. Removing DOX restored alternation and silencing one month later repeated the effects (Baseline, $n = 3/84$ vs. DOX1ON, $n = 15/84$ ($p = 0.002$, $z = 3.08$), 24/84 ($p < 0.001$, $z = 4.69$), 17/84 ($p = 0.001$, $z = 3.45$); Pre-DOX2, $n = 2/84$ vs. DOX2ON, $n = 15/84$ ($p = 0.001$, $z = 3.44$), 21/84 ($p < 0.001$, $z = 4.52$); $**p \leq 0.01$, $***p \leq 0.001$; Binomial Proportion (B.P.) Test). Control time points were not significantly different from each other (Baseline vs. Pre-DOX1, $p = 0.116$, $z = 1.57$; DOXOFF vs. Pre-DOX2, $p = 0.562$, $z = 0.58$). **f** Left–right forelimb alternation was not perturbed (All control vs. All DOXON, $p = 0.322$, $z = 0.99$). **g** No differences were observed in the hindlimb/forelimb stepping index (circles = individual animal's average hindlimb:forelimb step ratio at each time point; Control: $N = 24$ total average step ratios; DOX: $N = 30$ total average step ratios; bars represent group average $\pm$ S.D.; $p = 0.75$, critical $t = 2.003$, df = 52, two-sample $t$-test). **h** Left–right hindlimb alternation persisted during swimming ($n = 80$ stroke cycles/time point; $p = 0.20$, $z = 1.28$, B.P. test). In a-c, classic gait data were generated from a separate study, see Methods for detail. Illustrations by A. Pocratsky

hindlimb alternation, ranging from DOX1ON-D5 to DOX2ON-D5, over 1 month later. There was no significant difference between DOX1 and DOX2 in the magnitude of change in left–right hindlimb coordination during stepping (22.22% ($n = 56/252$) and 21.43% ($n = 36/168$) steps affected, respectively; $p = 0.85$; Binomial Proportion test), although the peak effects of silencing L2–L5 interneurons were observed during DOX1. Moreover, these perturbations to hindlimb alternation did not influence or "spread" to the forelimbs as alternation persisted (Fig. 4f). Despite the significant change in hindlimb coordination, the animals maintained a 1:1 stepping relationship between the forelimbs and hindlimbs (Fig. 4g).

We also investigated hindlimb coordination during swimming (Fig. 4h), a bipedal locomotor behavior where the limbs are unloaded and the proprioceptive and cutaneous feedback associated with plantar stepping is altered. Strikingly, hindlimb alternation remained intact (Fig. 4h; Supplementary Movie 2), suggesting that L2–L5 interneurons that were silenced do not participate in hindlimb alternation during a task where afferent feedback associated with stepping is altered/removed. Collectively, these data suggest that silencing L2–L5 interneurons selectively disrupts hindlimb alternation in a context-specific manner, without affecting the overall stepping ability of the animal.

**Silencing partially uncouples the hindlimbs during stepping**. In light of the range of coupling patterns expressed during hindlimb

stepping, we examined the underlying variability in the phase data. To do this, we measured the coefficient of variation in the hindlimb phase data at all Control and all DOXON time points, respectively (see Methods for detail). Silencing L2–L5 interneurons significantly increased the variability in hindlimb coordination in both the raw (Control: $10.65 \pm 2.36\%$ vs. DOXON: $28.33 \pm 9.02\%$, $p \leq 0.005$, paired t-test) and transformed data sets (Control: $6.88 \pm 1.53\%$ vs. DOXON: $16.11 \pm 3.87\%$, $p \leq 0.005$). The increase in phase variability raises an important question: does the change in hindlimb coordination reflect a functional uncoupling of the left–right limb pair?

Typically, the limb pairs at each girdle work together as a coupled unit during stepping[6]. This functional coupling ensures that they step in a consistent fashion, regardless of the gait. To measure this, we performed circular statistics on the raw data set to quantify the amount of concentration (Fig. 5) in the phase data[20]. Limb coupling is exemplified by a high degree of concentration in one direction (Fig. 5a, top). Alternatively, complete uncoupling indicates that the left and right hindlimbs are stepping independently from each other, with different frequencies, giving a uniform phase distribution around the circular plot (Fig. 5a, bottom)[21].

Prior to silencing, the hindlimbs were coupled at alternation with the preponderance of phases near 0.5 and with normal variability (Fig. 5b, top). During silencing, this concentration was significantly reduced. Instead, the coordination values became distributed around the circular plot (Fig. 5b, bottom, Supplementary Table 1).

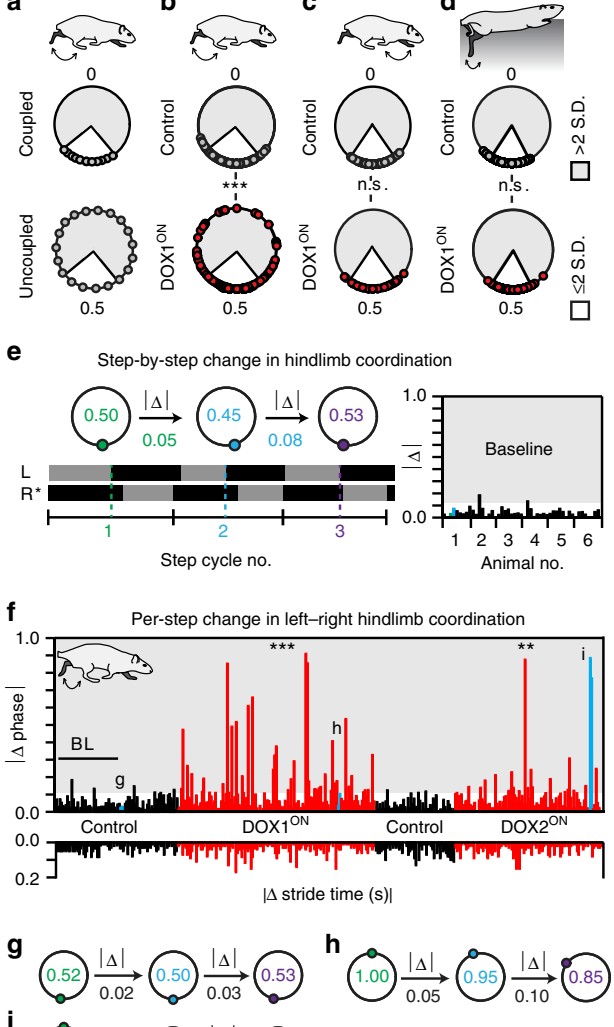

**Fig. 5** Silencing L2–L5 interneurons partially uncouples the hindlimbs, significantly increasing the step-by-step variability in left–right coordination. **a** Coupling and complete uncoupling schematics (white inset = normal variability observed at control time points). **b** Silencing L2–L5 interneurons significantly decreased phases concentrated at alternation during hindlimb stepping as compared to Control (Baseline + Pre-DOX1 vs. DOX1$^{ON}$D3,5,8; ***$p < 0.001$; $U^2 = 0.59762$; Watson's $U^2$ test, $n = 84$ step/time point). **c** Forelimb stepping ($U^2 = 0.004403$) and **d** hindlimb swimming ($U^2 = -2.8665$; $n = 80$ stroke cycles/time point) remained clustered at alternation (see Supplementary Tables 3, 4). **e** Example of swing–stance graph illustrating analysis of consecutive strides within one locomotor bout to determine the step-by-step (absolute) change in left–right coordination (*denotes reference limb for one complete stride cycle). Individual bars represent change in left–right coordination, per step, for each animal across all time points (right panel, e.g., Baseline, absolute change in coordination between steps 1 to 2 and 2 to 3 highlighted in green and blue, respectively). Shaded region indicates per-step changes in coordination beyond control variability. **f** (top panel) Silencing L2–L5 interneurons significantly increased step-by-step variability in hindlimb coordination (Baseline+Pre-DOX1 vs. DOX1$^{ON}$, $n = 3/80$ vs. 38/136 ($p < 0.001$, $z = 5.5$); DOX$^{OFF}$+Pre-DOX2 vs. DOX2$^{ON}$, $n = 2/53$ vs. 21/100 ($p = 0.00039$, $z = 3.56$); **$p < 0.01$, ***$p < 0.001$, B.P. test). Select examples of per-step changes in left–right coordination (vertical bars highlighted in blue) are emphasized in **g-i** below. The hindlimbs stepped with relatively consistent stride durations (bottom panel). Illustrations by A. Pocratsky.

Removing DOX restored the concentration of phase values towards alternation and silencing 1 month later replicated the effects. As anticipated, the forelimb phases remained concentrated at alternation (Fig. 5c) as did the hindlimbs during swimming (Fig. 5d; Supplementary Table 2). Therefore, we contend that silencing L2–L5 interneurons results in the partial uncoupling of the hindlimbs during overground stepping. Moreover, this uncoupling does not impact the forelimbs nor translate to a non-weight-bearing locomotor task.

Thus far, we have examined the effect of silencing L2–L5 interneurons on the overall stepping performance. We next explored how the disruption in left–right alternation influenced dynamic coordination on a step-by-step basis. We quantified the absolute change in phase per step and used this as an indicator for the relative consistency in hindlimb coordination (Fig. 5e). Consistent hindlimb coordination is typified by minor changes in phase on a stride-by-stride basis, which suggests the limbs are stepping in a regular, repeated fashion. Conversely, large changes in coordination per stride would indicate increased variability between the hindlimbs as they are stepping.

Figure 5f summarizes the absolute change in left–right hindlimb coordination, per step, for the individual locomotor bouts of each animal across all time points (Supplementary Figure 2). Any sequence of steps with a change in hindlimb coordination beyond the normal stride-by-stride variability is plotted in the shaded area. Prior to silencing, there were minor changes in step-by-step coordination with approximately 96% of all steps taken falling within the normal variability observed at control time points (Fig. 5f). These small changes were primarily concentrated around 0.5 (Fig. 5g). Silencing L2–L5 interneurons significantly increased the step-by-step variability in hindlimb coordination, as seen by the large spikes in absolute phase change (Fig. 5f, top, red bars). This included step sequences where the hindlimbs started in phase and then drifted out of phase (Fig. 5h) as well as instances of dramatic changes in coordination per step (Fig. 5i; Supplementary Figure 2g, i). Further exploration of this finding would require a longer walkway, where the increased number of contiguous step cycles would shed more light on the relative drift in left–right hindlimb coordination on a cycle-by-cycle basis. Notwithstanding, removing DOX restored the consistency in step-by-step coordination while resilencing 1 month later replicated the increased variability.

In addition to analyzing the silencing-mediated effects on dynamic hindlimb coordination, we also examined the per-step changes in stride time, which is the duration of the stance and swing phase for one step cycle. We calculated changes in stride time and matched them to the corresponding changes in hindlimb coordination. Surprisingly, the hindlimbs continued to step with stride durations that fell within the normal range despite the large shifts in left–right coordination (Fig. 5f). These data suggest that silencing L2–L5 interneurons does not disrupt left–right alternation by inducing a switch to a gallop- or bound-like stepping pattern. Instead, silencing L2–L5 interneurons appears to introduce dynamic instability into the system, allowing spontaneous shifts in left–right coordination to occur on a step-by-step basis. These events are in addition to locomotor bouts wherein the hindlimbs consistently stepped in a gallop- or bound-like fashion. However, all of these changes occurred alongside relatively invariable stride durations, suggesting that the disruptions to hindlimb coordination occur within the confines of a stable locomotor rhythm.

**Key locomotor features persist during alternation disruption.** Changes in interlimb coordination are tantamount to transition between the slower walk–trot gaits and the faster, more

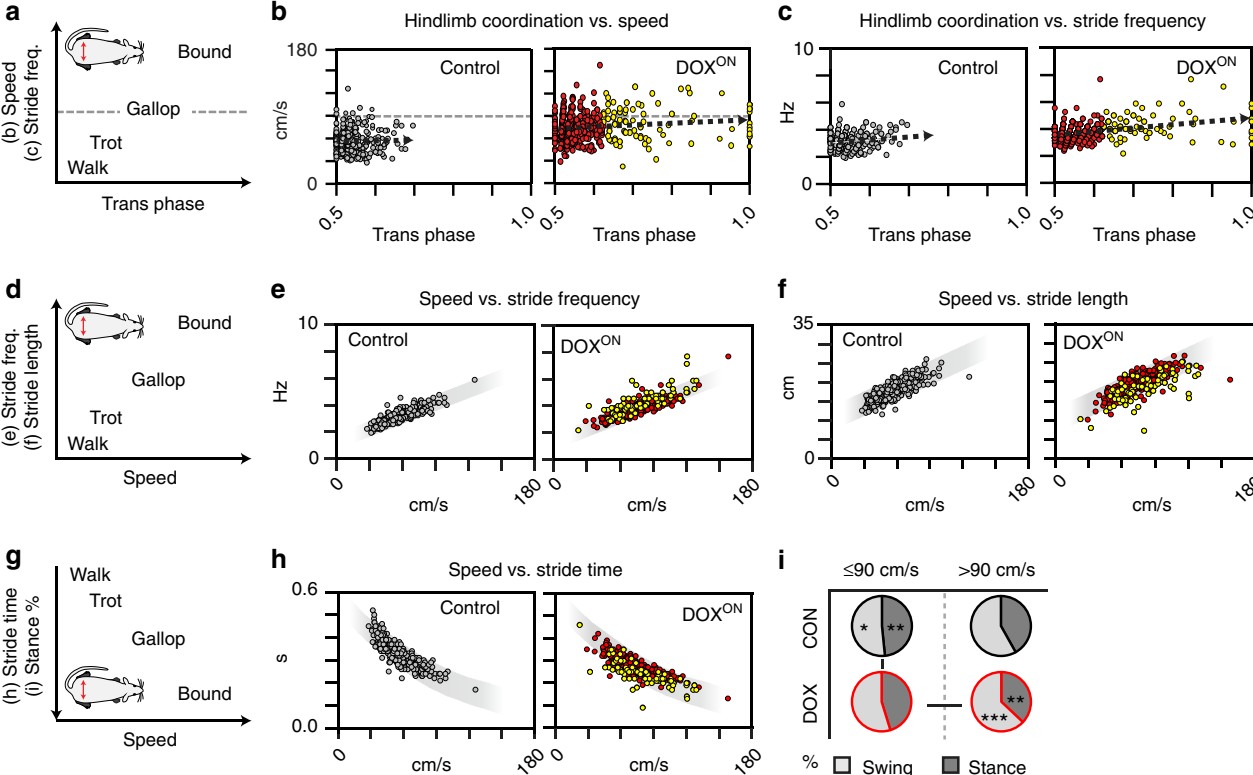

**Fig. 6** Silencing L2–L5 interneurons disrupts hindlimb alternation independent of speed and step frequency while preserving salient features that govern locomotion. **a** Schematic illustrating the general relationship between speed/step frequency and locomotor gaits. Dashed gray line indicates reported transition between the alternating trot and asynchronous gallop in the adult rat. **b** Silencing-induced changes in hindlimb coordination did not correlate with increased speed (Control, $r_S = -0.01$, $R^2 = 0.01\%$; DOX$^{ON}$, $r_S = 0.13$, $R^2 = 1.69\%$; Spearman's rank; see Supplementary Figure 3) nor increased **c** step frequency (Control: $r_S = 0.04$, $R^2 = 0.16\%$; DOX$^{ON}$: $r_S = 0.33$, $R^2 = 10.89\%$). **d**, **g** Schematics indicating the general relationship between speed and various spatiotemporal gait indices. Silencing L2–L5 interneurons does not affect these stereotypic associations, including **e** speed vs. step frequency (CON $r_S = 0.87$, $R^2 = 75.69\%$; DOX $r_S = 0.83$, $R^2 = 68.89\%$), **f** stride length (CON $r_S = 0.81$, $R^2 = 65.61\%$; DOX $r_S = 0.78$, $R^2 = 60.84\%$), **h** stride time (CON $r_S = -0.87$, $R^2 = 75.69\%$; DOX $r_S = -0.83$, $R^2 = 68.89\%$), and the relative decrease in **i** stance duration (Control vs. DOX$^{ON}$, %swing: $p = 0.011$; %stance: $p = 0.007$, each with critical $t = 2.228$, df = 10; independent $t$-test; DOX$^{ON}$ speed comparisons: %swing: $p = 0.00006$; %stance: $p = 0.0002$, each with critical $t = 2.57$, df = 5; paired $t$-test; group average of all Control and all DOX time points, respectively) (yellow = phases >2 S.D. control mean; Control: $n = 336$ steps; DOX$^{ON}$: $n = 420$; shaded region = 95% prediction interval for control). *$p \leq 0.05$, **$p \leq 0.01$, ***$p \leq 0.001$. Illustrations by A. Pocratsky

synchronous gallop–bound (Fig. 6a). We determined whether the observed perturbations in hindlimb coordination were associated with stepping speed. While we saw a generalized enhancement in speed during DOX$^{ON}$ (Supplementary Figure 3a), there was no association between the silencing-induced changes in hindlimb coordination and the speeds at which each step occurred (Fig. 6b, correlation coefficient = 0.13). This result was substantiated when we analyzed the individual time points, the averaged data sets, and the irregular steps only (Fig. 6, yellow circles) (Supplementary Figure 3b, e–g). Importantly, approximately 67% of these irregular steps occurred at speeds less than 90 cm/s, a velocity where the walk–trot alternating gaits (left–right limbs at each girdle move out of phase relative to one another) typically predominate in the adult rat[22] (Supplementary Figure 3c, d). We also examined the step frequency (number of steps/second) which usually increases with speed. It is at greater step frequencies that the more synchronous gaits typically occur where the left–right limb pairs at each girdle move in phase (full-bound) or with a slight phase shift (gallop) relative to one another (Fig. 6a)[7]. When we compared the silencing-induced changes in hindlimb alternation to step frequency, we once again saw no meaningful correlation (Fig. 6c). The individual time point comparisons corroborated these findings (Supplementary Tables 1, 2). We also saw no correlations between silencing-induced changes in hindlimb coordination and various gait parameters (Supplementary

Figure 3e–j, Supplementary Tables 1, 2). Therefore, silencing L2–L5 interneurons alters hindlimb coordination independent of locomotor speed and step frequency.

Due to the unexpected dissociation between changes in hindlimb coordination and step frequency/speed, we examined whether other principal features that govern locomotion were affected. During stepping, the limbs are coordinated in both space and time. This allows us to quantify locomotion using a set of spatiotemporal parameters, or gait indices, which include stride length, stance duration, swing duration, and stride time[18, 19]. Importantly, these parameters change with speed in a stereotypic, well-characterized manner (Fig. 6d, g)[19]. Silencing L2–L5 interneurons did not alter the primary spatiotemporal gait indices of step frequency (Fig. 6e, right panel), stride length (Fig. 6f)[19], stride and swing times (Fig. 6g, h), and stance duration (Fig. 6i). Some animals did not step at velocities greater than 90 cm/s at control time points, preventing a statistical comparison between the two speed categories (Fig. 6i, top, left vs. right). Nonetheless, the pattern and magnitude of change in swing–stance durations were similar (≤90 cm/s: swing $51.59 \pm 6.24\%$, stance $48.41 \pm 6.24\%$; >90 cm/s: swing $58.68 \pm 3.79\%$, stance $41.32 \pm 3.79\%$).

When we focused on the dispersion of the irregular hindlimb steps, it became apparent that the silencing-induced changes to alternation occurred over a relatively broad range of speeds, step

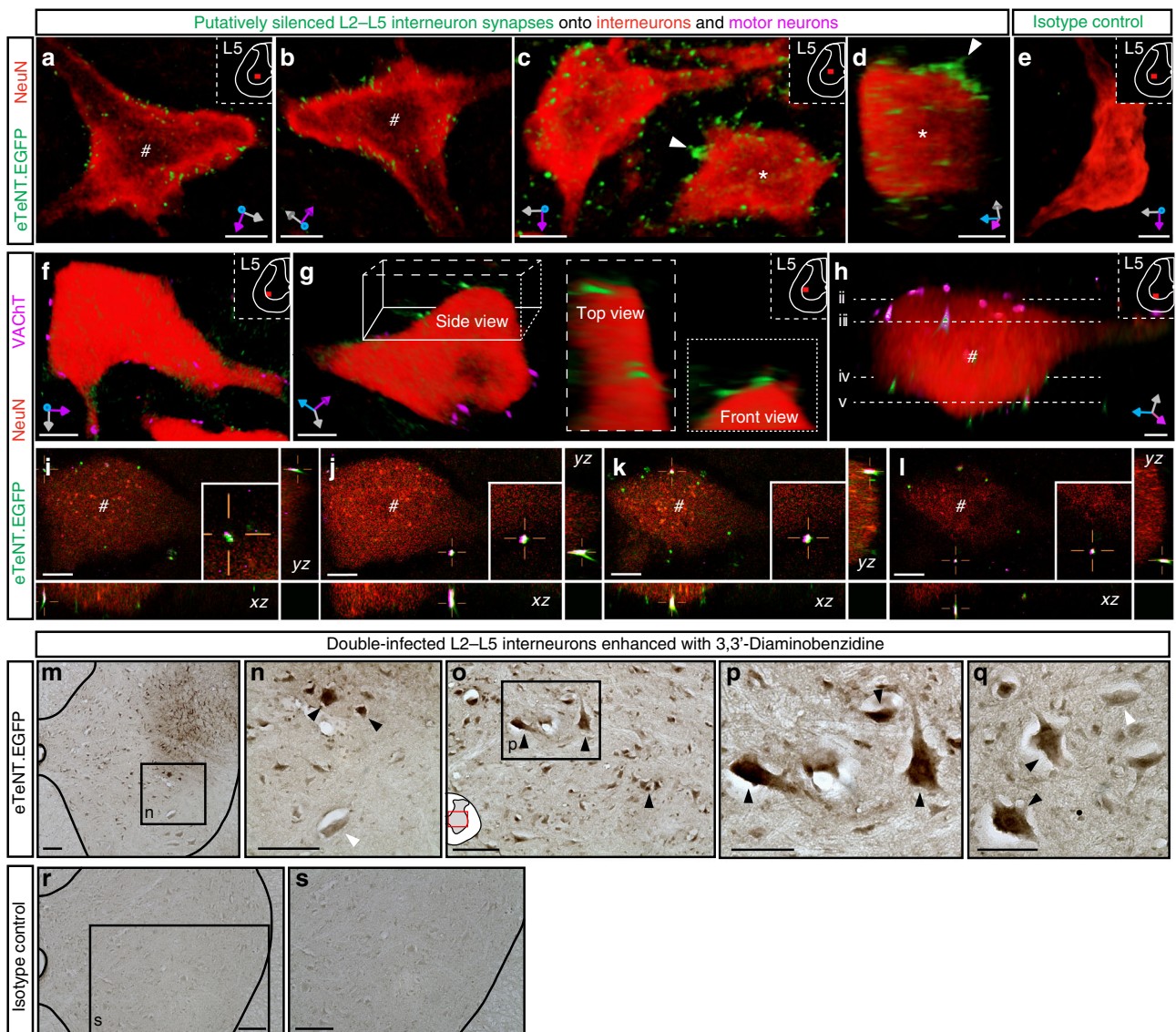

**Fig. 7** Immunohistochemical detection of eTeNT.EGFP in the double-infected L2–L5 interneurons. **a–d** Volumetric three-dimensional renderings show eTeNT.EGFP+ terminals closely apposed to somata and primary dendrites of neurons in the intermediate gray matter (#,* indicate same neuron rotated, x–y–z axes shown in gray/magenta/blue). **e** Isotype control showed little to no signal (nonimmune rabbit sera at 5 mg/ml, 1:5000 dilution). **f–h** Volume rendered images show eTeNT.EGFP+ terminals surrounding motor neurons marked by vesicular acetylcholine transferase (VAChT, magenta) in the caudal lumbar segments. Cross-sections throughout the z-stack confirm colocalization of VAChT with eTeNT.EGFP in the x–z and y–z planes (**i–l**, white signal in crosshairs; cross-sections throughout motor neuron shown in (**f**); **a**-**f** scale = 10 μm). Amplification of eTeNT.EGFP signal with 3,3′-Diaminobenzidine (DAB) revealed double-infected neurons within the intermediate gray matter of the L2 spinal segment (**m–q** dark arrowheads = eTeNT.EGFP-positive neurons; white arrowheads = eTeNT.EGFP-negative neurons). **r**, **s** Isotype controls revealed no DAB-enhanced eTeNT.EGFP signal. **m–o**, **r**, **s** scale = 100 μm; **p**, **q** scale = 50 μm

frequencies, and spatiotemporal indices. This further supports the notion that the perturbations to hindlimb alternation are not reflective of the traditional gaits[7]. Collectively, these data suggest that silencing L2–L5 interneurons disrupts hindlimb alternation independent of speed and step frequency, while preserving key stepping characteristics that are fundamental to locomotion.

**Immunohistochemical detection of eTeNT-positive synapses.** Following terminal assessments, animals were killed on DOX2[ON]-D5 for immunohistochemical (IHC) detection of eTeNT.EGFP-positive terminals (Fig. 7a–h; Supplementary Figure 4a–e) and double-infected cell bodies (Fig. 7i–l; Supplementary Figure 4f–ac). Cross-sections of the L4–L5 spinal segments were stained with anti-NeuN to label neuronal cell bodies

and proximal dendrites as well as anti-GFP to amplify eTeNT. EGFP (Fig. 7a–d). Immunoreactive eTeNT.EGFP-positive signal was found in close apposition to somata and primary dendrites of neurons in the ventral gray matter (Fig. 7a). Rotating the neurons in three dimensions confirmed that eTeNT.EGFP-positive terminals surrounded the somata (Fig. 7b), with patterns suggestive of complex branching (Fig. 7c, d). Co-staining with synaptophysin confirmed the presence of eTeNT.EGFP at the presynaptic terminal (Supplementary Figure 4a, b). A subset of these conditionally silenced synapses were inhibitory or excitatory in nature (Supplementary Figure 4b–d). We also observed a small subset of L2–L5 interneurons that were cholinergic as well (Supplementary Figure 6i).

To further characterize these caudal lumbar projections, we used anti-vesicular acetylcholine transferase (VAChT) as a

presynaptic marker for inputs onto motor neurons (refer to Supplementary Figure 6 regarding somatic vs. presynaptic VAChT immunoreactivity). We found eTeNT.EGFP colocalization with VAChT surrounding both the somata and proximal dendrites of motor neurons (Fig. 7f–l). Isotype controls showed little to no EGFP signal (Fig. 7e). When we screened for eTeNT. EGFP-positive, terminal-like immunoreactivity beyond the L4–L5 spinal segments, we found little to no signal (Supplementary Figure 4z–ac).

To better identify the silenced neurons, we used immunoperoxidase enhancement of the conditionally expressed eTeNT. EGFP. In the L2 spinal segment, we observed numerous immunoreactive interneurons (Fig. 7m–q, black arrowhead) throughout the intermediate gray matter intermingled with unlabeled neurons (white arrowhead). Isotype controls revealed no immunoreactivity (Fig. 7r, s). Few double-infected neurons were detected beyond the L2 spinal segment (Supplementary Figure 4f–j, n–q). These data suggest that the behavioral effects observed were a functional consequence of silencing L2 spinal neurons whose projections densely innervate the caudal lumbar segments.

**Bilateral L2–L5 interneurons have sparse local projections**. To better understand the anatomy of the L2–L5 interneurons, we performed tracing experiments. To compare the relative number of ipsilateral versus contralateral-projecting L2–L5 interneurons, we repeated the L5 injections using fluorescent tracers (Fig. 8a–e). Following retrograde labeling of the cell bodies, the L2 segment was dissected, cleared, imaged using light sheet fluorescence microscopy, and unbiased absolute cell counts of the ipsi- and contralateral subtypes were performed (Fig. 8c–e, FluoroEmerald shown alone for clarity; Supplementary Figure 5a). Interestingly, even though silencing L2–L5 interneurons disrupted left–right alternation ("contralateral influence") while preserving intralimb coordination ("ipsilateral influence"), quantitative analyses revealed no significant difference in the number of ipsilateral-versus contralateral-projecting interneurons. Therefore, L2–L5 interneurons constitute a bilaterally distributed pathway that anatomically interconnects the rostral-to-caudal lumbar segments.

Within the cleared L2 segments, L2–L5 interneurons were primarily concentrated within intermediate laminae (Fig. 8e). To investigate this further, we retrogradely labeled L2–L5 interneurons using cholera toxin beta subunit (CTB) conjugated to either AlexaFluor-594 or -647 (Fig. 8g–l) allowing the ipsi- and contralateral subtypes to be distinguished. A total of 2737 L2–L5 interneurons were analyzed for their laminar distributions. The significant majority of L2–L5 interneurons reside within the intermediate gray matter (Fig. 8j–l), primarily in lamina VII (49.9 ± 5.2%) and to a lesser extent lamina VIII (16.8 ± 5.0%). Of the total population, 14.1 ± 1.7% were within lamina V while fewer L2–L5 interneurons were found in laminae IX–X (3.0 ± 1.1% and 1.9 ± 0.9%, respectively). This distribution was similar to that observed using IHC detection of eTeNT.EGFP-positive neurons (Fig. 7).

While both ipsi- and contralateral subtypes reside primarily within laminae V, VII, and VIII (Fig. 8k, l), we observed differences in their relative density within laminae V and VIII. Significantly more ipsilateral-projecting L2–L5 interneurons were found in lamina V (18.9 ± 3.6% vs. 8.9 ± 1.6%, $p \leq 0.001$, two-way analysis of variance (ANOVA), Bonferroni post hoc $t$-test) while significantly more contralateral-projecting L2–L5 interneurons populated lamina VIII (28.0 ± 5.8% vs. 6.2 ± 4.0%, $p \leq 0.001$). This difference is clearly illustrated in heatmaps generated to visualize the L2–L5 interneuron population density within the

spinal gray matter. We found that the majority of ipsilateral interneurons resided adjacent to the central canal (Fig. 8k, lower panel) while the contralateral subtype densely populated the ventromedial gray matter below the central canal (Fig. 8l).

Previous studies performed in the isolated lumbar spinal cord of the neonatal rodent showed that descending commissural interneurons have collaterals within a segment of their somata[13, 14, 23]. Therefore, we hypothesized that L2–L5 interneurons would have dense projections locally within the rostral lumbar cord, an area critically involved in central pattern generation[24–27]. To test this, unilateral injections of CTB AlexaFluor-488 were given at L1 (Fig. 8m, n) in the same animals that received the aforementioned bilateral L5 injections (Fig. 8g–l). Any double-labeled neurons (Fig. 8o, q) would reflect an L2–L5 interneuron with a local projection (Fig. 8r, ~one segment above somata).

Of the labeled L2–L5 interneurons, few had resident collaterals (Fig. 8s). More local projections arose from the contralateral L2–L5 interneurons as compared to the ipsilateral (Fig. 8t). These collaterals were primarily commissural in nature (Fig. 8v), representing the most abundant projection pattern observed (Supplementary Figure 5c). Alternatively, the ipsilateral-projecting L2–L5 interneurons primarily had local collaterals ipsilateral to their somata (Fig. 8u). This purely ipsilateral pathway constituted the second-largest projection pattern observed (Supplementary Figure 5c). Notwithstanding, the proportion of L2–L5 interneurons with at-level projections appears to be small. These data illustrate that the majority of L2–L5 interneurons lack local projections in the rostral segments of the lumbar cord with only 12% having at-level projections that anatomically connect the two sides of the spinal cord.

Finally, we set out to identify synaptic inputs onto L2–L5 interneurons. They receive both excitatory (Fig. 8y) and inhibitory (Fig. 8z) drive, and appear to receive both 5HT (Fig. 8w) and vGLUT1-positive inputs (Fig. 8x). These data suggest that L2–L5 interneurons receive both descending and afferent inputs, although the direct source of vGLUT1–positive terminals cannot be identified. Together, these data show that L2–L5 interneurons are a bilaterally distributed pathway that resides primarily within the intermediate gray matter, an area critically involved in locomotion[24, 27–29]. The ipsi- and contralateral subtypes populate relatively discrete areas within the spinal gray matter with sparse projections throughout the rostral lumbar enlargement. Concomitant to their heterogeneous outputs[10, 11, 13–15], L2–L5 interneurons also receive a diverse set of inputs, suggesting that this pathway is highly integrated into the hindlimb locomotor circuitry.

## Discussion

Rhythm and pattern are precisely controlled by the hindlimb locomotor circuitry[1]. Together, these features change with speed in a stereotypic fashion[7]. This is a fundamental principle of locomotion[5]. We show here that a discrete component of stepping, left–right alternation, can be manipulated without influencing the other central features. This key finding is the focus of our discussion below.

Reversible silencing of spinal interneurons reveals that left–right alternation can be discretely manipulated independent of rhythm. There are five crucial findings that support this concept. The first is also the most obvious: effective locomotion continued during silencing. If L2–L5 interneurons were a part of the rhythm generating circuitry, silencing would have greatly impeded or even prevented the animals' ability to step. Secondly, the hindlimb/forelimb step ratio remained 1:1. This indicates that all four limbs stepped equally with no missteps or double steps, "mistakes" that would have affected the overall rhythm. Thirdly,

the changes in hindlimb alternation were not associated with step frequency or speed. As a function of speed, an increase in step frequency occurs along with changes in gait-specific coordination patterns (out of phase to in-phase)[7]. Fourthly, the step-by-step shifts in left–right coordination occurred alongside invariable changes in stride time, a defining feature of rhythm[30]. Finally, the relationship between speed and spatiotemporal (gait) indices was not affected. These findings suggest that within a stable locomotor

rhythm, silencing L2–L5 interneurons has affected patterned left–right movements. Moreover, the silencing did not affect the other pattern of stepping: intralimb coordination. Coordinated activity across the hindlimb joints persisted during silencing, as neither the overall range of motion nor the coordination between the proximal and distal angles were altered.

From these data, where do we place L2–L5 interneurons within the locomotor hierarchy? We propose that L2–L5 interneurons

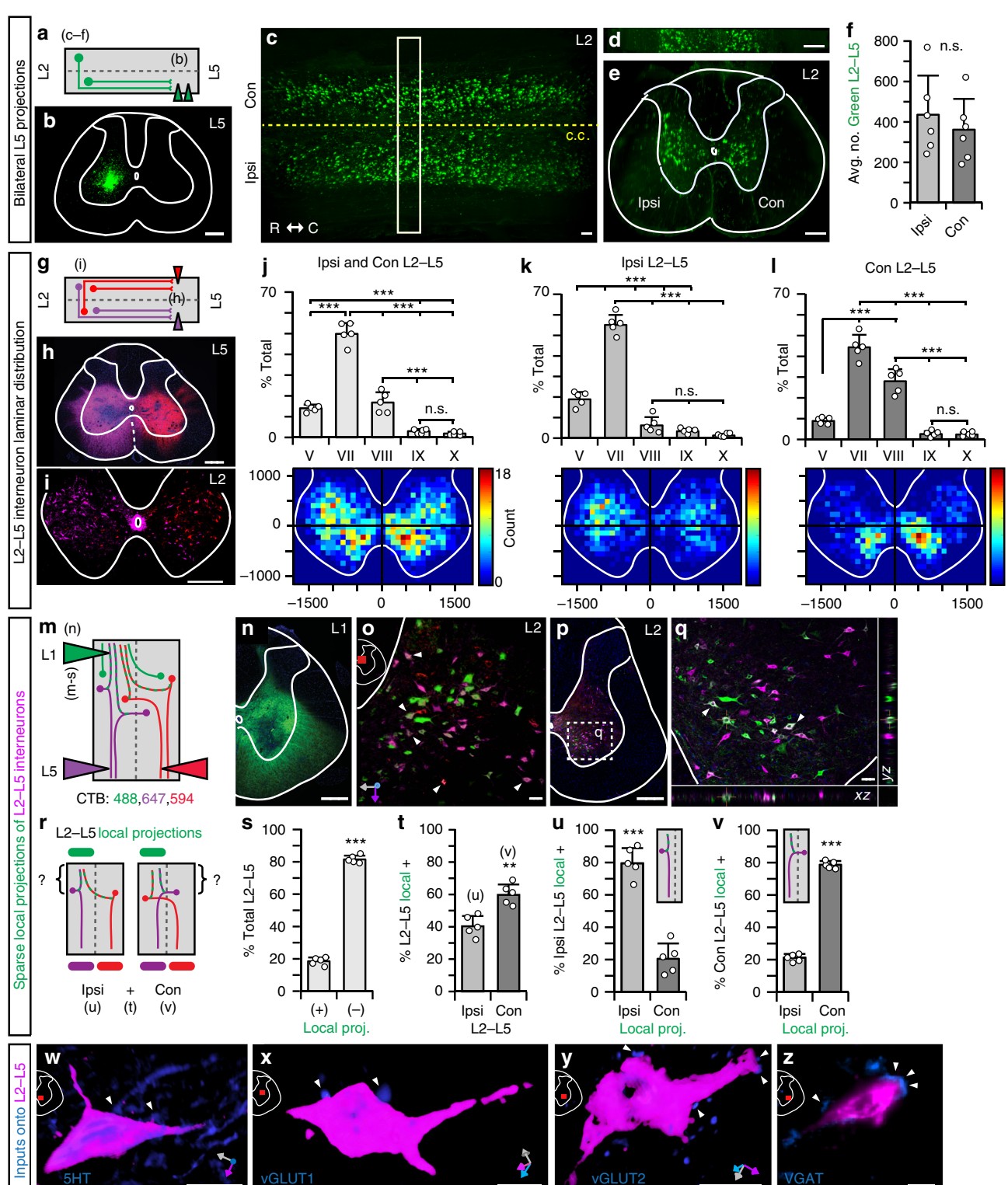

should be considered part of the left–right pattern formation layer[31] where they help to secure left–right alternation on a step-by-step basis during overground locomotion. This places them functionally "below" the rhythm generating circuitry and suggests that they distribute temporal information caudally from the rostral lumbar segments. We cannot reconcile any discrete role(s) the ipsi- and commissural subtypes might sub-serve as both were targeted in this network-focused functional dissection.

These results are in stark contrast to our hypothesized role for these interneurons in intralimb (flexor–extensor, multi-joint) coordination, anticipated based on previous studies that utilized in vitro neonatal rat and mouse spinal cords[10–12]. These studies explored the intrinsic properties, firing patterns, and synaptic output of commissural "L2–L5 interneurons". Based primarily on the timing of their activity and output onto L4/5 motoneurons during drug-induced locomotor-like activity, these interneurons were assigned putative roles in the flexor–extensor aspects of pattern formation. In turn, this was the framework on which we set out to assess their functional role in the mature, freely behaving rat. Our findings clearly illustrate the importance of taking hypotheses formed at the cellular and network levels and testing them at the systems level. However, significant caveats remain. We cannot reconcile why, at most, 30% of the hindlimb steps were disrupted. This may reflect a practical limitation of the dual-virus silencing system and/or the probability that functional populations of interneurons are unlikely to be purely segmentally defined. It is also possible that we achieved the most robust phenotype possible in our model given that the full ensemble of pathways that secure left–right coordination remains unknown. These issues could potentially be addressed by relating the magnitude change in left–right coordination back to the absolute number of double-infected interneurons (via IHC). However, the lentiviral package is randomly inserted into the host genome resulting in widely variable post hoc amplification of eTeNT. EGFP and inconsistency in the signal to noise ratio. Consistency in expression levels and a high signal to noise ratio are requirements for rigorous cell quantification, salient criteria our histological data to not meet.

Another unexpected result was the striking contrast between stepping and swimming. Swimming is primarily a bipedal (hindlimbs) locomotor task characterized by highly stereotypic rhythmic, left–right alternation[32] and flexion–extension durations that are distinct from those of stepping[33, 34]. During swimming, the limbs are unloaded, which greatly reduces signaling from Golgi tendon organs, a sensory system that typically conveys information about dynamic loading of the limbs[33]. The extension phase is dramatically reduced as compared to stepping[33] and while the duration of limb flexion is similar between the two behaviors, muscle recruitment patterns are distinct[34]. Therefore, while both behaviors can take the form of rhythmic left–right hindlimb alternation, it is clear there are stark differences underlying the expression of these alternating movements as silencing L2–L5 interneurons significantly alters stepping alone. In light of these differences, we cannot say that the lack of a phenotype during silencing reflects a lack of involvement of L2–L5 interneurons in hindlimb alternation during swimming. However, it is clear that the effects of silencing L2–L5 interneurons on hindlimb alternation are pronounced during stepping and inconspicuous during swimming.

Cracking the genetic code for programmed cell fate in the mouse spinal cord has enabled the selective ablation of various classes of transcriptionally specified neurons[2]. Of consequence to left–right alternation are the V0 and V2a interneurons. At reduced speeds and step frequencies, the inhibitory V0 interneurons ("V0d") secure the homologous limb pairs in alternating pattern indicative of the walking gait[7, 35]. As speed increases, the left–right limb pairs are secured in a faster-paced alternating pattern (trot) through the combined actions of the excitatory V0 subclass (V0v) and excitatory V2a interneurons[36–40]. We do not know whether L2–L5 interneurons modulate the activity of this genetically delineated locomotor network, or if they constitute a mixed population of V-class subtypes. Nonetheless, we reveal through the conditional silencing of L2–L5 interneurons that hindlimb alternation can be selectively manipulated independent of speed and step frequency. We believe this moment-by-moment flexibility in left–right coordination is a complementary feature to a speed-dependent, modular organization of the locomotor circuitry[7].

Using reversible silencing of spinal interneurons as a tool, we have revealed that a core component of the locomotor pattern can be selectively and reversibly manipulated without disrupting the other core features of stepping. The changes observed illustrate the nervous system's ability to adapt to a significant, but discrete perturbation. Therefore, the observed continuum of walking to hopping likely reflects the system's strategy to maintain effective stepping given the internal and external constraints associated with that particular behavior[41]. Furthermore, changing the behavioral conditions from stepping to swimming exposes how the functional importance of distinct pathways can be powerfully modulated by a reconfiguration of the whole system. Altogether, these data illustrate a striking freedom in an otherwise precisely controlled system, a phenomenon dependent on the behavioral context.

**Fig. 8** Anatomical characterization of L2–L5 interneurons in the adult rat spinal cord. **a** Bilateral injections of FluoroRuby (not shown) and FluoroEmerald (shown alone for clarity; **b**, injection site) were performed to retrogradely label L2–L5 interneurons for light sheet fluorescence microscopy (N = 3). **c** Cleared L2 segment (c.c. central canal, R-C rostrocaudal). **d**, **e** 100 μm cross-section. **f** Average number of FluoroEmerald-labeled L2–L5 interneurons (p = 0.434, t = 2.78, df = 4; independent t-test between means of equal variance) (refer to Supplementary Figure 4). **g** Schematic illustrating dual retrograde-labeling of L2–L5 interneurons (N = 5). **h** Representative L5 injection site and **i** bilateral labeling of L2–L5 interneurons. **j**–**l** (upper) Quantitative analyses of L2–L5 interneuron laminar distribution (N = 2737 L2–L5 interneurons; ***p ≤ 0.001, df = 40; two-way ANOVA, Bonferroni's post hoc t-test). **j**–**l** (lower) Heatmaps of L2–L5 interneuron distribution (each "pixel-bin": 100 × 100 μm). **m** Animals shown in **g** also received unilateral injection of CTB-Alexa-488 at L1 (**n**, representative injection site) to label L2–L5 interneurons with local projections. **o** Three-dimensional rendering of ipsi- (red) and contralateral (magenta) L2–L5 interneurons that are positive (arrows) as well as negative for local L1 projections (x–y–z axes shown in gray/magenta/blue). **p**, **q** Colocalization of 488 with 594 or 647 (arrows) confirmed in x–zand y–z planes (crosshairs, white signal). **r** Projection pattern analyses of L2–L5 interneurons. Quantification of L2–L5 projections shown in **r** are as follows: **s** p = 1.09E−10, **t** p = 0.0013, **u** p = 9.23E−10, **v** p = 1.61E−10; each with critical t = 2.306, df = 8; one-way ANOVA with Tukey's HSD post hoc t-tests; **p ≤ 0.01, ***p ≤ 0.001). **w**–**z** Immunohistochemical detection of plausible synaptic inputs onto L2–L5 interneuron cell bodies (**w** serotonergic inputs (5HT); **x** vesicular glutamate transporter 1 (vGLUT1, proprioceptive); **y** vesicular glutamate transporter 2 (vGLUT2, excitatory); **z** vesicular GABA transporter (⌈VGAT, inhibitory); L2–L5 interneurons labeled from experiment shown in **l**). Images shown in **w**–**y** are three-dimensional volume rendered images were taken at 100x magnification with 0.5–1.0 μm optical slices (z-stacks of 11–38 slices). Image in **z** was taken on an inverted microscope (10×) for comparison to volumetric rendered images. (**f**, **j**–**l**, **s**–**v** circles=individual means; bars=group mean ± S.D.; light gray = ipsi; dark = contralateral L2–L5 interneurons). **b**–**e**, **h**, **i**, **n**, **p** scale = 100 μm; **o**, **q**, **w**–**z** scale = 10 μm

## Methods

**General procedures**. Procedures were performed in accordance with the Public Health Service Policy on Humane Care and Use of Laboratory Animals, as well as the University of Louisville Institutional Animal Care and Use and Institutional Biosafety Committees. A total of $N = 16$ adult female Sprague Dawley rats (200–220 g; Envigo, IN, USA) were used. Animals were housed two per cage with ad libitum food and water under 12 h light/dark cycle. This project utilized Kentucky Spinal Cord Injury Research Center Neuroscience core facilities that were supported by P30 GM103507.

**Viral vector production**. Plasmid vectors were provided by Isa and colleagues[9]. HiRet-TRE-EGFP.eTeNT ($1.6 \times 10^7$ vp/ml) and AAV2-CMV-rtTAV16 ($4.8 \times 10^{12}$ vp/ml) were built following previously described methods[42, 43].

**Intraspinal injections of viral vectors**. Power analysis of previously obtained gait data revealed $N = 6$–8 was sufficient to detect a significant difference with 90–99% power. $N = 6$ rats were anesthetized (sodium pentobarbital, 50 mg/kg i.p.; Oak Pharmaceuticals Inc., IL, USA), placed into a custom-built spinal stabilization unit[44], and received a T13 laminectomy (rostral half) to expose spinal L5. HiRet-TRE-EGFP.eTeNT was bilaterally injected (0.5 μl/site, 1.5 mm rostrocaudal) into the intermediate gray matter (0.4 mm mediolateral, 1.4 mm dorsoventral) using a stereotaxic device (World Precision Instruments, FL, USA)[45]. Injections were given in two, 0.25 μl boluses with 3 min incubations to allow for viral uptake. The incision site was sutured in layers and the wound was closed with surgical staples. Postoperative gentamicin (20 mg/kg, s.c.; Boehringer Ingelheim, MO, USA), glycopyrrolate (0.02 mg/kg, s.c.; Butler Schein, OH, USA), and lactated Ringer's solution (10 c.c, s.c.; Hospira, IL, USA) were given. Buprenorphine (10 mg/kg, s.c.; Par Pharmaceuticals, NY, USA) was provided every 12 h for the first 48 h post surgery for pain management while prophylactic doses of gentamicin were administered for 7 days. Animals recovered voluntary bladder control within 24 h post surgery.

After 1 week, animals were reanesthetized (ketamine/xylazine; 80 mg/kg:4 mg/kg, i.p.; Hospira, IL, USA; Akorn Animal Health, IL, USA) and received a T12 laminectomy (rostral half, spinal L2) to expose spinal L2. Four bilateral injections of AAV2-CMV-rtTAV16 were given (0.5 μl/site, 1.5 mm rostrocaudal, 0.6 mm mediolateral, 1.5 mm dorsoventral). The injection protocols and postoperative care were followed as described above. The reversal agent Yohimbine (0.1 mg/kg, i.m.; Lloyd Laboratories, IA, USA) was given to counteract the effects of xylazine. Animals recovered for 9 days before pre-DOX assessments. No animals were excluded from the study based on a priori criteria of normal gait at Pre-DOX1.

**Experimental design**. Doxycycline hydrochloride (DOX, 15 mg/ml; Fisher Scientific BP26531, NH, USA) was dissolved in 3% sucrose water and provided ad libitum for 5–8 days. Approximate volumes of consumption were recorded and replenished daily. All behavioral assessments were performed during the light cycle time period (morning through early afternoon).

Functional testing was performed prior to injections (Baseline), before DOX (Pre-DOX1), during DOX (DOX1$^{ON}$D1-D8), and 1 week post DOX (DOX$^{OFF}$). Assessments were repeated 1 month later (Pre-DOX2, DOX2$^{ON}$D1–D5) to assess the reproducibility of locomotor changes. Before Baseline, animals were acclimated to the stepping/swimming chamber. Stepping was spontaneous and volitional. During all control time points, animals spontaneously walked or trotted where the left–right limb pairs moved out of phase (alternation) relative to one another. No spontaneously expressed gallop or bound gaits were observed during these time points. Animals did not receive task-specific or positive/negative reinforcement training. The order of animal testing was random. Due to the overt change in behavior during silencing, it was impossible to blind the experimenters to control versus DOX$^{ON}$ time points. Raters were blinded to animal-specific outcomes across time points. Each animal served as its own control based on the following: each surgery is unique concerning the proportion of total L2–L5 interneurons that are double infected, there is inherent variability in transgene expression across animals, and there exists normal interanimal variability in behavior. Control (Baseline, Pre-DOX1, DOX$^{OFF}$, Pre-DOX2) versus experimental (DOX1$^{ON}$D1–8, DOX2$^{ON}$D1–5) time point comparisons were made on an individual and group basis. Group data are shown.

**Three-dimensional hindlimb kinematics**. Hindlimb kinematic analysis was performed as previously described[16, 32]. Briefly, the skin overlying the anterior rim of the pelvis (iliac crest (I)), head of the greater trochanter (hip (H)), lateral malleolus of the ankle (A), and the metatarsophalangeal joint of the toe (T) was marked thereby describing hindlimb movement using three segments (I–H, H–A, A–T) and two angles (proximal: I–H–A, distal: H–A–T). Animals freely walked in a Sylgard-coated (Sylgard 184 Silicone Elastomer Kit, Dow Corning, MI, USA) plexiglass walkway tank (length×width×height: 150 cm, 30 cm, 14 cm) while high-speed (100 frames/s) videos were acquired from one ventral and two sagittal viewpoints. Videos were analyzed using MaxTraq, MaxMate, and MaxTraq3D software (Innovision Systems, MI, USA). A minimum of two stepping passes per hindlimb were analyzed that met the following criteria: animal walked at least three-fourths the length of the tank, or approximately 1 m (4–5 contiguous step cycles recorded), continuous walking with no distracted behavior, trajectory was relatively straight with minimal lateral deviations, and visually representative of the animals' overall locomotor behavior.

Two-dimensional stick figures were generated for each animal across all time points (Control: $N = 48$; DOX$^{ON}$: $N = 60$). Stick figures shown are representative of control hindlimb stepping and DOX$^{ON}$ hindlimb synchronous stepping.

To clarify movements of the three limb segments and two angles during locomotion, schematics were generated from five select moments throughout the step cycle (Fig. 2). These events include the end of stance phase, mid-swing phase, start of stance phase, mid-stance phase, and end of stance phase (defined from ventral recordings). Concomitant with these tracings are the two-dimensional hindlimb schematics of the locomotor bout in which these events occurred (Fig. 2, vertical limb movements (y-axis) across time (x-axis)).

Intralimb coordination was described by two features: temporal coordination between the proximal and distal hindlimb angles during each step cycle (Fig. 2) and peak-to-trough excursions, or range of motion (Supplementary Figure 5). To measure the temporal coordination between the angles, the duration of peak-to-peak excursions of the distal angle were measured for each step cycle. Within this peak-to-peak interval, the maximal excursion of proximal angle was identified. By dividing the onset time of peak proximal excursion by the peak-to-peak distal excursion, the temporal relationship between these two intralimb angles can be quantified. This intralimb phase value ranges from 0 to 1 wherein 0/1 denotes normal, coordinated movements between the two angles, while 0.5 indicates out-of-phase, dys-coordinated movements. Values were graphed on a polar plot and analyzed using circular statistics. Data from the left and right hindlimbs are shown (Fig. 2). To analyze the range of motion for the proximal and distal angles, the peak-to-trough excursions of the proximal and distal angles were calculated throughout the step cycles for each animal across all time points. Individual and group means were calculated from these events. Group data for the left and right hindlimbs are shown (Supplementary Figure 5, top of box plot denotes mean ± S. D. peak excursion, bottom of box plot denotes mean ± S.D. trough excursion). No significant differences between the left and right hindlimbs were found across all intralimb measures.

**Volitional overground gait analysis**. Ventral recordings were used to analyze the timing of individual paw contacts and lift offs. A minimum of 4 passes (or 8 step cycles, defined as stance+swing) were analyzed per animal following the previously defined criteria. Stereotypic exploratory behavior was qualitatively defined as frequent-to-consistent pausing/hesitation that was concomitant with either visual distractions or interactions with the external environment (e.g., sniffing, licking). Stepping passes displaying these overt behaviors were not analyzed. The classic gaits illustrated in Fig. 4 were generated from a separate study where $N = 12$ adult female Sprague Dawley rats were trained using positive reinforcement and food rewards to step across a long runway track.

Gaits are described with respect to the hindlimbs. The walk–trot gait reflects: (1) alternating[5] and (2) out-of-phase movements of the left–right limb pair (phase value = 0.5)[46] ("symmetric" gait[47]). The bound gait is described by left–right hindlimb movements that are (1) synchronous and (2) in phase (interlimb coordination value = 0/1). Gallop ("intermediate" gait[7]) is described as (1) asynchronous movements wherein the hindlimbs step at a (2) slight phase shift (interlimb coordination = 0.25/0.75).

**Step sequence pattern (SSP) analysis**. SSP describes the order 0 at which all four limbs step[17]. There are three main SSPs (alternate, cruciate, rotate) and each pattern is subdivided based on which limb initiates the step sequence (e.g., alternate pattern subtype "a" versus "b," wherein the right or left forelimb initiates the step sequence, respectively). SSPs do not reflect interlimb coordination or the swing–stance durations. To analyze the SSP, the limb that initiates the locomotor bout was identified. Thereafter, the subsequent footfall order of the remaining limbs was determined. These analyses were repeated for all step cycles analyzed across all animals and time points ($N = 6$–12 step sequence patterns/animal/time point, $N = 44$–66 total step sequence patterns/time point). After calculating the number and type of SSP each animal expressed per time point, the group averages and standard deviations were calculated. No rotary SSPs were observed throughout the study. SSPs are illustrated as a series of circles that are interconnected by lines to show the order of limb recruitment.

**Hindlimb coordination analysis**. To quantify hindlimb coordination, the time of initial contact for the left hindlimb was divided into the length of time for one complete stride cycle of the reference right hindlimb. The following are potential issues regarding phase analyses in freely stepping animals: (1) interchangeability in lead limb and (2) basal level of variability in alternation. To normalize the limbs to account for any interanimal variability, the 0–1 phase data was transformed to a linear scale of 0.5–1.0 (convert <0.50 to its reciprocal >0.50). To quantify silencing-induced changes in coordination beyond normal variability, all control time points (baseline, pre-DOX1, DOX$^{OFF}$, and pre-DOX2) were averaged (hindlimb mean = 0.55; forelimb mean = 0.54). Any phase value >2 S.D. from this mean is irregular (hindlimb: >0.63; forelimb: >0.62). The proportion of phases >2 S.D. were compared across time points for forelimbs and hindlimbs during stepping. Raw phase data were used for circular statistics.

**Magnitude change in alternation and interanimal variability**. To compare the proportion of disrupted steps observed at DOX1$^{ON}$ and DOX2$^{ON}$, the total number of silencing-induced disrupted steps (relative to total steps taken) were calculated at DOX1$^{ON}$D3–D8 and DOX2$^{ON}$D3–D5, respectively (calculated on an individual animal (interanimal variability) and group basis). The group peak effect (overall maximal disruption to hindlimb alternation) was observed during DOX1$^{ON}$-D5. For individual animals, maximal disruptions to hindlimb coordination were observed at the following time points: DOX1$^{ON}$-D5 (3 animals), DOX1$^{ON}$-D8 (1 animal), and DOX2$^{ON}$-D5 (2 animals).

**Coefficient of variation (COV)**. COV describes the amount of variability relative to the mean[36, 48]. To measure hindlimb coordination variability during stepping, each animal's mean and S.D. were calculated across all Control and DOX$^{ON}$ time points, respectively (raw and transformed phase data). The COV was calculated by dividing each animal's S.D. by its mean (value represented as a percentage). Group COV data were compared using a paired $t$-test to detect a significant difference in variability in left–right coordination.

**Poincaré plots**. A subset of hindlimb phase data from Control and DOX$^{ON}$ time points were selected for Poincaré plot generation (Supplementary Figure 2; $n = 95$ and 96 total step cycles, respectively; the total number step cycles sampled are derived from collection of multiple individual locomotor bouts)[49]. Poincaré plots illustrate time series data with each data point reflecting a pair of successive events (e.g., sequential steps; $x$-axis step $n$; $y$-axis step $n + 1$). The shape of the plot describes the "evolution of the system" or the inherent variability in the time series data[50].

**Per-step changes in left–right hindlimb coordination**. As a discrete measure of variability in left–right hindlimb coordination, the moment-by-moment changes in hindlimb coordination were calculated with each successive step cycle. Hindlimb phase values were calculated from the time of initial contact for the left hindlimb divided into the length of time for one complete stride cycle of the reference right hindlimb. Next, the absolute change in phase was calculated on a step-by-step basis and then plotted over time. Any value >2 S.D. from the control mean change is plotted in the shaded area (hindlimb mean = 0.043, S.D. >0.113; forelimb mean = 0.066; S.D. >0.131). A subset of these data were selected for illustration where the $x$- and $y$-axes were extended to aid in visual representation (Supplementary Figure 2).

**Locomotor speed and the spatiotemporal gait indices**. To measure changes in locomotor velocity during Control and DOX$^{ON}$ time points, the overall speed (speed of a one complete locomotor bout, start to finish) and instantaneous speed (speed on a step-by-step basis) were calculated. Individual and group mean overall speeds were calculated at each time point. Group mean and S.D. data are shown (Supplementary Figure 5). The change in minimum and maximum speeds expressed (from Control to DOX$^{ON}$) are shown.

Comparisons were made with the transformed phase data set to determine if silencing-induced changes correlated with speed or various gait parameters. For the individual step data set, hindlimb phase values were compared to the instantaneous speed (distance traveled per step over time, centimeters/second), step frequency (inverse of the time for one stride cycle, Hertz), or stride length (distance traveled per step, centimeters) for all control and all DOX$^{ON}$ time points, respectively. Individual time point comparisons were also made (Supplementary Tables 1–2). Data were plotted as three-dimensional scatter graphs as well as two-dimensional contour plots with speed shown in color. The frequency of coordination values observed were analyzed with respect to speeds ≤90 cm/s (walk–trot[51–54]) and >90 cm/s (gallop–bound[51, 52, 54]) at individual as well as collapsed control and DOX$^{ON}$ time points. The individual step data sets were averaged and then processed for correlations with and without controlling for speed.

**Swim phase analysis**. The stepping tank was filled with 7–8 inches of 25–28 °C water and a neoprene-covered exit ramp was attached to one end. A high-speed camera was set up 18 inches from the tank to record the swim passes. A minimum of 4 passes (or 8 stroke cycles) were analyzed per animal following the previously defined criteria. The peak downward extension of the toe was digitized for both hindlimbs to determine the phase relationship during swimming. The time of peak downward extension of the left toe was divided into the length of time for one complete stroke cycle of the reference right hindlimb. Values were transformed as described above and the proportion of phases >2 S.D. from transformed control mean were compared across time (mean = 0.54; 2 S.D. >0.64).

**Viral tissue processing**. Animals were killed after terminal assessments at DOX2$^{ON}$-D5 with an overdose of sodium pentobarbital, then transcardially perfused with 0.1 M phosphate-buffered saline (PBS) (pH 7.4) followed by 4% paraformaldehyde. Spinal cords were dissected, post-fixed overnight, and transferred to 30% sucrose for 3–4 days at 4 °C. Spinal segments L1–L6 were dissected, embedded in tissue freezing medium, cryosectioned at 30 μm in 5 sets, and stored at −20 °C.

**IHC detection of eTeNT.EGFP**. Putatively silenced synapses were detected with a 3-day staining protocol. Day 1 consisted of hydration, blocking, and an overnight incubation with a solution containing rabbit anti-GFP (1:5000) and guinea pig anti-NeuN (1:500). Day 2 consisted of three, 5 min washes (0.1 M PBS) followed by a second overnight incubation with mouse anti-synaptophysin at 1:10,000 (Millipore MAB5258-50UG). Secondary antibodies were applied during day 3 (anti-rabbit AlexaFluor-488, anti-guinea pig AlexaFluor-594, and anti-mouse AlexaFluor-647; Jackson Immunoresearch 715-605-151).

To detect putatively silenced synapses onto caudal lumbar motor neurons, we modified our standard IHC protocol[55]. Here, we used PBS and Tween-20 (0.1%, dissolved in PBS) instead of Tris-buffered saline (TBS) and Triton X-100. In select cases, no detergent was used (dependent on signal to noise ratio). With this protocol, sections were stained with anti-vesicular acetylcholine transporter (VAChT; guinea pig, 1:100; Millipore AB1588) as a marker for synapses onto motor neurons[56]. This was used in conjunction with rabbit anti-GFP (1:5000) and mouse anti-NeuN (1:200; Millipore MAB377). This protocol was used to detect putatively silenced, excitatory and inhibitory synapses onto neurons through co-staining with guinea pig anti-vesicular glutamate transporter 2 (vGLUT2; 1:5,000; Millipore AB2251-I) or goat anti-vesicular GABA transporter (VGAT; 1:500, Frontier Institute, VGAT-Go-Af620). Secondary antibodies used were donkey anti-rabbit AlexaFluor-488 (1:200), anti-mouse AlexaFluor-546 (1:200; Invitrogen, A10036), anti-guinea pig AlexaFluor-647 (1:200; Jackson Immunoresearch, 706-606-148), and anti-goat AlexaFluor-647 (1:200; Jackson Immunoresearch, 705-605-147), respectively.

Negative controls include nonimmune sera (anti-rabbit IgG: Jackson Immunoresearch 711-005-152; anti-guinea pig IgG: Santa Cruz Biotechnology sc-2711, TX, USA; anti-mouse IgG: Jackson Immunoresearch 715-007-003). The entire lumbar neuraxis (L1–L6) as well as sacral and mid-thoracic segments were screened for eTeNT.EGFP-positive, terminal-like signal using fluorescent and immunoperoxidase reaction protocols (described in detail below). Antibody validation was provided through vendor production sheets.

Fluorescent images were captured using an Olympus FluoView 1000 confocal microscope with the oil immersion 100× objective using 488, 543, and 647 lasers (Olympus, PA, USA) (refer to the "Confocal microscopy equipment and settings" section below for further detail). Z-stacks acquired were 53–68 slices at 0.4 μm optical steps for neurons within the intermediate gray matter. Motor neuron z-stacks were 80–106 optical slices. The raw.oif files were imported into Amira 3D software (Visualization Sciences Group, MA, USA) for volume rendering to qualitatively assess the relative density and distribution of EGFP.eTeNT-positive terminals onto neurons throughout laminae V–IX at spinal L4–5. The 3D images were rotated about the $x$-, $y$-, and $z$-axes to verify close apposition of eTeNT-positive putative terminals onto NeuN-positive somata and primary dendrites.

To confirm colocalization of eTeNT.EGFP with VAChT, synaptophysin, or vGLUT2, individual.tif files were imported into Nikon NIS-Elements using the ND convert macro to generate an. nd2 stack. The .nd2 stacks were screened for candidate synapses that showed colocalization of eTeNT.EGFP with VAChT. The stacks were rendered using the slice view function to generate orthogonal crosshairs that allowed confirmation of colocalization (white signal per the overlap of green and far red secondary antibodies) in both the xz and yz planes. Images shown are from select optical slices throughout the z-stack.

Using a modified immunoperoxidase protocol we screened for double-infected cell bodies throughout L1-rostral L3, eTeNT.EGFP-positive, terminal-like structures in caudal L3–S3, and eTeNT.EGFP signal in spinal segments distal to where the L2–L5 interneurons reside (mid-thoracic, sacral).

Slides were removed from −20 °C and "super warmed" at 67 °C for 30 min. Heat-induced epitope retrieval (HIER) was performed by incubating the slides in a boiling solution of sodium citrate buffer for 40 min (pH 6.0, 95–100 °C with gentle mixing using a magnetic spin vane). If there were issues with sections remaining adherent to the slides, the slides were cooled for 30 min at room temperature after HIER.

The sections were then outlined with a hydrophobic slide marker (Pap Pen, Research Products International Corporation, IL, USA) and transferred to a humidor chamber. Sections were rinsed with 0.1 M PBS (pH 7.4) for two, 5 min washes and then treated with 3% hydrogen peroxide for 10 min. We then used the 3,3′-Diaminobenzidine (DAB) IHC Select kit (DAB500, Millipore), but modified the company protocol (increased the duration of blocking, secondary antibody, streptavidin horseradish peroxidase, and DAB incubations to 10 min each). Primary antibody (rabbit anti-GFP at 1:5000 in a solution of 0.1 M PBS, 0.5% bovine serum albumin, and 5% normal goat serum) incubated overnight at 4 °C with gentle shaking. The negative control was nonimmune rabbit sera (anti-IgG at 5 mg/ml, 1:5000).

Prior to DAB immunoreaction, the slides were transferred from the large humidor chamber to smaller trays. During the chromagen reaction, slides were shaken vigorously and rotated clockwise every 2 min during 10 min of incubation to prevent the settling of precipitated DAB onto the sections. The reaction was terminated by pouring rinse buffer into the tray. The solution was poured off, fresh rinse buffer was applied, and slides were vigorously shaken for 30 min. Sections were then treated with an increasing ethanol gradient (75, 95, and 100%; each 3 min) followed by xylenes (3 min). Slides were coverslipped with Permount (ThermoFisher Scientific, SP15-500, MA, USA) and air dried overnight. Images were acquired using a Nikon Eclipse E400 microscope connected to a Spot RT

CCD digital camera with 10×, 20×, and 40× objectives (refer to "Upright microscopy equipment and settings" section for more detail).

**Neuronal labeling, light sheet microscopy, and cell counts**. A total of $N = 5$ rats were anesthetized with ketamine/xylazine (80 and 4 mg/kg; i.p.). The surgical methods and injection coordinates used for the fluorescent dextrans were identical to that of the lentiviral vector. Then, 10% FluoroEmerald (ThermoFisher Scientific, D1820; dissolved in sterile water; Butler Schein, cat. no. 002488, OH, USA) and 10% FluoroRuby (Fluorochrome; CO, USA) were injected on the left and right sides of the spinal L5 with respect to the dorsal viewpoint.

After 2 weeks of retrograde transport, animals were killed and spinal cords dissected as described above. The L5 segment was processed for histology as described above. Slides were hydrated in 0.1 M TBS, coverslipped, and imaged for a priori inclusion criteria of injection site accuracy (spinal level, laminae VII–VIII; no animals were excluded from analyses). Images were captured using a Nikon TiE 300 inverted microscope with the 10× objective and green fluorescent protein (GFP) and TexRed filter settings (Nikon, NY, USA) ("Inverted microscopy equipment and settings"). The entire L2 segment was isolated and optically cleared following previously described methods[57]. Thereafter, the entire L2 segment was isolated and optically cleared following previously described methods[57].

Images were acquired using light sheet fluorescence microscopy with 488 and 594 lasers (La Vision BioTec, Germany) as previously described[57]. Images were imported into Bitplane Imaris software system (Oxford Instruments, CT, USA) for analysis. Counts of FluoroRuby and FluoroEmerald-labeled neurons were performed manually, blinded to ipsi- and contralateral designation. Of the 4783 L2–L5 interneurons counted, none had dual ipsi- and contralateral projections. Data shown are from $N = 3$. For clarity, FluoroEmerald is shown alone. Power analysis revealed that $N \geq 38$ animals would have been necessary to detect a significant difference between ipsi- and contralateral L2–L5 interneurons (power ≥80%). Figure 4 schematics and definition of ipsi- and contralateral are with respect to the in vivo injection site as opposed to the dorsal viewpoint during surgery.

**Neural labeling, cell counts, heatmaps, and histology**. Power analysis demonstrated that $N = 3$–5 animals was sufficient to detect a significant difference (power >99%) in the number of L2–L5 interneurons that were positive versus negative for local projections. Thus, $N = 5$ rats were anesthetized with ketamine/xylazine and received a complete T12–T13 laminectomy to expose spinal L1 through L5. CTB subunit (type B, 1% solution in 0.1 M PBS at pH 7.4; Molecular Probes, OR, USA) conjugated to the following AlexaFluors were used (with respect to dorsal viewpoint): CTB-488 at right L1 (0.5 mm mediolateral; 1.3 mm dorsoventral), CTB-594 at left L4–5, and CTB-647 at right L4–5 (both at 0.5 mm mediolateral and 1.4 mm dorsoventral). Two different CTB fluorophore conjugates were used to distinguish between ipsi- and contralateral L2–L5 interneurons (ipsi-CTB 594, contra-CTB 594, ipsi-CTB 647, and contra-CTB 647). We chose to use L1 as our injection site to identify local L2–L5 interneuron collaterals for the following reasons: the rostral lumbar circuitry is critical for locomotor generation[24, 25, 58], it is well documented that local collaterals typically branch off within 1.5 segments of their cell body[13, 23], and injecting at L1 would permit bilateral counts at L2. Injecting at level with the L2–L5 interneurons would have greatly limited our analyses. After 2 weeks of retrograde transport, animals were killed following methods described above.

Spinal T13-L6 was dissected and post-fixed in 4% paraformaldehyde for 1 h followed by cryopreservation in 30% sucrose. The cords were embedded and sectioned at 30 µm in 5 sets such that adjacent sections were 150 µm apart rostrocaudally. All animals met a priori inclusion criteria for injection site accuracy.

Colocalization of CTB-488 with CTB-594 or CTB-647 was confirmed using an Olympus FluoView 1000 confocal microscope with a water immersion 20× objective and 488, 543, and 635 lasers ("Confocal microscopy equipment and settings"). The z-stacks acquired (10–15 slices at 4 µm optical steps) were imported into Nikon NIS-Elements software and rendered using the slice view with orthogonal crosshairs to illustrate colocalization. After confirmation of colocalization, one complete set was hydrated, stained with Hoechst (1:1000; ThermoFisher Scientific, #62249), coverslipped, and imaged with the 10× objective on the inverted microscope using the 4',6-diamidino-2-phenylindole (DAPI), GFP, TexRed, and CY5 filters ("Inverted microscopy equipment and settings"). Power analysis showed that analyzing $n = 5$–7 sections/animal could detect a significant difference (94–99% power) for the following: (1) %L2–L5 positive versus negative for local projections; (2) %L2–L5 positive for local projections ipsilateral versus contralateral; (3) %ipsi-L2–L5 positive for ipsilateral-local versus contralateral-local; and (4) %contra-L2–L5 positive for ipsilateral-local versus contralateral-local. CTB-labeled neurons were counted in 7 randomly selected sections per animal throughout spinal L2 as defined by Rexed laminae. Counts were performed in a blinded fashion on images taken using fixed exposure time and look-up table (LUT) settings. The following a priori criteria were used: the CTB-positive neurons must be located in laminae V–X, must colocalize with the nuclear marker Hoechst or have an overt nucleolar space, and the strength of the CTB signal should be approximately two times greater than the immediate background shown quantitatively with the horizontal intensity profile function. The following CTB-labeled neurons were counted: 488$^+$, 594$^+$, 647$^+$, 488$^+$/594$^+$, and 488$^+$/647$^+$. A total of 5884 L2–L5 interneurons and 2961 resident L2 neurons (488 alone) were

counted. Respect is paid to the in vivo injection site with regard to schematics and ipsi- and contralateral categorization. Images shown are representative. Data shown are proportional counts of L2–L5 interneurons.

L2 laminae borders[59] were traced onto a template that was then placed over the image to identify where the L2–L5 interneuron cell bodies were located within the gray matter. Counts were performed on a sampling of the L2–L5 interneurons labeled in the experiment described above (3-4 sections throughout the rostrocaudal extent of L2). In all, $N = 2737$ L2–L5 interneurons were assessed for laminar distribution throughout the spinal gray matter. Counts were performed by an experimenter who was blinded to the experimental conditions. Laminae of interest were V–X. Lamina VI was not distinguished due to the ambiguity in its laminar borders between V and VII[59].

The taxonomy feature in Nikon Elements Advanced Research (Nikon Instruments, NY, USA) was utilized to mark neurons positive for Alexa-594 or Alexa-647. Uniquely colored hashmarks were used for each fluorophore as well as for the left and right sides. These distinctive hashmarks were used to distinguish between ipsilateral and commissural L2–L5 interneurons for heatmap generation. Any neuron that showed possible co-expression of Alexa-594 and Alexa-647 were also marked during the taxonomy counts (e.g., intense red signal with faint far red signal). Confocal microscopy ruled out co-expression between these two fluorophores and these neurons were excluded from heatmap analyses. Heatmaps were generated using a custom MatLab script (v. 2016b). Absolute values of neuron position were obtained, normalized to the distance from the central canal, and then binned into 100 µm groups.

Cross-sections were stained for the following neuronal markers: excitatory[60] (guinea pig anti-vGLUT2, 1:5000, Millipore, AB2251), inhibitory[61] (rabbit anti-VGAT, 1:100, Millipore, AB5062P; goat anti-VGAT, 1:500, Frontier Institute, VGAT-Go-Af620), putative proprioceptive[62, 63] (guinea pig anti-vGLUT1, 1:750, Millipore, AB5905), and serotonergic[64] (rabbit anti-5HT, 1:10,000, Sigma-Aldrich, S5545, MO, USA). Anti-rabbit DyLight405, anti-guinea pig DyLight405, and anti-goat DyLight405 (1:100; Jackson Immunoresearch, 711-475-152, 706-475-148, and 705-475-003, respectively) were used as secondary antibodies, respectively.

Confocal images were taken at 100× magnification as described above with step sizes of 0.5 µm or 1.0 µm and optical slices ranging from 11 to 38 to mitigate photobleaching with the 405 laser. Raw .oif files were imported into the Amira 3D software system for volumetric rendering to qualitatively assess inputs onto L2–L5 interneurons through rotational changes about the $x$-, $y$-, and $z$-axes.

**Microscopy equipment and settings**. Images of immunoperoxidase-amplified eTeNT.EGFP were taken on a Nikon Eclipse E400 microscope (Nikon, NY, USA) connected to a Spot RT CCD digital camera (Flex 64Mp Mosaic model 15.2). The image acquisition software used was the SPOT Basic package (version 5.1). Using the DIA-ILL filter, images were captured at 10× (N.A. 0.30, DIC L, W.D. 16 mm), 20× (N.A. 0.50, DIC M, W.D. 2.1), and 40× (N.A. 0.75, DIC M, W.D. 0.72). The sensor pixel size was 7.4 µm by 7.4 µm at a depth of 24 bits per pixel (RGB). The color enhancement acquisition settings (gamma, contrast, and saturation) were set to 1.0 with a color temperature of 5000. Auto white balance was enabled during image acquisition with manual exposure settings ranging from 10 to 20 ms. Light settings were manually adjusted with subsequent increases in magnification. Images were saved as .tif files. The post hoc image processing was performed in the ImageJ software package (ImageJ 1.50b, NIH, MD, USA). Brightness and contrast settings were adjusted uniformly across the entire image and applied equally to the isotype control images. No pseudo-coloring or nonlinear adjustments were performed.

Images of fluorescent tracer labeling (injection sites, cell bodies) were taken on a Nikon TiE 300 inverted microscope connected to an Andor Zyla digital camera (VSC-00436). The Nikon Elements Advanced Research software package (64 bit, build 728) was used for image acquisition. Images were captured using the Plan Apo 10x objective (N.A. 0.45, DIC N1, W.D. 4.0, refractive index 1.0). GFP (excitation: 450–490 nm; emission: 500–550 nm; dichroic: 495 nm), TexRed (excitation: 540–580 nm; emission: 593–668 nm; dichroic: 585 nm), DAPI (excitation: 325–375 nm; emission: 435–485 nm; dichroic: 400 nm), and CY5 (excitation: 580–650 nm, emission: 663–737 nm; dichroic: 660 nm) filters were used accordingly. Images of FluoroRuby and FluoroEmerald injection sites were captured with exposure times of 500 and 800 ms for GFP and TexRed, respectively. Images of CTB AlexaFluor-488, -594, and -647 injection sites were each captured with an exposure time of 100 ms for the GFP, TexRed, and CY5 filters, respectively. With regard to cell counting, all images of retrogradely labeled L2–L5 interneurons were captured with exposure times of 300 ms, 1 s, and 2 s for the DAPI, GFP, TexRed, and CY5 filters, respectively. LUT settings were fixed across all images. Large multichannel images were captured at 0.65 µm/pixel with a bit depth of 16. Acquisition settings include the following: 1 × 1 binning, 16 bit rolling shutter readout mode, 560 MHz readout rate, 0.25% dual gain conversion, with a spurious noise filter (captured at −0.4 °C). 3 × 3 or 4 × 4 tiled images were stitched together via the optimal path setting with 15% overlap. Step-by-step focus was enabled wherein every 2 fields were optically configured using select filters (range: 30 µm, step size: 5.0 µm). Images were saved as .tif files. The post hoc image processing was performed using the Nikon Elements Advanced Research Image Analysis Software package (version 3.2, 64 bit). Images were adjusted for brightness/contrast (uniformly across the entire image), scaled, cropped, and saved as 8 bit files. No pseudo-coloring or nonlinear adjustments were performed.

Images of eTeNT.EGFP putative terminals were taken on an Olympus Fluoview FV1000 confocal microscope (Olympus America, PA, USA) with FV10-ASW (version 4.1) acquisition software. Images (1024 × 1024 pixels; 12 bit depth) were captured using the oil immersion UPLFLN 100× objective (N.A. 1.3, W.D. 0.20) at 0.124 μm/pixel and 20 μs/pixel (0.4 μm optical slices). Select images were acquired at 512 × 512 pixels (0.414 μm/pixel) to mitigate fluorophore quenching. Scans were acquired in a one-way direction using both the line sequential and line Kalman integration (2–3 iterations) modes. The following lasers were used, with laser power and voltage settings reported as ranges to reflect the multiple staining conditions: (1) 488 (excitation: 488 nm, emission: 520 nm) at 10–15% power and 575–600 volts; (2) 543 (excitation: 543 nm, emission: 572 nm) at 30–40% power and 715–755 volts; and (3) 635 (excitation: 635 nm, emission: 668 nm) at 20–35% power and 675–740 volts. Images were acquired in two separate scans and rendered together in Amira (488 and 635 followed by 546).

Images of double-labeled L2–L5 interneurons were captured using the UPLFLN 20× water immersion objective (N.A. 0.50, W.D. 2.1) at 1.242 μm/pixel and 20 μs/pixel (4 μm optical slices). The aforementioned acquisition settings were used, with the following adjustments to the laser settings: (1) 488 at 5–10% power and 435–600 volts; (2) 543 at 25–30% power and 595–705 volts; and (3) 635 at 5–15% power and 580–660 volts.

Images of plausible synaptic inputs onto L2–L5 cell bodies were captured using the UPLFLN 100× objective with the aforementioned acquisition settings (20–40 μs/pixel scan rate; 0.5–1.0 μm optical slice). The 405 (excitation: 422 nm, emission: 425 nm; 5–10% power and 600–640 volts) and 635 (20–25% power and 600–655 volts) lasers were used. All confocal images were saved as .oif files.

Z-stacks of confocal images were combined into a three-dimensional composite image using the Amira software package (v6.0). All image manipulations were conducted according to the instructions present in the manufacturer's user guide. Images were processed as previously described[65]. Briefly, intensity-related artifacts detected throughout the stack were mitigated using an algorithm designed to fit an exponential curve to the average intensities. Stacks were deconvolved with a point spread function based on the numerical aperture (N.A. 1.3 for 100 ), light wavelength (405, 488, 543, and 635 nm), and refractive index of the medium used (1.518 for oil) during image acquisition.

**Statistical analyses.** Statistical analyses were performed using IBM SPSS v22 software package (IBM, NY, USA). Parametric and nonparametric comparisons were performed as needed[20, 21, 66–68]. Variance was assessed through Levene's test for homogeneity of variances or the F-test, when appropriate. Differences between groups were considered statistically significant for p-values ≤ 0.05. Two-tailed p-values are reported unless otherwise specified.

**Step sequence pattern analysis.** Step sequence pattern[17] data are shown as the percent of total patterns observed and were analyzed using repeated measures ANOVA with group as a factor followed by Tukey's honest significant difference (HSD) post hoc t-test for DOX1[ON] and DOX2[ON] separately (data shown are mean ± S.D.). Data shown are from 47–66 step sequence patterns per time point (note: range due to variability in the contiguous number steps for each locomotor bout analyzed per animal per time point).

**Step and swim phase analysis.** For the raw and transformed phase data as well as the absolute step-by-step (or stroke by stroke for swim) change in coordination, significant differences in frequency of phase values >2 S.D. from control mean were detected using the Binomial Proportion Test[67]. Coefficient of variation data were analyzed using the paired t-test (Control versus DOX[ON]). Levene's Test for Equality of Variances was used to test for a normal distribution of the phase data. Note that at control time points, e.g., Baseline, the data had a non-normal distribution (highly clustered at 0.5). Outliers, any value >3 S.D. at each time point, were excluded. Mean and S.D. were re-calculated to verify no additional outliers.

**Circular statistics.** Circular statistics was performed on the raw phase data to analyze phase distribution. Classically, parametric tests are used to determine whether the circular data are from a uniform distribution. These analyses are based on strict assumptions regarding the data distribution and are restricted to either uniform or unimodal patterns[20, 21]. Our data did not fit these criteria. Furthermore, we had no evidence to support a unimodal distribution with the same degree of concentration (e.g., relative concentration at each of the four control time points). Instead, we used nonparametric circular statistics to test the null hypothesis that the two time points being compared had the same phasic direction (concentrated or clustered)[20]. Time point comparisons were performed using the nonparametric two-sample $U^2$ tests. Thereafter, all control time points and DOX[ON] time points were respectively grouped and compared as well. The "coupled" circular plot shown in Fig. 5 reflects data generated from Baseline while the "uncoupled" circular plot are theoretical results. The p-values are reported as ranges based on critical values of Watson's $U^2$ as reported in Table D.44 of Zar[20].

**Cumulative frequency distributions.** The nonparametric Kolmogorov–Smirnov (KS) test was used to compare differences in the cumulative frequency distributions of raw and transformed hindlimb phase values over time. The KS test was also used to compare cumulative frequency distributions of transformed hindlimb phases at speeds ≤90 cm/s (walk–trot) and >90 cm/s (gallop–bound) over time.

**Phase versus speed and its relationship to spatiotemporal gait indices.** The overall locomotor speed was analyzed using repeated measures ANOVA with groups as a factor followed by Tukey's HSD post hoc t-test for DOX1[ON] and DOX2[ON] separately (data shown are mean ± S.D.). Instantaneous phase comparisons were made with Spearman's rank correlation test against the following gait parameters: stance time, swing time, stride time, stride frequency, stride length, and speed. The 95% prediction intervals were used as a visual aid for distribution of speed–stride length and speed–stride time. Prediction intervals were calculated from each control data set and then superimposed onto the corresponding DOX[ON] panels. The relative percentage of swing and stance durations were compared between two speed groups (≤90 cm/s or >90 cm/s) during DOX[ON] using the paired t-test. At speeds ≤90 cm/s, the relative percent of swing and stance durations were compared between all control and all DOX[ON] time points using the independent t-test between means of equal variance. Statistical analyses of control time points at speeds >90 cm/s were not possible due to few animals stepping at this velocity. Pearson's and part correlations (controlling for speed) followed by the Bonferroni correction for Type I errors were used on the averaged data sets.

**Hindlimb range of motion and intralimb coordination.** Excursion of the proximal and distal angles for each hindlimb was analyzed using mixed model ANOVA followed by Bonferroni post hoc t-test (data shown are mean excursions ± S.D.). Left and right hindlimb HAT–IHA phase values were analyzed using circular statistics as described above. Data shown in Fig. 2 reflect comparisons between collapsed Control and DOX[ON] time points, respectively.

**L2–L5 cell counts and laminar distribution.** Absolute L2–L5 interneuron cell counts from cleared L2 segments were analyzed using the independent t-test between means of equal variance. Proportional counts of L2–L5 interneurons differentially labeled with CTB were analyzed using one-way ANOVA followed by Tukey's post hoc t-test as well as independent t-tests between means with equal variance. One-way ANOVA followed by Tukey's post hoc t-tests were performed to analyze the laminar distribution of L2–L5 interneurons. Two-way ANOVAs followed by Bonferroni's post hoc t-tests were performed to analyze the percent total of ipsilateral versus contralateral L2–L5 cell bodies that were distributed throughout laminae V–X. Data shown are mean ± S.D.

**Code availability.** Custom-built Excel add-in macros were used to analyze kinematic and gait data and custom-designed MatLab script was used to generate heatmaps. For access to these macros, please contact the corresponding authors.

**Data availability.** The authors declare that the main data supporting the findings of this study are available within the manuscript and its Supplementary Information files.

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

## Acknowledgements

We thank Drs Tadashi Isa and Akiya Watakabe for providing the viral vector plasmids, Russell M. Howard for assistance in vector production, Christine Yarberry for surgical expertise, and Josiah Hardin for data analysis support. We thank Dr. Y. Ping Zhang and Dr. Christopher B. Shields for providing the custom-built spinal stabilizers for intraspinal injections. We also thank Dr. Andrew Todd for suggesting the goat anti-VGAT primary antibody and Dr. Ron Harris-Warrick for commentary on a previous version of this manuscript. This project utilized Kentucky Spinal Cord Injury Research Center

(KSCIRC) Neuroscience core facilities that are supported by P30 GM103507 (to S.R.W.). The experiments were supported by Kentucky Spinal Cord and Head Injury Research Trust Grant 13-14, NS089324 (to D.S.K.M. and S.R.W.), Norton Healthcare, and the Commonwealth of Kentucky Challenge for Excellence (to S.R.W.).

## Author contributions

A.M.P. designed the study with input from S.R.W. and D.S.K.M., performed the experiments, analyzed the data, completed the immunohistochemistry and image processing, prepared the figures, and drafted the manuscript. S.R.W. and D.S.K.M. edited the manuscript. D.A.B. assisted in behavioral testing and was instrumental in statistical analyses. J.R.M. contributed to behavioral testing, data acquisition, and Excel macro development. J.E.B. performed the CTB cell counts and provided confocal microscopy support. A.S.R. assisted in surgery, postoperative care, behavioral testing, and data collection. P.T. performed tissue clearing, light sheet fluorescence microscopy, and absolute cell counts. G.J.R.S. developed the MatLab script to build heatmaps of L2–L5 interneuron laminar distributions. All authors read the final version of manuscript.

## Additional information

**Competing interests:** The authors declare no competing financial interests.

