## [Peer Review File · Nature Communications]

Reviewers' comments:

Reviewer #1 (Remarks to the Author):

The study by Pocratsky et al deals with the neurons that project from the L2 to the L5 segments in the rat and their effects on the locomotor pattern. When these intersegmental neurons are inactivated through doxycycline (the interneurons are affected through double-virus injections in L 5 and L 2, respectively), it leads to a change in the locomotor pattern. In the wild type the hind limbs show strict alternation as in walk or trot, and at higher speed they can break into a gallop or a bound. Animals with inactivation of L5-projecting L2 interneurons change the pattern, so that the two hind limbs are less well coordinated, with a less strict alternation even at low speeds. The pattern of activity in each of the hind limbs is on the other hand reported as unchanged. This was an unexpected outcome. The report provides a detailed analyses and the text is clear and the Figures mostly to the point

General comments:

A very large part of the current literature with genetic approaches to interneuron analyses shows the effect of deletions of the different subgroups of developmentally defined groups (V0 to V3 and different subdivisions). Although the authors do not use this approach, it is surprising that they do virtually not cite this large body of current literature. V0v and V0d interneurons are both known to be critical for alternating locomotor behaviour related to walk and trot, respectively. This would seem most relevant to discuss, but they do not even mention these important findings of Bellardita and Kiehn 2015, although they cite their study for baseline data of interlimb coordination. It would seem likely that the L2-L5 intersegmental neurons could well influence the excitability of these two types of V0 interneurons responsible for coordinating alternation in both walk and trot.

The title of the manuscript "Silencing spinal interneurons: a continuum of walking to hopping" does in the reviewer's mind is overstating the findings

Specific comments

Lines 57 – 68 and Fig 2. The authors measure "joint angles" between iliac rest – hip – ankle or hip – ankle toe omitting the knee – provides some complication for the evaluation. The movement recordings in Fig 2 are not sufficiently detailed to allow the conclusion that there are no changes in in the intralimb-coordination, although the overall movement pattern appears largely unchanged. Would require a more detailed analysis – preferably EMG.

125-183 and Fig 4-5. It is clear that after inactivation of the L2-L5 interneurons one limb in relation to the other can drift to resume the preferred alternating pattern. It would be interesting to know if the drift is gradual as in "relative coordination", when there is a gradual change over some cycles to later return to the preferred 0.5 coupling. Fig 5f contains data relevant for such an analysis– but the resolution is not sufficient – would be interesting to extend the DOX recordings so that it could be analysed/plotted in an appropriate way. When wild-type animals (intact or spinal) walk on a split belt – large differences in belt speeds can result in 2:1 couplings between the two limbs in which they maintain a preferred phase coupling. This shows that the interneuronal system can adapt beyond the fixed couplings in walk – trot and bound and in gallop the coupling is in fact more flexible.

308-316 The fact that the swimming phenotype after DOX remains strictly alternating, is interesting, but can possibly be explained by many factors – the limb movements in water becomes automatically different from that of walking with ground contact, varying load, and inevitable minor perturbations. Whether swimming actually represents a different motor program as argued, rather than an adaptation of the walking motor program to modified physical constraints appears unclear (line 316).

286-291 The authors ask themselves where they should place the L2-5 interneurons in the “locomotor hierarchy” and end up in suggesting that they should be part of the “left-right pattern information layer”. The authors have made a distinct observation regarding the effect of a deletion of the L2-L5 interneurons, but the reviewer feels it is meaningless to discuss in those soft terms, when in reality the connectivity and relation to V0 neurons should be one main primary aim to consider.

In conclusion the authors have shown that the deletion of L2-L5 interneurons leads to less secure alternating pattern, but still alternation is preferred. The connectivity of these interneurons is largely unknown both on the incoming and efferent side.

Reviewer #2 (Remarks to the Author):

Referee's report on manuscript entitled Silencing spinal interneurons: a continuum of walking to hopping by Pocratsky, Burke, Morehouse, Beare, Riegler, Tsoufas, Whittemore, Magnuson.

This study investigates the role of spinal interneurons originating in lumbar segment 2 and terminating in the fifth lumbar segment, which the authors label as L2-L5 interneurons. Many attempts have been made to decipher the role of spinal interneuron subtypes using a variety of genetic models of neonatal mice. However, this investigation constitutes the first successful attempt using genetic tools (viral transfection and reversible functional knock-down) and in an adult rat model to experimentally manipulate the activity of interneurons involved in locomotor behavior. As such this paper opens new interesting avenues in which role of specific subsets of interneurons in the freely behaving adult animal can be examined. Nonetheless, I have several issues that should be addressed by the authors.

Two main issues warrant discussion. First, the degree of transfection provided by the protocol used. The authors report (line 302) that 30% of hindlimb steps were altered by silencing the L2-L5 interneurons. Could this relatively low proportion be related to the amount of transfection and do they have any data available on this important point? To effectively address this question, the authors could compare the distribution of the number of transfected neurons with the number of FluoroRuby and FluoroEmerald-retrogradely labeled neurons in order to establish a ratio that may (or may not) correspond to the effect on actual stepping behavior. In another approach to testing the variability related to transfection level, the authors could correlate the ratio of altered steps for each animal to subsequent anatomical data from immunohistochemical detection of EGFP.eTeNT-positive neurons. In any case, even if these approaches are not technically feasible, it has to be discussed more than in a single line (see line 303).

A second important issue concerns whether the strength of the observed effect is somehow related to the localization/distribution of the locomotor-generating circuit kernel in the spinal cord. That the core of the locomotor network is located in the rostral lumbar cord is no longer an issue and this is now established and very well documented (including by the present authors in several earlier publications: e.g., Magnuson and Trinder, *J. Neurophysiol.*, 77: 200-206, 1997; Hadi et al., *J Neurosurg.*, 93: 266-275, 2000;). However, this rhythmogenic kernel is not only restricted to the L2 segment but also overlaps extensively with the L1 segment (see for example Antri et al., *Plos One*, 2011). Therefore, could it be possible that if transfection was not only restricted to L2-L5 interneurons but also included interneurons in L1, would the behavioral effects have been greater. This point, which relates to extent to which the essential components of the locomotor CPG network are targeted by the protocol employed, should be addressed.

Line 74: and throughout the manuscript: The authors have to clarify the terminology used here and

subsequently. Classical gait studies refer to 'symmetric' or 'asymmetric gaits and some authors such as Grillner (1981) refer to the alternating pattern when two limbs on the same girdle as phase-opposed. Here, it rather seems that this terminology refers to the alternation between hind-and-forelimbs. Could you be more explicit and define to what precisely your terminology refers to?

Line 75: is there a reference for the "zig-zag" pattern? Why do you not use classical phase bar diagrams with the bar indicating the time of contact for each limb. It is visually much more relevant and allows indicating SD as well see overlap between individual limbs.

Line 78: again a problem with terminology. I do not believe that 'cruciate pattern' is a standard term. Please rephrase or define.

Line 81: In this Figure, which illustrates the effect of silencing the L2-L5 interneurons as well as in the following Figures, it seems that the effect of DOX2 is somewhat weaker than the first DOX1 test. Was this systematically observed, and what could this be due to?

Line 93-103: The presentation of the Fig. 4 a-c is somewhat ambiguous. It is not clear whether these derive from actual data collected in the study or represent only a theoretical illustration. This should be clearly specified. Also, it should be mentioned whether if the tested animals were occasionally spontaneously performing galloping or bounding gaits.

Line 131-150 : The differences between the concept of "dispersion and concentration" is not obvious, and I am not sure that there is an added value to detail each item. In my opinion these two variables are redundant. Either you can strongly justify that they are complementary or please just detail one of these two variables, otherwise it is quite confusing because one expects something different from comparing "dispersion" to "concentration", and if not, why.

Line 151-174: Although single step data are of value, why did the authors not calculate the coefficient of variation, which is classically used to underline the point-by-point variability, therefore providing a meaningful index of variability?

Line 198: "...that the more synchronous gaits...". As mentioned above, please use single, unambiguous terminology. Does the term 'synchrony' here refer to synchrony between limbs on the same girdle ,or between ipsilateral limbs ?

Line 201: Please use "correlation" rather than "association"

Figure 6: Visual inspection of Figure e-h shows that the speed values collected in the DOX-treated animals are higher than in the control condition (apparently almost 30cm/s more). I could not find in the manuscript if you mention that the speed was enhanced during DOX treatment. Could you provide such an indication and comment on it.

Minor comments

In the authors' affiliation the address " 5Miami project to cure paralysis... " does not refer to any author in the authors' list.

Line 15-16: " ...the spinal coordinates the muscles distributed... " Can you rephrase as " ... the spinal cord coordinates the activity of muscles distributed... "

Line 45: Please suppress " likely "

Line 74: Rather than "Traditionally, the primary pattern used by rodents " , could you rephrase as "The primary pattern commonly used by rodents "

Line 90: " repeated patter " What does this mean exactly ?

Line 120: " ...suggesting that L2-L5 interneurons do not participate in hindlimb... "; I would rather

rephrase as " ...suggesting that the L2-L5 interneurons that were silenced do not participate in hindlimb... "

Line 165: I could not find Fig. 5h

Line 267: please insert the 'Discussion' heading

Figure 2a-b: there is no y scale for movement amplitude nor an x scale. Also, what was the sampling frequency for the control time points?

Figure 5a-d: as in Fig. 4a-c, it is not clear whether the data points presented here were acquired by the authors or are only theoretical. Please indicate if the latter is the case.

Figure 7: I could not find any legend for the yellow bars in o and p.

Reviewer #3 (Remarks to the Author):

Pocratsky et al. explore the contributions to locomotor behavior of a subpopulation of neurons in the lumbar spinal cord that send descending projections from L2 to L5. Using modern combinatorial viral approaches developed for use in the primate, they find that rat hindlimb alternation can be selectively disrupted by silencing L2-L5 neurons, while other aspects of locomotion, including intralimb coordination and rhythm, remain unimpaired. They conclude that the hierarchical organization of locomotion can be dissected with these types of selective manipulations, revealing a flexible modular organization to motor circuits that collectively produce a more complex behavioral output.

The authors present a well-designed analysis of behavioral perturbations, focusing on systems level quantification and characterization, as described in the introduction. The major novel contribution of the work is the demonstration that left-right alternation can be manipulated independently from other discrete elements of locomotor behavior, and that this disruption is mediated by loss of L2-L5 signaling. The use of a combinatorial viral approach for silencing L2-L5 spinal projection neurons provides a reasonable level of specificity relative to traditional lesion and pharmacological approaches, and the reversible nature of these perturbations provides convincing internal controls. Moreover, the kinematic characterization is thorough and well-designed, statistical analysis is appropriate, and methodological descriptions are detailed enough that others could reproduce the analysis.

While behavioral characterization is the clear strength of this manuscript, there are some weaknesses in characterizing the circuits affected. By performing these studies in rats instead of mice, the specificity of the manipulations is limited. Ideally these studies could be augmented by future work using mouse genetic tools that restrict silencing to specific classes of spinal cord interneurons. For the time being, a more detailed characterization of the circuits being manipulated would strengthen this manuscript. Overall, the study provides a convincing behavioral dissection of locomotor output, and by providing more analysis on the particular circuits affected, this manuscript would be suitable for publication and provide progress and some novel insight for the field. A few suggestions for improvement are listed below:

- 1) The authors designed their viral perturbations to affect both ipsilateral and contralateral projections. However, the authors miss an opportunity to explore whether both ipsi and contralateral projections contribute to the observed phenotype. Unilateral viral injections in both L5 and L2 could help resolve whether the targeted limb is affected by silencing only the ipsi or only the contralateral L2-L5 projection. While the primary limitation of performing these studies in the rat is circuit specificity, these unilateral experiments could help further refine the circuit perturbations to subsets of L2-L5 neurons.

- 2) The analysis of synaptic connections in Fig 7a-c is not very informative. It is unclear where the

NeuN+ neuron is located and what type of neuron is being examined. The authors could refine the anatomical characterization by exploring contacts selectively onto motor neurons (through retrograde motor neuron labeling from muscle, or immunohistochemical label). Similarly, target interneurons could be labeled immunohistochemically to indicate their excitatory or inhibitory identity, and combined with laminar location, could provide more information on which neurons are targeted by L2-L5 projections. As is, there is minimal information about target L5 neuron identity, providing little novel insight into circuit architecture.

3) Similarly, the analysis of L2 neuron location and identity in Fig 7d-h is not very informative. In addition to quantifying the absolute number of ipsi and contralateral L2 neurons labeled, the authors could provide information about laminar distribution and excitatory or inhibitory identity (through immunohistochemistry).

4) The characterization of local projections in Fig7i-r is not very convincing, given that collaterals would need to project rostrally to L1 to be labeled by CTB 488. What about local collaterals in L2 and in segments between L2 and L5? The authors could label L2-L5 neurons using their combinatorial viral labeling approach with a reporter that fills the neuron and permits morphological characterization, and look for local synaptic boutons in L2, L3 etc. using immunohistochemical labeling for synaptic markers.

5) The finding that limb alternation is not affected during swimming behavior is particularly interesting. This would suggest that sensory afferent feedback provides a prominent input to L2-L5 projection neurons during locomotion. It would be informative to know whether the L2 neurons that project to L5 receive significant proprioceptive input. This could be analyzed by labeling sensory afferents via intramuscular CTB injection and quantifying afferent synaptic contacts on to labeled L2-L5 neurons.

6) The authors remark on the surprising finding that intralimb coordination was not affected, given suggestive evidence from prior electrophysiological studies that L2-L5 neurons might be involved in flexor-extensor coordination across joints. The kinematic data presented in Fig 2 and Suppl. Fig 1 supports their conclusions, but they note that vertical movements in the hip are affected. While IHA and HAT joint angular excursions appear normal, how do the authors interpret abnormal kinematics of hip movement? Could this reflect an aspect of altered intralimb coordination?

7) The authors do not provide any histological analysis to confirm that their viral injection approaches are segment specific. Are double-infected neurons restricted to L2, or did viral spread produce labeling (and thus silencing) of neurons in other lumbar segments as well?

8) Minor comment- line 165 refers to Fig 5h – this appears to be a typo and should read Fig. 5f.

Reviewer #1 (Remarks to the Author):

The study by Pocratsky et al deals with the neurons that project from the L2 to the L5 segments in the rat and their effects on the locomotor pattern. When these intersegmental neurons are inactivated through doxycycline (the interneurons are affected through double-virus injections in L5 and L2, respectively), it leads to a change in the locomotor pattern. In the wild type the hind limbs show strict alternation as in walk or trot, and at higher speed they can break into a gallop or a bound. Animals with inactivation of L5-projecting L2 interneurons change the pattern, so that the two hind limbs are less well coordinated, with a less strict alternation even at low speeds. The pattern of activity in each of the hind limbs is on the other hand reported as unchanged. This was an unexpected outcome. The report provides a detailed analyses and the text is clear and the Figures mostly to the point.

General comments:

A very large part of the current literature with genetic approaches to interneuron analyses shows the effect of deletions of the different subgroups of developmentally defined groups (V0 to V3 and different subdivisions). Although the authors do not use this approach, it is surprising that they do virtually not cite this large body of current literature. V0v and V0d interneurons are both known to be critical for alternating locomotor behaviour related to walk and trot, respectively. This would seem most relevant to discuss, but they do not even mention these important findings of Bellardita and Kiehn 2015, although they cite their study for baseline data of interlimb coordination. It would seem likely that the L2-L5 intersegmental neurons could well influence the excitability of these two types of V0 interneurons responsible for coordinating alternation in both walk and trot.

We have included a paragraph in the discussion that is focused on the genetically-encoded interneurons which have been implicated in left-right alternation. We pay special attention to the V0 interneurons, per the reviewer's comment. Please refer to page 19-20, lines 382-394.

The title of the manuscript "Silencing spinal interneurons: a continuum of walking to hopping" does in the reviewer's mind is overstating the findings.

While we agree that our observations do not support an evenly represented continuum of walking to hopping, we definitely observe and report the entire continuum as part of the phenotype observed during silencing, therefore we respectfully request that the title remains even if it is more casually descriptive than scientific.

Specific comments:

Lines 57 – 68 and Fig 2. The authors measure "joint angles" between iliac rest – hip – ankle or hip – ankle toe omitting the knee – provides some complication for the evaluation. The movement recordings in Fig 2 are not sufficiently detailed to allow the conclusion that there are no changes in the intralimb-coordination, although the overall movement pattern appears largely unchanged. Would require a more detailed analysis – preferably EMG.

We acknowledge the complication with Figure 2. To address this concern, we have taken the following steps.

First of all, we have clarified how we measured limb kinematics by using a three-segment, two-angle model that ignores the knee.

Secondly, we generated schematics of five key instances throughout the step cycle (Figure 2a,c). We traced the position of the hindlimbs during these defined events to provide a visual representation of intralimb movements throughout the step cycle. We believe this provides clarification regarding movements of the three hindlimb segments and two joint angles during control and DOX^{ON} time points, respectively. Concomitant with these illustrations are the excursions of the proximal and distal angles (listed below the schematics). The peak-to-trough excursions of the proximal and distal joint angles during stepping were not significantly affected by conditional silencing (Supplementary Figure 1).

Third, we added x and y-axes to the stick figures to clarify the spatial movements of the limbs over time (Figure 2b,d). The schematics illustrated in Figure 2a,c were generated from the locomotor bouts illustrated in Figure 2b,d.

Fourth, we provided additional detail regarding the temporal coordination between the proximal and distal angles. In Figure 2e-h we show four representative examples of the excursion of the proximal and distal hindlimb angles. In these traces, we demarcate the timeframe of peak-to-peak excursion of the distal angle and where the proximal angle had peak excursion. The resultant coordination value for each example is then shown in the circular plots (blue circles).

Finally, we provided additional clarification regarding the overall excursion of the proximal and distal angles. In Supplementary Figure 1 (panels a,d), we show representative traces of the peak-to-trough excursion of the proximal and distal angles independently (as compared to the Figure 2e-h where they are combined). Here, we overlaid the DOX^{ON} traces onto that of Control, demonstrating that there is a slight, but insignificant decrease in the peak excursion of the proximal hindlimb angle during silencing. This principle is illustrated in Figure 2a,c wherein the peak angular excursion, which typically occurs when the limb is maximally extended at the end of stance phase (illustrated in Figure 2a,c, panels i,v), is slightly reduced during DOX^{ON}. Similarly, when we overlay the Control and DOX^{ON} traces of the distal angle excursion (Supplementary Figure 1d), it is clear that silencing does not alter the overall range-of-motion (peak-to-trough) during stepping.

We agree with the reviewer that EMG recordings during stepping would provide a more detailed assessment of how silencing L2-L5 interneurons affects muscle activation during stepping. However, based on a growing literature demonstrating good agreement between EMG and kinematics/gait (Gillis & Biewener 2001; Leblond et al 2003; Thota et al 2005) we believe the findings presented are well supported even in the absence of supporting EMG data.

125-183 and Fig 4-5. It is clear that after inactivation of the L2-L5 interneurons one limb in relation to the other can drift to resume the preferred alternating pattern. It would be interesting to know if the drift is gradual as in “relative coordination”, when there is a

gradual change over some cycles to later return to the preferred 0.5 coupling. Fig 5f contains data relevant for such an analysis– but the resolution is not sufficient – would be interesting to extend the DOX recordings so that it could be analysed/plotted in an appropriate way. When wild-type animals (intact or spinal) walk on a split belt – large differences in belt speeds can result in 2:1 couplings between the two limbs in which they maintain a preferred phase coupling. This shows that the interneuronal system can adapt beyond the fixed couplings in walk – trot and bound and in gallop the coupling is in fact more flexible.

We are not completely certain as to what the reviewer was explicitly referring to when they stated “...would be interesting to extend the DOX recordings so that it could be analyzed/plotted in an appropriate way.” If by “extension” the reviewer referred to having a longer stepping tank to increase the number of contiguous steps taken, we have commented on this issue in our results section (pages 10, lines 170-172). Indeed, if there were more contiguous steps taken, we could better address the relative drift in coordination.

If by “extension” the reviewer referred to extending the data plotted in Figure 5f thereby increasing the resolution, then we have addressed this issue in Supplementary Figure 2. Here, we plot a sampling of Control and DOX^{ON} data (n=95 and 96 per step changes in left-right coordination, respectively) wherein the graphs are extended vertically and horizontally. Moreover, we use this sampling of data to create Poincaré plots as a means to further address the phase drift question. These plots (also called return or phase delay maps) are a way to illustrate the phase relationship of one hindlimb step (step n+1, y-axis) as a function of the previous step (step n, x-axis) (Golinska 2013). The raw left-right phase data of the hindlimbs is plotted, not the absolute difference in coordination with each step cycle. Each circle denotes two successive step cycles, where the left-right hindlimb phase relationship of the first step is plotted on the x-axis (step n) and the phase relationship of the subsequent step taken is plotted on the y-axis (step n+1). The shape of the Poincaré plots allows visualization of the variability in the hindlimb phase data, which ultimately describes the evolution of interlimb coordination with each step cycle. In these plots, it is easier to identify instances where the hindlimbs gradually drift towards alternation (Supplementary Figure 2h, circles that “trail” towards the middle of the plot) or if there were dramatic shifts in left-right coordination with successive steps taken (blue circles).

308-316 The fact that the swimming phenotype after DOX remains strictly alternating, is interesting, but can possibly be explained by many factors – the limb movements in water becomes automatically different from that of walking with ground contact, varying load, and inevitable minor perturbations. Whether swimming actually represents a different motor program as argued, rather than an adaptation of the walking motor program to modified physical constraints appears unclear (line 316).

We agree that there is little evidence to suggest that stepping and swimming are the result of “different locomotor programs” as we stated in the discussion. As such, we have changed our language from “...while stepping and swimming patterns likely arise from similar or overlapping neural pathways, they represent different locomotor programs” to the following: “Therefore, while both behaviors can take the form of rhythmic left-right hindlimb alternation, it is clear

there are stark differences underlying the expression of these alternating movements as silencing L2-L5 interneurons significantly alters one form, but not the other.” (Page 19, lines 375-377)

286-291 The authors ask themselves where they should place the L2-5 interneurons in the “locomotor hierarchy” and end up in suggesting that they should be part of the “left-right pattern information layer”. The authors have made a distinct observation regarding the effect of a deletion of the L2-L5 interneurons, but the reviewer feels it is meaningless to discuss in those soft terms, when in reality the connectivity and relation to V0 neurons should be one main primary aim to consider.

We have updated our manuscript to discuss a plausible relationship between L2-L5 interneurons and V0 interneurons. Please refer to the discussion section, pages 19-20, and lines 382-394.

In conclusion the authors have shown that the deletion of L2-L5 interneurons leads to less secure alternating pattern, but still alternation is preferred. The connectivity of these interneurons is largely unknown both on the incoming and efferent side.

We have performed additional immunohistochemical analyses to further explore plausible supraspinal and sensory inputs onto L2-L5 interneurons. Please refer to our new figure (Figure 8) where we shown these neurons likely receive proprioceptive (vGLUT1) as well as serotonergic (5HT) inputs, immunoreactivity that is likely from supraspinal centers (as described in detail by Ghosh & Pearse, 2014). Further detail regarding these analyses can be read in our response to reviewer #3.

Reviewer #2 (Remarks to the Author):

Referee's report on manuscript entitled Silencing spinal interneurons: a continuum of walking to hopping by Pocratsky, Burke, Morehouse, Beare, Riegler, Tsoulfas, Whittemore, Magnuson.

This study investigates the role of spinal interneurons originating in lumbar segment 2 and terminating in the fifth lumbar segment, which the authors label as L2-L5 interneurons. Many attempts have been made to decipher the role of spinal interneuron subtypes using a variety of genetic models of neonatal mice. However, this investigation constitutes the first successful attempt using genetic tools (viral transfection and reversible functional knock-down) and in an adult rat model to experimentally manipulate the activity of interneurons involved in locomotor behavior. As such this paper opens new interesting avenues in which role of specific subsets of interneurons in the freely behaving adult animal can be examined. Nonetheless, I have several issues that should be addressed by the authors.

Two main issues warrant discussion. First, the degree of transfection provided by the protocol used. The authors report (line 302) that 30% of hindlimb steps were altered by silencing the L2-L5 interneurons. Could this relatively low proportion be related to the amount of transfection and do they have any data available on this important point? [Authors' response #1]. To effectively address this question, the authors could compare the distribution of the number of transfected neurons with the number of FluoroRuby and

FluoroEmerald-retrogradely labeled neurons in order to establish a ratio that may (or may not) correspond to the effect on actual stepping behavior [Authors' response #2]. In another approach to testing the variability related to transfection level, the authors could correlate the ratio of altered steps for each animal to subsequent anatomical data from immunohistochemical detection of EGFP.eTeNT-positive neurons [refer to Authors' response #1]. In any case, even if these approaches are not technically feasible, it has to be discussed more than in a single line (see line 303).

[Author's response #1]:

We do not know whether the modest effects observed are due to low numbers of double-infected L2-L5 interneurons. In our initial submission, we did not include data related to the double-infection of L2-L5 interneurons at the level of the cell bodies. This is due to the technical difficulties we faced in amplifying the detectable signal of this conditionally-expressed protein. However, we recently developed an immunohistochemical protocol that allowed the detection of these infected neurons. Please refer to Figure 7 and Supplementary Figure 4 where we show images of double-infected L2-L5 interneurons. The details of this protocol are described in our methods section (pages 31-33, lines 660-688). Despite these advances in our immunohistochemical techniques, we still cannot reliably perform absolute cell counts to determine the number of double-infected neurons. Rigorous cell counting requires two key features: (1) consistency in expression levels and (2) a high signal-to-noise ratio to detect positive signal above background “noise.” Our histological data do not meet these criteria for the following reasons.

First and foremost, the lentiviral package (eTeNT.EGFP transgene) is randomly inserted into the host cell genome. Therefore, if the transgene is inserted into a highly transcribed area, then *post hoc* amplification of the conditionally expressed protein should yield dark immunoreactive labeling. However, if the transgene is randomly inserted into a poorly transcribed area, then *post hoc* amplification of eTeNT.EGFP would yield weakly labeled neurons. Therefore, in these cross-sections we observe a spectrum of immunoreactivity (dark to weak, negative for non-infected neurons). As such, setting a “baseline threshold” for inclusion criteria proves to be highly subjective and challenging.

Second, we were unable to sufficiently elevate the signal of the conditionally-expressed eTeNT.EGFP fusion protein beyond background noise. This limitation is problematic as it relates to double-infected neurons with weak-to-modest expression of eTeNT.EGFP as described above. Additionally, the proximity of the double-infected neurons to the AAV2 injections also proved to be a confounding influence for signal:noise (refer to Figure 7g, image taken was proximal to injection site). There were four AAV2 injection sites performed at L2 in this study. Therefore, it is clear there would be serious issues in performing absolute cell counts of every double-infected neuron throughout the entire L2 segment. These issues are both technical in nature as well as in our interpretation of any numbers obtained (per random transgene insertions cell-to-cell, signal:noise issues, proximity to injection sites, etc).

In light of the limitations described above, we believe that generating the absolute cell count data of double-infected L2-L5 interneurons is not technically feasible. Moreover, we believe that if we were to attempt absolute cell counts, the numbers generated would likely reflect an under-sampling of the total population infected (primarily due to the two issues described above). This under-sampling would make it difficult for us to relate the histological data to the modest changes observed in hindlimb coordination.

The reviewer requested that we discuss in greater detail whether or not these approaches are technically feasible. We have expanded our discussion to address these concerns (please refer to pages 18-19, lines 357-367).

[Authors' response #2]: While we value the reviewer's suggestion to compare the distribution of infected neurons to that of the tracer-labeled population (dextrans or CTB conjugates), there is a key limitation to this approach. The uptake mechanisms and diffusion properties between fluorescent tracers and viral vectors are profoundly different. The lentivirus is a pseudotyped HIV-1 vector designed for enhanced uptake at the terminal field (via its fusion of rabies virus glycoprotein [extracellular and transmembrane domains] with cytoplasmic domain of vesicular stomatitis virus) (Kinoshita et al 2012; Kato et al 2011 [*Hum Gene Ther* 22, 1511-1523]; Kato et al 2011 [*J Neurosci*]; Kato et al 2011 [*Hum Gen Ther* 22, 197-206]). Alternatively, the fluorescently-conjugated cholera toxin beta subunit tracer is taken up by binding to the monosialotetrahexosylganglioside receptor (GM1) (Lencer & Tsai 2003). Stark differences also apply for labeling at the level of the cell body. For example, the AAV2 virus is neurotropic and infects the cell bodies through cell surface glycan binding (heparin sulfate proteoglycan for serotype 2) (Murlidharan et al 2014). Alternatively, an anterograde tracer such as biotinylated dextran amine (BDA) labels cell bodies through an unknown mechanism, although it is likely endocytotic in nature (Reiner et al 2000). Therefore, due to the disparate nature between viral infections and tracer uptake, we believe that using tracer data as a "stand in" or approximation for double-infected cell bodies is non-ideal.

A second important issue concerns whether the strength of the observed effect is somehow related to the localization/distribution of the locomotor-generating circuit kernel in the spinal cord. That the core of the locomotor network is located in the rostral lumbar cord is no longer an issue and this is now established and very well documented (including by the present authors in several earlier publications: e.g., Magnuson and Trinder, *J. Neurophysiol*, 77: 200-206, 1997; Hadi et al., *J Neurosurg.*, 93: 266-275, 2000). However, this rhythmogenic kernel is not only restricted to the L2 segment but also overlaps extensively with the L1 segment (see for example Antri et al., *Plos One*, 2011). Therefore, could it be possible that if transfection was not only restricted to L2-L5 interneurons but also included interneurons in L1, would the behavioral effects have been greater. This point, which relates to extent to which the essential components of the locomotor CPG network are targeted by the protocol employed, should be addressed.

We agree that a greater effect may have resulted if we had silenced L1 and L2, however we believe it is unlikely to have been a fundamentally different effect. Nonetheless, to address this concern indirectly, we screened for double-infected neurons throughout the rostral lumbar

segments (L1-rostral L3) using our recently developed immunohistochemical protocol (detailed in our expanded methods section, pages 31-33, lines 660-688). Briefly, we found numerous double-infected neurons at the L2 segment that were distributed throughout the intermediate gray matter (Figure 7, Supplementary Figure 4). The apparent size, shape, and distribution pattern of these double-infected neurons was similar to what we observed when we retrogradely labeled L2-L5 interneurons with a fluorescent tracer (Figure 8). When we screened for double-infected neurons at the L1 and L3 segments, we saw very few immunoreactive neurons (Supplementary Figure 4), thus confirming that silencing is largely limited to L2. We have updated our methods (pages 31-33, lines 660-688) and results (page 14, lines 255-256) to reflect these new analyses.

Line 74: and throughout the manuscript: The authors have to clarify the terminology used here and subsequently. Classical gait studies refer to 'symmetric' or 'asymmetric gaits and some authors such as Grillner (1981) refer to the alternating pattern when two limbs on the same girdle as phase-opposed. Here, it rather seems that this terminology refers to the alternation between hind-and forelimbs. Could you be more explicit and define to what precisely your terminology refers to?

The data shown in Figure 3 reflects the analysis of the order in which the limbs step during overground locomotion (step sequence pattern). These data do not reflect the temporal, interlimb relationship during stepping. Therefore, when we state that the alternate pattern (as defined by Cheng et al 1997) is “...characterized by alternation of the hind- and forelimbs with each step,” we refer to a limb recruitment pattern of (limb contact #1) forelimb, (2) hindlimb, (3) forelimb, and (4) hindlimb stepping. In this context, the limbs step in an “alternating” fashion such that each subsequent step that the animal takes occurs between the pelvic and shoulder girdles, respectively.

We fully acknowledge the confusion this statement has raised and appreciate the reviewer’s request to clarify our language. We have changed the text to the following: “...characterized by the following footfall order (e.g. alternate step sequence pattern, subtype a): (step 1) right forelimb, (2) left hindlimb, (3) left forelimb, and (4) right hindlimb...” (Refer to page 6, lines 72-75).

Line 75: is there a reference for the "zig-zag" pattern? Why do you not use classical phase bar diagrams with the bar indicating the time of contact for each limb. It is visually much more relevant and allows indicating SD as well see overlap between individual limbs.

The “zig-zag” pattern was used to describe the visual appearance of the alternate step sequence pattern when graphed. There is no reference for this term and it has been removed from the manuscript.

Step sequence pattern analysis measures the order in which the limbs step during locomotion. It does not directly assess the swing-stance durations of the stride cycle. This was our rationale for representing the data as a series of circles connected by lines (Figure 3a-c, right panels) to denote the limb recruitment patterns.

We agree with the reviewer that the classical phase line graphs are pertinent. We have included duty cycles (swing-stance graphs) from multiple animals across different time points (please refer to Supplementary Figure 2).

Line 78: again a problem with terminology. I do not believe that 'cruciate pattern' is a standard term. Please rephrase or define.

We acknowledge the reviewer's concern that the term "cruciate pattern" is likely non-traditional. This term has been defined in the manuscript as a footfall order of "forelimb-forelimb-hindlimb-hindlimb" (page 6, lines 78-79).

Line 81: In this Figure, which illustrates the effect of silencing the L2-L5 interneurons as well as in the following Figures, it seems that the effect of DOX2 is somewhat weaker than the first DOX1 test. Was this systematically observed, and what could this be due to?

Although it appears as though the effects of silencing L2-L5 interneurons during DOX2 were weaker as compared to DOX1, we did not observe a significant difference in the overall proportion of steps that were disrupted (22.22% versus 21.43%, $p=0.849$ with Binomial Proportion test). We observed that peak hindlimb disruption occurred during DOX1 in $N=4$ animals while $N=2$ animals had peak effects during DOX2. We have updated our results as well as methods to reflect these analyses (please refer to page 7-8, lines 112-114; page 27, lines 552-559).

Line 93-103: The presentation of the Fig. 4 a-c is somewhat ambiguous. It is not clear whether these derive from actual data collected in the study or represent only a theoretical illustration. This should be clearly specified. Also, it should be mentioned whether if the tested animals were occasionally spontaneously performing galloping or bounding gaits.

The schematics, swing-stance graphs, and polar plots shown reflect data that was collected in a separate study. We have updated the figure legend (page 55, lines 1208-1209) and methods (page 25, lines 519-522) to reflect this issue.

No spontaneously-expressed gallop or bound-like gaits were observed across all control time points. We have updated our methods to address this concern (page 22, line 454).

Line 131-150: The differences between the concept of "dispersion and concentration" is not obvious, and I am not sure that there is an added value to detail each item. In my opinion these two variables are redundant. Either you can strongly justify that they are complementary or please just detail one of these two variables, otherwise it is quite confusing because one expects something different from comparing "dispersion" to "concentration", and if not, why.

The authors agree with the reviewer concerns regarding the redundancy of the phasic dispersion and concentration analyses. The dispersion dataset has been removed from the manuscript.

Line 151-174: Although single step data are of value, why did the authors not calculate the coefficient of variation, which is classically used to underline the point-by-point variability, therefore providing a meaningful index of variability?

We chose to analyze the per-step changes in left-right coordination as this measure endows greater resolution in how silencing L2-L5 interneurons affected hindlimb stepping on a dynamic, stride-by-stride basis.

We value the reviewer's request to investigate the coefficient of variation and have since processed our functional data to report on this variability index. We found that the coefficient of variation of left-right hindlimb coordination was significantly increased during conditional silencing of L2-L5 interneurons as compared to control time points ($28.33 \pm 9.02\%$ vs $10.65 \pm 2.36\%$, $p < 0.005$ with paired t-test; $t = 2.57$; $df = 5$). Please refer to page 8, lines 128-134 for the summary of these data. We have also updated our methods to reflect how these data were calculated and analyzed (page 27, lines 560-565).

Line 198: "...that the more synchronous gaits...". As mentioned above, please use single, unambiguous terminology. Does the term 'synchrony' here refer to synchrony between limbs on the same girdle or between ipsilateral limbs?

We have clarified our language in the results to address this concern. Here, we state the following: "...that the more synchronous gaits typically occur wherein the left-right limb pairs at each girdle move in-phase (e.g. full bound) or at a slight phase-shift (e.g. gallop) relative to one another." (Page 11, lines 199-202)

We appreciate the reviewer's request to use single, unambiguous terminology. We have included a paragraph in our methods to address this issue (pages 25-26, lines 523-528).

Line 201: Please use "correlation" rather than "association"

We have used "correlation" instead of "association" per the reviewer's request (page 12, line 205-207).

Figure 6: Visual inspection of Figure e-h shows that the speed values collected in the DOX-treated animals are higher than in the control condition (apparently almost 30cm/s more). I could not find in the manuscript if you mention that the speed was enhanced during DOX treatment. Could you provide such an indication and comment on it?

We have created an additional table (shown in Supplementary Figure 3a) to reflect the overall speed across all time points, respectively (data shown as group average). Here, we show that overall locomotor speed was significantly enhanced during DOX1^{ON}-D5 and DOX1^{ON}-D8 when compared to Pre-DOX1 (each $p < 0.5$, repeated measures analysis of variance followed by Tukey's *post hoc* honest significant difference t-test). Comparing these DOX^{ON} time points to Baseline and DOX^{OFF} yielded no significant differences. Similarly, we observed a significant enhancement in overall locomotor speed when we compared Pre-DOX2 to DOX2^{ON}-D3 and DOX2^{ON}-D5, respectively (each $p < 0.01$). We also denote the overall speed range (minimum-

maximum) for the group during Control and DOX^{ON} time points, respectively. The group change (from Control to DOX^{ON}) in overall minimum speed expressed was 12.07±7.51 cm/s. The group change in overall maximum speed achievable was 19.20±5.31 cm/s. Our results (page 11, lines 194-197) and methods (page 28, lines 581-586) have been updated to reflect these new analyses.

However, in our examination of a more discrete measure of speed (instantaneous speed, or the speed at which an individual limb moves with each step taken), we observed no meaningful correlation between the silencing-induced changes in left-right hindlimb coordination and the speed at which these events occurred (Figure 6b, Spearman Rank correlation coefficient of 0.13, accounting for 1.69% of the variance). We speculate that this generalized enhancement of overall speed achieved (but not at the level of individual steps taken and their phase-speed relationship) is likely, in part, due to the faster-paced steps taken at speeds >90 cm/s whose coordination values ranged from 0.5 to 0.7 (Figure 6b, circles above the gray dashed line).

Minor comments:

In the authors' affiliation the address "⁵Miami project to cure paralysis..." does not refer to any author in the authors' list.

We thank the reviewer for identifying this mistake. It is now corrected.

Line 15-16: "...the spinal coordinates the muscles distributed..." Can you rephrase as "...the spinal cord coordinates the activity of muscles distributed..."

We have updated the text accordingly. Please refer to page 3, lines 15-16.

Line 45: Please suppress "likely"

The word "likely" has been removed from the results heading. Page 5, lines 46-47.

Line 74: Rather than "Traditionally, the primary pattern used by rodents", could you rephrase as "The primary pattern commonly used by rodents"

We have updated the text accordingly (page 6, lines 72-73).

Line 90: "repeated pattern" What does this mean exactly?

There are numerous coupling schema between different limb pairs (e.g., left-right limb pairs at each girdle, homolateral versus diagonal hindlimb-forelimb). The temporal coordination between select limb pairs is relatively well-defined for the classic gaits (Bellardita & Kiehn 2015). For example, in the walk or trot gaits the left and right hindlimbs move out-of-phase relative to one another (interlimb coordination pattern: alternation) while the diagonal hindlimb-forelimb pairs move in-phase. As such, we used "patterns" to collectively describe the interlimb coupling schema observed during a particular gait.

“Repeated” refers to the stability at which the various coordination relationships are expressed in select gaits. For example, animals will locomote in a walk or trot gait for multiple step cycles. As such, the coupling patterns described above are expressed in a “repeated” manner.

We recognize the relative ambiguity of how we described this feature. As such, we have changed our language from “distinct, repeated patterns” to “interlimb coupling patterns.” (Page 6, lines 88-89)

Line 120: “...suggesting that L2-L5 interneurons do not participate in hindlimb... ”; I would rather rephrase as “ ...suggesting that the L2-L5 interneurons that were silenced do not participate in hindlimb... ”

We have updated the text accordingly. (Page 8, lines 120-123)

Line 165: I could not find Fig. 5h

Line 165 was a typo. We thank the reviewer for identifying this mistake. It is now corrected to reflect Fig. 5f, not h.

Line 267: please insert the 'Discussion' heading

We have updated the text to include a “Discussion” heading.

Figure 2a-b: there is no y scale for movement amplitude nor an x scale. Also, what was the sampling frequency for the control time points?

Figure 2 has been updated to reflect the vertical movements of the limb segments with each stride (y-axis in centimeters) as the animal actively stepping over time (x-axis in milliseconds). We have also updated the figure legend to reflect the sampling frequency, which was 100 frames/second (please refer to page 53, lines 1154-1155 in the manuscript).

Figure 7: I could not find any legend for the yellow bars in o and p.

The data originally shown in Figure 7 is now in Figure 8. Here, all the bars that show quantitative data are gray. The ipsilateral versus contralateral L2-L5 interneurons are shown with different intensities of gray, a distinction that is now noted in the figure legend (page 59, line 1286-1287).

Reviewer #3 (Remarks to the Author):

Pocratsky et al. explore the contributions to locomotor behavior of a subpopulation of neurons in the lumbar spinal cord that send descending projections from L2 to L5. Using modern combinatorial viral approaches developed for use in the primate, they find that rat hindlimb alternation can be selectively disrupted by silencing L2-L5 neurons, while other aspects of locomotion, including intralimb coordination and rhythm, remain unimpaired. They conclude that the hierarchical organization of locomotion can be dissected with these

types of selective manipulations, revealing a flexible modular organization to motor circuits that collectively produce a more complex behavioral output.

The authors present a well-designed analysis of behavioral perturbations, focusing on systems level quantification and characterization, as described in the introduction. The major novel contribution of the work is the demonstration that left-right alternation can be manipulated independently from other discrete elements of locomotor behavior, and that this disruption is mediated by loss of L2-L5 signaling. The use of a combinatorial viral approach for silencing L2-L5 spinal projection neurons provides a reasonable level of specificity relative to traditional lesion and pharmacological approaches, and the reversible nature of these perturbations provides convincing internal controls. Moreover, the kinematic characterization is thorough and well-designed, statistical analysis is appropriate, and methodological descriptions are detailed enough that others could reproduce the analysis.

While behavioral characterization is the clear strength of this manuscript, there are some weaknesses in characterizing the circuits affected. By performing these studies in rats instead of mice, the specificity of the manipulations is limited. Ideally these studies could be augmented by future work using mouse genetic tools that restrict silencing to specific classes of spinal cord interneurons. For the time being, a more detailed characterization of the circuits being manipulated would strengthen this manuscript. Overall, the study provides a convincing behavioral dissection of locomotor output, and by providing more analysis on the particular circuits affected, this manuscript would be suitable for publication and provide progress and some novel insight for the field. A few suggestions for improvement are listed below:

1) The authors designed their viral perturbations to affect both ipsilateral and contralateral projections. However, the authors miss an opportunity to explore whether both ipsi and contralateral projections contribute to the observed phenotype. Unilateral viral injections in both L5 and L2 could help resolve whether the targeted limb is affected by silencing only the ipsi or only the contralateral L2-L5 projection. While the primary limitation of performing these studies in the rat is circuit specificity, these unilateral experiments could help further refine the circuit perturbations to subsets of L2-L5 neurons.

We fully acknowledge that the approach we used here precludes us from ascribing unique functional role(s) to the ipsi- and commissural subpopulations, a limitation we have stated in our discussion (page 18, lines 347-349). Indeed, at first blush addressing these questions appears relatively straightforward with unilateral injections to silence the ipsilateral pathway and contralateral injections to silence the commissurals. However, it is critical to note that only one set of ipsilateral (e.g. left L2 interneurons that project to L5) or commissural (e.g. left L2 interneurons that project to right L5) subtypes can be studied, respectively. Therefore, in these experiments we would be investigating a subset (e.g. left L2-L5 interneurons) of a subset (e.g. ipsilateral L2-L5 interneurons, left and right sides). Because we would be silencing a considerably small fraction of L2-L5 interneurons with respect to the otherwise intact L2-L5 network (essentially one-fourth of the network as there are two ipsi- and two commissural

subtypes), let alone the entire locomotor circuitry, we believe the potential for functional compensation (thereby masking the behavioral deficits) would be high.

In light of this credible issue, the profound effects it would have on our interpretations, and the length of time required to complete these studies, we respectfully contend that addressing the L2-L5 interneuron subtype questions raised by the reviewer are beyond the purview of this study.

2) The analysis of synaptic connections in Fig 7a-c is not very informative. It is unclear where the NeuN+ neuron is located and what type of neuron is being examined. The authors could refine the anatomical characterization by exploring contacts selectively onto motor neurons (through retrograde motor neuron labeling from muscle, or immunohistochemical label). Similarly, target interneurons could be labeled immunohistochemically to indicate their excitatory or inhibitory identity, and combined with laminar location, could provide more information on which neurons are targeted by L2-L5 projections. As is, there is minimal information about target L5 neuron identity, providing little novel insight into circuit architecture.

We have completed the following histological analyses to address these concerns.

First, we performed immunohistochemical analyses to identify eTeNT.EGFP-expressing putative synapses onto motor neurons in the caudal lumbar spinal cord. Motor neurons were identified through the use of the anti-vesicular acetylcholine transporter antibody (VACHT), an established marker of cholinergic neurons (Gilmore et al 1996; Ferguson et al 2003). The images shown reflect three-dimensional volumetric rendering of the entire z-stack (Figure 7d-f) as well as orthogonal cross-sections of select optical slices to effectively illustrate xz-yz co-localization of eTeNT with VACHT (Figure 7f_{ii-v}). (Results: page 13, lines 245-250; methods: page 30, lines 627-632; page 31, lines 645-655)

Second, we now show that some of the silenced L2-L5 interneuron putative synapses are excitatory in nature (Supplementary Figure 4). To do this, we co-stained for the vesicular glutamate transporter 2 (vGLUT2), a well-known excitatory neurotransmitter marker (Freneau et al 2001; Bai et al 2001). (Results: page 13, lines 240-241; methods: page 30, lines 627-637)

Unfortunately, we could not ascertain if eTeNT co-localized with inhibitory neurotransmitter markers. From our experience, the antibodies that consistently and reliably labeled inhibitory synapses as well as the amplification of eTeNT.EGFP (anti-GFP) are both rabbit host species. Other host species of anti-GFP (chicken, goat, and mouse) yielded poor results. Similarly, our attempts to utilize other known markers of inhibitory neurotransmitters yielded inadequate labeling. We performed antigen retrieval in an attempt make use of the non-rabbit anti-GFP primary antibodies in conjunction with the rabbit anti-VGAT. Once again, these endeavors proved unsuccessful. Nonetheless, based on work by Butt *et al* 2002 (on a plausible anatomical correlate of the pathway studied here), we speculate that synapses derived from L2-L5 interneurons are also likely inhibitory in nature. We have updated our text to address this limitation and our supposition of inhibitory L2-L5 interneurons (page 13, lines 241-244).

Addressing the reviewer's request to ascertain the neurotransmitter phenotype of the neurons targeted by silenced L2-L5 interneurons proves to be challenging for several reasons. This is

described below as the challenges we face here are the same challenges we face in determining the neurotransmitter identity at the level of the L2-L5 interneuron cell body.

3) Similarly, the analysis of L2 neuron location and identity in Fig 7d-h is not very informative. In addition to quantifying the absolute number of ipsi and contralateral L2 neurons labeled, the authors could provide information about laminar distribution and excitatory or inhibitory identity (through immunohistochemistry).

We have taken the following steps to address this request.

First, we sampled from the total number of L2-L5 interneurons that were counted (n=5,884; data shown in Figure 8r) for laminar distribution analyses. We analyzed n=2,737 L2-L5 interneurons for their overall laminar distribution as well as that of the ipsilateral and commissural subpopulations, respectively. The results of these analyses are shown in Figure 8i-k (upper panels).

Second, we generated heatmaps using a custom designed MatLab program to visualize the medio-lateral, dorso-ventral positioning of L2-L5 interneurons in the spinal L2 gray matter (Figure 8i-k, lower panels). These heatmaps highlight the difference between the ipsilateral and commissural L2-L5 interneurons in their laminar distributions. We have updated the results (page 15, lines 287-291), figure legends (page 58, lines 1272-1273), and the methods (pages 36-37, lines 769-787) to reflect these new analyses.

To directly address the reviewer's commentary regarding the L2-L5 interneuron neurotransmitter phenotype at the level of the cell body would be challenging. Our most robust neurotransmitter markers (anti-VGAT for inhibitory, anti-vGLUT2 for excitatory) do not label the cell bodies. Immunohistochemical expression of glutamate has been extensively documented in the cortex (Fujiyama et al, 2001; Kaneko et al, 2002; Kaneko & Fujiyama, 2002; Varoqui et al, 2002; Ziegler et al, 2002). However, detection of glutamatergic neurons in the spinal cord is problematic and inconsistent (Fernandez-Lopez et al 2012; Walberg et al 1990). We found no convincing documentation of such in the literature. Even the development of specific glutamatergic antibodies following *in situ* hybridization (as target validation) have failed to generate primary antibodies that can robustly label glutamatergic expression throughout the somata using standard immunohistochemical techniques (Landry et al 2004; Todd 2010). Numerous glutamate-positive interneurons have been identified within the intermediate gray matter using *in situ* hybridization (Oliveira et al 2003; Landry et al 2004; Fernandez-Lopez et al 2012). However, this protocol is not compatible with fluorescent tracer analyses (Watakabe et al 2010). It is possible that L2-L5 interneurons could express glutamate at the level of their cell bodies as they also reside within laminae VII-VIII. Moreover, we show that terminal-like structures derived from double-infected L2-L5 interneurons co-localize with vesicular glutamate transporter 2, indicating that at least a subset of L2-L5 interneurons are excitatory in nature (Supplementary Figure 4). (Refer to results, page 13, lines 240-241; methods: page 30, lines 627-637; page 31, lines 655-661).

Known inhibitory interneuron markers primarily label interneurons within the superficial laminae of the dorsal horn where L2-L5 interneurons do not reside. Parvalbumin-expression, a

GABAergic marker, labels neurons within laminae II-III (Hughes et al 2012; Molgaard et al 2014). Calretinin, a marker for inhibitory neurons that are GABA-negative within the spinal cord, primarily labels neurons within the superficial laminae of the dorsal horn (Molgaard et al 2014). Calbindin is another marker for GABA-negative inhibitory interneurons, but these neurons primarily reside within lamina II with some immunoreactivity documented in laminae I, II, and IV (Antal et al 1991). Finally, somatostatin immunoreactivity, which is pronounced in the hippocampus, shows little signal in the spinal cord. Of the interneurons that are weakly-labeled, the majority reside within lamina II (Proudlock et al 1993; Heinke et al 2004).

Taken together, a systematic characterization of the L2-L5 interneuron neurotransmitter phenotype expression at the level of the cell body using immunohistochemistry would likely not be feasible. However, based on the published findings of Oliveira *et al* (2003) and Molgaard *et al* (2014) and the electrophysiology studies performed by Kiehn and colleagues (Butt et al 2002; Kiehn & Butt 2003; Quinlan & Kiehn 2007), we speculate that L2-L5 interneurons are a heterogeneous pathway comprised of excitatory and inhibitory neuronal subtypes. We address this in the results section of our revised manuscript (page 13, lines 241-244).

4) The characterization of local projections in Fig7i-r is not very convincing, given that collaterals would need to project rostrally to L1 to be labeled by CTB. What about local collaterals in L2 and in segments between L2 and L5? The authors could label L2-L5 neurons using their combinatorial viral labeling approach with a reporter that fills the neuron and permits morphological characterization, and look for local synaptic boutons in L2, L3 etc. using immunohistochemical labeling for synaptic markers.

To address the reviewer's question regarding the presence or absence L2-L5 interneuron-derived collaterals throughout the lumbar neuraxis, we screened for eTeNT.EGFP-positive terminal-like structures using two different immunohistochemical techniques. First, we used an immunofluorescence protocol to screen for eTeNT.EGFP-positive synapses onto neurons in the intermediate and ventral gray matter throughout the lumbar neuraxis. With this approach, we observed eTeNT.EGFP immunoreactivity throughout the terminal field zone (L4-L5), as shown in Figure 7a-b. We observed sparse eTeNT.EGFP terminal-like signal throughout the intervening segments between caudal L2 and rostral L4 (Supplementary Figure 4). Second, we used a heat-induced epitope retrieval and immunoperoxidase protocol to screen for eTeNT.EGFP-positive signal throughout the lumbar neuraxis. While this protocol revealed dense, eTeNT.EGFP-positive terminal-like structures at L4-L5, we observed little-to-no immunoreactivity at L3 and L6 (Supplementary Figure 4). Together, these data suggest that L2-L5 interneurons densely project to the caudal lumbar segments (L4-L5) while projections to the rostral (L1, Figure 8l-r), intermediate (L3, Supplementary Figure 4), and caudal-most lumbar segments (L6, Supplementary Figure 4) are sparse.

The reviewer suggested that we implement a modified version of the combinatorial Tet^{On} virus system. Unfortunately, our Tet^{On} Materials Transfer Agreement precludes us from modifying the constructs. We do not have permission to replace the eTeNT.EGFP transgene with a fluorescently-tagged synapse labelling construct (SynTag) to screen for L2-L5 synaptic boutons throughout the lumbar neuraxis.

We fully acknowledge that a systematic, rigorous investigation of the anatomical underpinnings of this pathway is required in the future. However, the experiments required to fully address these anatomy-focused questions are beyond the scope and focus of this functional-based paper.

5) The finding that limb alternation is not affected during swimming behavior is particularly interesting. This would suggest that sensory afferent feedback provides a prominent input to L2-L5 projection neurons during locomotion. It would be informative to know whether the L2 neurons that project to L5 receive significant proprioceptive input. This could be analyzed by labeling sensory afferents via intramuscular CTB injection and quantifying afferent synaptic contacts on to labeled L2-L5 neurons.

To determine whether L2-L5 interneurons receive proprioceptive feedback, we stained for vesicular glutamate transporter 1 (vGLUT1)-positive inputs onto retrogradely-labeled L2-L5 interneurons. vGLUT1 is a known synaptic marker of primary afferents (Oliveira et al 2003; Todd et al 2003; Alvarez et al 2004; Mentis et al 2011; Pecho-Vrieseling et al 2009). Figure 8 shows confocal images of vGLUT1-positive signal in close apposition to the somata and proximal dendrites of L2-L5 interneurons.

Collectively from these new analyses, we believe we have addressed (to the best of our abilities) the four key anatomy-focused concerns raised by the reviewer in our functional study of L2-L5 interneurons. **(Reviewer's comment #2)** We have refined our anatomical characterization through exploration of eTeNT.EGFP-positive contacts onto motor neurons (see VACHT staining in Figure 7d-f) and as well as the neurotransmitter phenotype of the putatively-silenced synapses (refer to vGLUT2 staining in Supplementary Figure 4). **(Reviewer's comment #3)** We have quantitatively described the laminar distribution of L2-L5 interneurons throughout the spinal gray matter, revealing a significant difference between the ipsilateral and commissural subtypes (refer to Figure 8i-k). These quantitative analyses are paired with heatmaps to visualize the dorso-ventral, medio-lateral distribution densities of all L2-L5 interneurons as well as the ipsi- and commissural subtypes. **(Reviewer's comment #4)** Using a recently developed immunohistochemical protocol, we assayed for eTeNT.EGFP-positive, putatively silenced synapses derived from double-infected L2-L5 interneurons throughout the lumbar neuraxis. This approach was used in complementation with our improved immunofluorescent protocol (change buffers, detergents, blocking/primary/secondary solutions). With both techniques, we detected little-to-no eTeNT.EGFP-positive, terminal-like signal throughout the lumbar neuraxis apart from the L4-L5 terminal field zone. **(Reviewer's comment #5)** Finally, we have identified the presence of vGLUT1-positive synapses (indicative of proprioceptive inputs) in close apposition to L2-L5 cell bodies and proximal dendrites (refer to Figure 8w).

In addition to these requests, we have also extended our profiling beyond vGLUT1 of plausible synaptic inputs onto L2-L5 interneurons. In Figure 8v,x-y, we show that L2-L5 interneurons appear to receive excitatory inputs (marked by vGLUT2) as well as inhibitory inputs (anti-VGAT immunoreactivity, which labels glycinergic and GABAergic synapses). We also show that L2-L5 interneurons also receive serotonergic inputs (marked by 5HT), which suggests that these neurons receive descending drive from supraspinal centers. This is supported by published literature which suggests that almost all of the 5-HT-positive axons distributed throughout the

spinal cord are derived from serotonergic neurons in the brainstem (e.g. medullary raphe pallidus, raphe obscuris, and raphe magnus; Dahlstroem & Fuxe 1964; Takeuchi et al 1982; Azmitia & Gannon 1986; Azmitia 1999; Ballion et al 2002).

6) The authors remark on the surprising finding that intralimb coordination was not affected, given suggestive evidence from prior electrophysiological studies that L2-L5 neurons might be involved in flexor-extensor coordination across joints. The kinematic data presented in Fig 2 and Suppl. Fig 1 supports their conclusions, but they note that vertical movements in the hip are affected. While IHA and HAT joint angular excursions appear normal, how do the authors interpret abnormal kinematics of hip movement? Could this reflect an aspect of altered intralimb coordination?

We suggest that the increased vertical displacement of the hip (“hip height”) observed during conditional silencing is likely a byproduct of the hop-like phenotype in the hindlimbs. In Figure 2, we show that when the hindlimbs are moving in synchrony (“hopping”), the vertical displacement of the hip is increased (shown schematically in Figure 2c during mid-swing). However, in spite of these changes to “hip height” (shown in Figure 2d, black horizontal trace), the proximal-to-distal coordination of intralimb movements remains unaffected (Figure 2a,c; joint angles listed below schematics; excursions shown in e-g).

7) The authors do not provide any histological analysis to confirm that their viral injection approaches are segment specific. Are double-infected neurons restricted to L2, or did viral spread produce labeling (and thus silencing) of neurons in other lumbar segments as well?

We screened for eTeNT.EGFP-immunoreactive cell bodies throughout the rostral lumbar segments (L1-rostral L3) using our recently developed immunohistochemical protocol. We found that the majority of double-infected cell bodies were at the L2 spinal segment (Figure 7g-k). These darkly-labeled neurons (enhanced with 3,3'-Diaminobenzidine) appeared to be distributed throughout the intermediate gray matter, similar to what we observed with our fluorescent tracer labeling (Figure 8d, h, n, p). Alternatively, very few double-infected neurons were found beyond the L2 spinal segment. A representative example of this sparse labeling is shown in Supplementary Figure 4, where we detected a (weakly) double-infected cell body at the L1 spinal segment.

8) Minor comment- line 165 refers to Fig 5h – this appears to be a typo and should read Fig. 5f.

We kindly thank the reviewer for pointing of this error. We have corrected the line to state Fig. 5f instead of Fig. 5h.

Reviewers' comments:

Reviewer #1 (Remarks to the Author):

The manuscript by Procratsky et al.'s has been significantly improved by the changes made as a response to my comments. For instance, Figure 2 is an important addition.

With regard to my question (line 123-183) regarding "relative coordination" a term introduced by von Holst in the thirties – the authors acknowledge that they do not quite understand the question. As a response they have added the data in the supplementary figure 2 related to the cycle to cycle changes in phase relation, a useful addition, but they have not answered my question. Thus, if you over your 95 steps plot the phase difference of one hindlimb over the other in successive steps you will in the control have stable phase relations of 0.5 or around 0. In your DOX group you may have a much larger drift from cycle to cycle. If there would be a relative coordination you would have several cycles remaining rather close to 0.5 and then a progressive rapid drift to the next point when the coordination may latch on the next coordination point at 0.5 or perhaps around 0 for a while. Relative coordination has been shown in a number of vertebrates (fish and mammals) and invertebrates, when the coordination between CPGs have been partially compromised. To test whether you have a relative coordination or not, just make a plot of the phase lag between one hindlimb as a reference and the other for the successive 95 steps. You have all the information available and the plot can be most likely be completed in an hour or so. This information whether it may imply a relative coordination or random variations should be added and the plot should be included eg in the Supplementary Figure 2.

I note that the authors like to retain their title, which I still find being to overstate the content.

Reviewer #2 (Remarks to the Author):

The authors have addressed all my comments and extensively modified their paper.

Reviewer #3 (Remarks to the Author):

Pocratsky et al. have revised their manuscript exploring the locomotor contributions of a subpopulation of neurons in the lumbar spinal cord that send descending projections from L2 to L5. Their central finding is that rat hindlimb alternation can be selectively disrupted by silencing L2-L5 neurons, while other aspects of locomotion, including intralimb coordination and rhythm, remain unimpaired.

The authors have revised the manuscript and added additional behavioral and anatomical analyses. Despite these additions, I have some concerns about the conclusions drawn from some of the new immunohistochemical data, described below. With this in mind, my original conclusion still stands: the behavioral analysis is a clear strength of this manuscript, but the characterization of the circuits of interest needs to be strengthened if the assertions that the authors make are to be supported. With respect to the behavioral phenotype, the revisions and additional analysis further strengthen the conclusions, and the findings are of sufficient interest and import for publication. However, in attempting to characterize the anatomy of the L2-L5 interneurons, the authors make conclusions that cannot be supported by their existing data.

I will specifically comment on the points the authors raise in the rebuttal. If these specific issues can be adequately addressed, the manuscript would be suitable for publication.

1) The authors identify and attempt to address four anatomy-focused concerns (rebuttal pg 16).

a. They state that they have identified putative synaptic contacts from the affected L2-L5 interneurons onto spinal motor neurons. This analysis is confusing and, from my reading, not correct. Perhaps the authors could clarify:

In the rebuttal letter they state that “motor neurons were identified through the use of the anti-vesicular acetylcholine transporter antibody (VAcHT)”. However, in the manuscript text they state that they use VAcHT “as a presynaptic marker for inputs onto motor neurons”(line 246). These are two very different things. VAcHT will stain both motor neuron cell bodies, as well as cholinergic inputs to neurons in the spinal cord (termed C-boutons, arising from cholinergic spinal interneurons (see Fig 3a: Arvidsson et al., *Journal of Comparative Neurology*, 1997)). The authors seem to be claiming that the neuron in Fig7d-f is a motor neuron. If that is the case, why are they examining colocalization of VAcHT with eTeNT.EGFP in presynaptic terminals? If this cell is a motor neuron, which is what they appear to be saying, it should label with VAcHT in the soma – it is difficult to tell whether this neuron is VAcHT+ (magenta) with the strong red NeuN stain. I suspect it is not a motor neuron, given the lack of clear magenta in the soma, and given its relatively small size- it appears to be approximately the same size as the interneuron in Fig 7a-c. Rather, the authors seem to be identifying eTeNT.EGFP+/ VAcHT+ synaptic terminals, which would be an indication that at least some of the affected L2-L5 interneurons are cholinergic (this could be tested by looking for somatic colocalization of eTeNT.EGFP and VAcHT in L2). Regardless, as written there seems to be confusion about the experiment- if the authors want to demonstrate synaptic contacts onto motor neurons, they need to show a VAcHT+ soma in the ventral horn receiving eTeNT.EGFP+ synaptic contacts. Moreover, if this can be shown, it would be helpful to know, at least in a rough sense, the abundance of this motor neuron innervation- is it very common to find L2-L5 projections onto motor neurons? Finally, in the legend for Figure 7 the authors repeatedly refer to panel g, when I think they mean panel f, which adds to the confusion.

The additional analysis in Suppl. Fig 4 showing eTeNT.EGFP+/VGLuT2+ terminals onto L5 neurons is helpful, at least in demonstrating the excitatory identity of some of the affected neurons, though quantification is lacking. Are the majority of eTeNT.EGFP+ contacts excitatory, or are eTeNT.EGFP+/VGLuT2+ terminals rare? I understand the difficulties of neurotransmitter immunohistochemical identification that the authors raise, but given the success of their VGLuT2 and VAcHT staining, some indication of the relative abundance of excitatory and cholinergic L2-L5 synaptic contacts would be helpful.

b. The authors now describe the laminar distribution of L2-L5 interneurons through quantification and heat map visualization. These experiments are convincing and improve the manuscript.

c. The authors explore the abundance eTeNT.EGFP+ collaterals throughout the lumbar cord and show the majority of contacts reside in L4-L5. This analysis, while purely qualitative, does support their conclusions.

d. Finally, the authors explore the nature of the inputs to L2-L5 neurons. This is particularly important given the finding that limb alternation is not affected during swimming behavior, raising the question of whether sensory afferents provide prominent input to L2-L5 projection neurons. The authors show evidence for serotonergic (5HT), excitatory (VGLuT2) and inhibitory (VGAT) input (though only showing apposition to the cell body without colocalization with synaptic markers is not entirely

convincing). However, the most important question – do they receive proprioceptive feedback – is not adequately addressed. The authors rely solely on VGLuT1 stain, which does label proprioceptive afferents, but also labels descending corticospinal inputs. There is no way to determine the source of the VGLuT1+ inputs with the current data. As I suggested, this could be analyzed by labeling sensory afferents via intramuscular CTB injection and identifying CTB+ synaptic contacts onto labeled L2-L5 neurons. Also, it would be helpful if the authors stated how the L2-L5 interneurons were labeled in Fig 8v-y (retrograde CTB or FluoroEmerald?)

Overall, the problem with the anatomical analysis is that the conclusions are not fully supported by the data. The authors state in the rebuttal letter that “the experiments required to fully address these anatomy-focused questions are beyond the scope and focus of this functional-based paper.” That is well taken, but at a minimum, the data presented needs to support any anatomical conclusions made.

2) I share reviewer #2’s concern that the relatively low proportion of hindlimb steps affected (30%) could be related to the efficacy of viral transduction, the degree of silencing, or restriction to a subset of neurons involved in this behavior (i.e. only L2 and not neighboring segments). The authors address this point in the discussion and improve their analysis by more fully characterizing the location of double-infected neurons. That said, some degree of correlation between the severity of phenotype in each individual mouse and the number of identified double-infected neurons could help. Do mice with higher rates of viral transduction have a higher proportion of affected steps? I recognize the legitimate limitations in generating absolute cell counts described by the authors. That said, they do rely on amplification protocols to detect infected neurons and synaptic terminals in multiple parts of the manuscript. If the technique is reliable enough to identify at least the rough preponderance of infected neurons in L2 and their synaptic terminals in more caudal regions of the spinal cord, then one would think it should be reliable enough to give at least a rough estimate of viral transduction efficiency in each mouse, which could then be correlated to the behavioral data on a mouse-to-mouse basis. At the very least, if no additional immunohistochemical analysis is performed, the authors could describe the level of variability between individual mice- do some mice display a severe phenotype (much greater than 30% of steps affected) while others show only minor disruption? The degree of inter-mouse variability in phenotype severity would be informative.

Minor points:

- typo in line 172- “shed more light on the...”
- line 313- descending efferent input would not be called “feedback”. Perhaps change to “interneurons receive both descending efferent input and afferent feedback (assuming data is presented to support this conclusion- see above)
- Figure 5e legend (line 1220)- what is “*reference”?
- Figure 7 legend- g mistakenly used instead of f a few times, as mentioned above

Reviewers' comments:

Reviewer #1 (Remarks to the Author):

The manuscript by Pocratsky et al has been significantly improved by the changes made as a response to my comments. For instance, Figure 2 i's an important addition.

With regard to my question (line 123-183) regarding “relative coordination” a term introduced by von Holst in the thirties the authors acknowledge that they do not quite understand the question. As a response they have added the data in the supplementary figure 2 related to the cycle to cycle changes in phase relation, a useful addition, but they have not answered my question. Thus, if you over your 95 steps plot the phase difference of one hindlimb over the other in successive steps you will in the control have stable phase relations of 0.5 or around 0. In your DOX group you may have a much larger drift from cycle to cycle. If there would be a relative coordination you would have several cycles remaining rather close to 0.5 and then a progressive rapid drift to the next point when the coordination may latch on the next coordination point at 0.5 or perhaps around 0 for a while. Relative coordination has been shown in a number of vertebrates (fish and mammals) and invertebrates, when the coordination between CPGs have been partially compromised. To test whether you have a relative coordination or not, just make a plot of the phase lag between one hindlimb as a reference and the other for the successive 95 steps. You have all the information available and the plot can be most likely be completed in an hour or so. This information whether it may imply a relative coordination or random variations should be added and the plot should be included eg in the Supplementary Figure 2.

We thank the reviewer for providing clarification on their question regarding “relative coordination.” We believe there might a misunderstanding regarding the data shown in Figure 5 and Supplementary Figure 2.

The reviewer stated that to analyze relative coordination, we should plot “over the 95 steps the phase difference of one hindlimb over the other in successive steps...” Similarly, the reviewer suggested that we “...make a plot of the phase lag between one hindlimb as a reference and the other for the successive 95 steps.” We interpret these statements to indicate that the reviewer believes the 95 step cycles described in Supplementary Figure 2 are contiguous. This is not the case and we apologize if we have described these analyses to indicate otherwise. The data shown in Supplementary Figure 2 and Figure 5 reflect per-step changes in coordination from a collection of individual locomotor bouts from each animal across time. As we previously described in the methods section, locomotor bouts that meet the selection criteria for analyses are approximately 1 meter in length (lines 477). This corresponds to approximately 4-5 contiguous step cycles. The relatively small number of step cycles is due to two key methodological considerations. First, the stepping enclosure is approximately 150 cm in length. We found that using a longer stepping tank (300 cm) requires animals to receive “positive reinforcement training” to encourage the completion of one locomotor bout without any exploration/hesitations/pauses across the elongated walkway (e.g. animals’

receive food rewards for completing one bout, start-to-finish). Therefore, while the number of contiguous step cycles is fewer in the 150 cm long tank, we believe the “mode of stepping” to be more natural (e.g. no human influence on the expressed motor behavior, a confounding variable). The second reason we observe fewer contiguous step cycles is that our video recordings are designed to exclude both the initiation and termination of locomotion. Therefore, while the animals can take up to 8 contiguous step cycles in the chamber, the first and last steps taken are not recorded (obstructed from the field of view by the chamber setup). This rules out any confounding effects derived from the propulsive/braking forces applied during stepping onset/offset.

Therefore, if the reviewer indeed thought that the data shown in Supplementary Figure 2 represented ~95 contiguous step cycles, we agree that these data would be most meaningful for a more rigorous investigation of the relative coordination (per their request). Due to the limited number of contiguous step cycles, we cannot fully address the reviewer’s question as to whether or not there are several step cycles that remain close to 0.5 followed by a “progressive rapid drift” to the next latch point on the coordination spectrum. This is why we strictly showed per-step changes in left-right coordination without extracting additional information beyond this data point. We previously noted these limitations in the results, stating that more contiguous step cycles are required to thoroughly explore the relative drift in left-right hindlimb coordination (lines 176-178). Data shown in Figure 5 and Supplementary Figure 2 were analyzed with respect to these limitations as both the per-step changes in left-right coordination as well as the phase relationships across successive steps were plotted within an individual locomotor bout, not across/between them.

We have further clarified these methodological details in the manuscript (lines 477-481, 571-574)

I note that the authors like to retain their title, which I still find being to overstate the content.

We have changed the title to the following: “Reversible silencing of lumbar spinal interneurons unmasks a flexible, task-specific network for securing hindlimb alternation.”

-

Reviewer #2 (Remarks to the Author):

The authors have addressed all my comments and extensively modified their paper.

-

Reviewer #3 (Remarks to the Author):

Pocratsky et al. have revised their manuscript exploring the locomotor contributions of a subpopulation of neurons in the lumbar spinal cord that send descending projections from L2 to L5. Their central finding is that rat hindlimb alternation can be selectively disrupted by silencing L2-L5 neurons, while other

aspects of locomotion, including intralimb coordination and rhythm, remain unimpaired.

The authors have revised the manuscript and added additional behavioral and anatomical analyses. Despite these additions, I have some concerns about the conclusions drawn from some of the new immunohistochemical data, described below. With this in mind, my original conclusion still stands: the behavioral analysis is a clear strength of this manuscript, but the characterization of the circuits of interest needs to be strengthened if the assertions that the authors make are to be supported. With respect to the behavioral phenotype, the revisions and additional analysis further strengthen the conclusions, and the findings are of sufficient interest and import for publication. However, in attempting to characterize the anatomy of the L2-L5 interneurons, the authors make conclusions that cannot be supported by their existing data.

I will specifically comment on the points the authors raise in the rebuttal. If these specific issues can be adequately addressed, the manuscript would be suitable for publication.

1) The authors identify and attempt to address four anatomy-focused concerns (rebuttal pg 16).

a. They state that they have identified putative synaptic contacts from the affected L2-L5 interneurons onto spinal motor neurons. This analysis is confusing and, from my reading, not correct. Perhaps the authors could clarify:

In the rebuttal letter they state that motor neurons were identified through the use of the anti-vesicular acetylcholine transporter antibody (VACHT). However, in the manuscript text they state that they use VACHT as a presynaptic marker for inputs onto motor neurons (line 246). These are two very different things. VACHT will stain both motor neuron cell bodies, as well as cholinergic inputs to neurons in the spinal cord (termed C-boutons, arising from cholinergic spinal interneurons (see Fig 3a: Arvidsson et al., Journal of Comparative Neurology, 1997). The authors seem to be claiming that the neuron in Fig7d-f is a motor neuron. If that is the case, why are they examining colocalization of VACHT with eTeNT.EGFP in presynaptic terminals? If this cell is a motor neuron, which is what they appear to be saying, it should label with VACHT in the soma; it is difficult to tell whether this neuron is VACHT+ (magenta) with the strong red NeuN stain (refer to Authors' response #1a-i). I suspect it is not a motor neuron, given the lack of clear magenta in the soma, and given its relatively small size- it appears to be approximately the same size as the interneuron in Fig 7a-c. Rather, the authors seem to be identifying eTeNT.EGFP+/ VACHT+ synaptic terminals, which would be an indication that at least some of the affected L2-L5 interneurons are cholinergic (this could be tested by looking for somatic colocalization of eTeNT.EGFP and VACHT in L2) (Authors' response #1a-ii). Regardless, as written there seems to be confusion about the experiment- if the authors want to demonstrate synaptic contacts onto motor neurons, they need to show a VACHT+ soma in the ventral horn receiving eTeNT.EGFP+ synaptic contacts (refer to Authors' response #1a-i). Moreover, if this can be shown, it

would be helpful to know, at least in a rough sense, the abundance of this motor neuron innervation- is it very common to find L2-L5 projections onto motor neurons (Authors' response #1a-iii)? Finally, in the legend for Figure 7 the authors repeatedly refer to panel g, when I think they mean panel f, which adds to the confusion (Authors' response #1a-iv).

The additional analysis in Suppl. Fig 4 showing eTeNT.EGFP+/VGlut2+ terminals onto L5 neurons is helpful, at least in demonstrating the excitatory identity of some of the affected neurons, though quantification is lacking. Are the majority of eTeNT.EGFP+ contacts excitatory, or are eTeNT.EGFP+/VGlut2+ terminals rare? I understand the difficulties of neurotransmitter immunohistochemical identification that the authors raise, but given the success of their VGlut2 and VAcHT staining, some indication of the relative abundance of excitatory and cholinergic L2-L5 synaptic contacts would be helpful (Authors' response #1a-v).

We acknowledge the reviewer's concern with some of the conclusions drawn from our histological data. We have taken the following steps to address these issues.

(Authors' response #1a-i) In the results section, we state that VAcHT was used as a marker for presynaptic inputs onto motor neurons. We utilized VAcHT instead of choline acetyltransferase (ChAT), perhaps a more "traditional" marker for motor neurons as the protocol required to detect its expression was incompatible with that of amplifying eTeNT.EGFP. We acknowledge that published data illustrates VAcHT labels both the presynaptic (cholinergic) puncta as well as the motor neuron somata, leading to the ambiguity in the identity of neurons shown in Fig. 7d-f (e.g. lack of somata staining, per reviewer's commentary). However, based on our experience, we see considerably weak VAcHT signal in the somata as compared to the puncta (similar observations have been documented elsewhere; Wootz et al 2013, J Comp Neurol). This is illustrated in Clarification Figure 1 (provided for the benefit of reviewer without the intent of publication), where two scans of the same retrogradely-labeled motor neurons were acquired: one optimized for the VAcHT⁺ puncta and the other for the VAcHT⁺ somata (details of this experiment are described below in Authors' response #1d-ii). It is clear that when the imaging conditions are optimized for the puncta, the somata are not labeled (little-to-no signal) (Clarification Figure 1; image acquisition settings: 405 laser set to 680 V, 10% laser power). Alternatively, when the imaging conditions are optimized to the VAcHT⁺ somata, the puncta become oversaturated (Clarification Figure 1; image acquisition settings: 755V, 30% laser power). We have observed weak VAcHT somatic labeling in three separate experiments: (1) at the L5 terminal field of double-infected (eTeNT.EGFP⁺) L2-L5 interneurons, (2) at spinal L2 where the CTB-labeled L2-L5 interneurons reside, and (3) in retrogradely-labeled motoneurons following intramuscular injections into various hindlimb muscles. Notably, this spans different experimental designs (virus, intraspinal tracer, intramuscular tracer) and different fixation methods (post-fixation overnight versus post-fixation for 1-3 hours). This is likely the reason as to why there appears to be no VAcHT signal in the somata in Fig. 7d-f as we do not use optical configurations that lead to oversaturation of the fluorophore (an apparent requirement to detect VAcHT in the somata). In the Clarification Figure 2 below, we have isolated the red and far red channels, separating VAcHT from NeuN of

the neuron shown in Fig. 7e. Here, we adjusted the look up table values (LUTs) to “reveal” VACHT in the somata, which was previously masked by the intense NeuN signal, as the reviewer suggested.

Clarification Figure 1.

Clarification Figure 1. VACHT shows poor immunoreactivity in the somata as compared to pre-synaptic puncta using our standard optical configurations. Images shown in (a_{i-ii}) and (b_{i-ii}) reflect VACHT-positive immunoreactivity with optimal confocal configurations set for the pre-synaptic puncta (not shown are CTB-labeled MNs from intramuscular injections; refer to “ghost” neurons in a_{ii} and b_{ii}). Left panels reflect raw VACHT signal (blue). Right panels reflect the inverted image to show saturation of the fluorophore based on image acquisition settings. Ideal optical configurations for fluorophore detection yield gray-white signal (white arrowheads). Images shown in (c_{i-ii}) and (d_{i-ii}) reflect VACHT-positive immunoreactivity with optical configurations set for the somata. Note that when VACHT signal is detected in the somata, the pre-synaptic puncta become oversaturated (red signal, white arrowheads). Images shown reflect one optical slice (0.4 μ m) through a 20-30 slice z-stack captured at 100x (scale bar=20 μ m).

Clarification Figure 2.

Clarification Figure 2. Separation of red and far red channels to reveal weak VACHT-signal in the somata. (a-e) Representative example of *post hoc* image modification to reveal VACHT-positive signal in the somata and presynaptic puncta. (a) Target putative motor neuron from Figure 7e for *post hoc* image modifications. (b) Orthogonal cross-section (one optical slice) through z-stack illustrating VACHT immunoreactivity in puncta that surround NeuN-positive somata. (c-d) NeuN and VACHT signal shown isolated for clarification. (e) Post hoc enhancement of VACHT reveals immunoreactivity in the somata. Note that this signal is considerably weak as compared to the VACHT immunoreactivity in the puncta (similar to what we show in **Clarification Figure 1**).

The reviewer also questioned the relative size of the motor neurons shown in Fig. 7d-f as compared to the interneurons in Fig. 7a-c. We believe this discrepancy in apparent size is likely due to two key considerations: (1) the three-dimensional presentation of the volume-rendered neurons (e.g. differential rotation in XYZ planes) and (2) where within the z-stack motor neuron cross-sections were sampled for orthogonal representation (e.g. sample through the mid-point of the somata vs beginning/end) (Figure 7).

(Authors' response #1a-ii) The reviewer indicated that a subset of L2-L5 interneurons could be cholinergic (per eTeNT-VACHT co-localization in puncta at L5), suggesting that a somatic expression of VACHT should also exist. Despite the poor VACHT somatic labeling observed (described above), we still screened for the co-localization of VACHT with our CTB-labeled L2-L5 interneurons at the level of the cell bodies (samples screened were generated from experiments shown in Figure 8l). While most L2-L5 interneurons appear to reside just dorsal to where we observe robust VACHT immunoreactivity (Clarification Figure 4a-d), we did observe instances of somatic co-localization (panel e). These data are provided for the benefit of the reviewer.

Clarification Figure 4.

(Authors' response #1a-iii) Assessing the “relative abundance” of eTeNT.EGFP innervation onto motor neurons proves to be challenging (apart from the technical difficulties associated with IHC) for the following reasons. eTeNT.EGFP is a fusion protein. As such, it is not localized to a discrete cellular location (e.g. synaptic terminals). Instead, eTeNT.EGFP is present throughout the rostrocaudal length of the double-infected neuron, from somata (Figure 7g-k), axons and their collaterals (plausible examples shown in Supplementary Figure 4r-y), all the way to synapses (Figure 7a-f). Therefore, each cross-section likely includes (dense) axonal collateralization as L2-L5 interneuron-derived fibers innervate the caudal lumbar

enlargement to synapse onto their end targets (putative collateral branches shown in Supplementary Figure 4, panels s-u, x). This makes it challenging to reconcile, in a general sense, the amount of eTeNT.EGFP-positive inputs onto motor neurons (or interneurons) as we observe abundant green signal within laminae VII-X (near NeuN-positive somata and dendrites) that does not strictly co-localize with synapse-related markers (e.g. synaptophysin, VACHT, VGAT, vGLUT2). It is technically unfeasible to quantitatively address this question in our model. Notwithstanding, in a different experimental model (isolated spinal cords from p1-p4 Wistar rats), Birinyi et al (2003) revealed that n=291/632 labeled motor neurons received contacts from commissural L2 interneurons (a potential correlate of the commissural L2-L5 interneuron studied here).

(Authors' response #1a-iv) We kindly thank the reviewer for pointing out this error. We have corrected the text to state panel f instead of panel g.

(Authors' response #1a-v) Assessing the relative abundance of L2-L5 synaptic inputs that are excitatory or cholinergic presents with the same technical challenges we described above in our Authors' response #1a-iii. Again, quantifying the relative proportion of green (eTeNT.EGFP) that co-localizes with vGLUT2 (or VACHT) is challenging as we see more green signal that does not co-localize with this marker (e.g. a cross-section will show putative axon collaterals, arbors, and synapses, of which not all are excitatory). These technical challenges make it difficult to reconcile, with any level of certainty, what relative abundance of vGLUT2 or VACHT co-localization with eTeNT.EGFP. Our experimental model is not designed to properly answer these questions. As such, it would be difficult to draw any firm anatomical conclusions as the data would be generated from a method not suitable to test these hypotheses.

In our previous response letter, we discussed our technical difficulties in identifying putatively silenced inhibitory synapses (e.g. antibodies that consistently labeled inhibitory synapses and eTeNT.EGFP were both rabbit host species, poor immunoreactivity outcomes if we switched to other host species primaries). The authors would also like to update the reviewer(s) regarding the neurotransmitter phenotype of the putatively silenced synapses. Following advice from Dr. Andrew Todd, we solicited a goat anti-vesicular GABA transporter antibody (VGAT) from the Frontier Institute in Japan. With this new primary we were able to detect putatively silenced inhibitory synapses (please refer to Supplementary Figure 4). We have updated the results and methods sections accordingly (lines 245-247 and 640-646, respectively).

To conclude, we fully acknowledge the limitations of the TetOn system. While it is a powerful technique to probe the functional role(s) of anatomically-defined pathways, there is a limited ability to investigate, with great detail, the anatomical underpinnings of the any changes observed. We appreciate the reviewer's acknowledgement that fully addressing these anatomy-focused questions are beyond the scope and focus of this functional-based paper.

b. The authors now describe the laminar distribution of L2-L5 interneurons through quantification and heat map visualization. These experiments are convincing and improve the manuscript.

c. The authors explore the abundance eTeNT.EGFP+ collaterals throughout the lumbar cord and show the majority of contacts reside in L4-L5. This analysis, while purely qualitative, does support their conclusions.

d. Finally, the authors explore the nature of the inputs to L2-L5 neurons. This is particularly important given the finding that limb alternation is not affected during swimming behavior, raising the question of whether sensory afferents provide prominent input to L2-L5 projection neurons. The authors show evidence for serotonergic (5HT), excitatory (VGLuT2) and inhibitory (VGAT) input (though only showing apposition to the cell body without colocalization with synaptic markers is not entirely convincing) (Authors' response #1d-i). However, the most important question; do they receive proprioceptive feedback, is not adequately addressed. The authors rely solely on VGLuT1 stain, which does label proprioceptive afferents, but also labels descending corticospinal inputs (Authors' response #1d-ii). There is no way to determine the source of the VGLuT1+ inputs with the current data. As I suggested, this could be analyzed by labeling sensory afferents via intramuscular CTB injection and identifying CTB+synaptic contacts onto labeled L2-L5 neurons. Also, it would be helpful if the authors stated how the L2-L5 interneurons were labeled in Fig 8v-y (retrograde CTB or FluoroEmerald?) (refer to Authors' response #1d-i).

Overall, the problem with the anatomical analysis is that the conclusions are not fully supported by the data. The authors state in the rebuttal letter that the experiments required to fully address these anatomy-focused questions are beyond the scope and focus of this functional-based paper; That is well taken, but at a minimum, the data presented needs to support any anatomical conclusions made.

(Authors' response #1d-i) Regarding the reviewer's request for clarification on the labeling paradigm to visualize L2-L5 interneurons (Fig. 8v-y), images shown were generated from the experiments illustrated in Figure 8 panel I (CTB-AlexaFluor-594 and -647 injected at L5, CTB-AlexaFluor-488 injected at L1). Therefore, while the neurons shown in Figure 8v-y are exclusively 647⁺, they were imaged from spinal cord samples that contained 488, 594, and 647 fluorophores. We have clarified this in the manuscript (please refer to the figure legend, lines 1296-1297). This underscores the reviewer's concern regarding the "not entirely convincing" input staining we performed (e.g. 5HT, vGLUT1, vGLUT2, VGAT), all of which was not co-localized with a synapse-specific marker (e.g. synaptophysin). As these spinal cross-sections already had 488, 594, and 647 labeling, we were limited to 405 (blue) as our fluorophore readout for the various input markers. Co-localization of these input markers with a synapse-specific label (e.g. synaptophysin) would require the use of (and *post hoc* separation of) five fluorophores. Our current microscopy setup does not allow us to perform this experiment as we are limited to four lasers (405, 488, 594, and 647).

However, in an attempt to address the reviewer's concern, we performed histological analyses on tissue that was generated from an earlier, unrelated study. N=5 adult female Sprague Dawley rats received unilateral injections of 10% FluoroEmerald at L5 to retrogradely label L2-L5 interneurons (in addition to FluoroRuby injected at a cervical region of interest, not shown). Using this tissue, we detected the co-localization of synaptophysin (synapse-specific marker) with VGAT or vGLUT1 onto retrogradely-labeled L2-L5 interneurons (Clarification Figure 5). Similar results were observed for 5HT and vGLUT2 (not shown). Note that using fluorescent dextrans to label L2-L5 interneurons yielded considerably more punctate labeling in the somata as compared to CTB. These images are provided for the benefit of the reviewer.

Clarification Figure 5.

(Authors' response #1d-ii) We acknowledge the limitations with our approach to detect proprioceptive feedback onto L2-L5 interneurons. We chose to screen for vGLUT1 as a marker for afferent feedback based on a literature review (Alvarez et al 2004, J Comp Neurol; Mentis et al 2006, Neuron; Demireva et al 2011, Cell; de Nooij et al, 2013, Neuron) as well as data which suggests that less than 2% of vGLUT1-positive synapses (in the ventral gray matter) are derived from the corticospinal tract (Betley et al 2009, Cell). However, we respect the reviewer's concern regarding our interpretation

of the vGLUT1 histology. We recognize that this is not an enriched marker for afferent fibers (to the exclusion of any other fiber type) and we apologize if our language has indicated otherwise.

The reviewer suggested that we screen for sensory input onto L2-L5 interneurons via intramuscular CTB injections with *post hoc* screening for CTB-positive synaptic contacts onto retrogradely-labeled L2-L5 interneurons. Before we performed the necessary combinatorial intraspinal and intramuscular tracer injections, we first wanted to validate the intramuscular injections. Specifically, we wanted to know if we could reliably generate (robust) CTB-positive puncta labeling in lamina V, an area that is known to receive dense afferent input (Watson 2009). We anticipated seeing relatively dense CTB-positive, puncta-like signal throughout the deep dorsal horn and into the intermediate gray matter (in addition to the retrogradely labeled motor neurons) following injections into select hindlimb muscles.

Adult female Sprague Dawley rats received a series of CTB-AlexaFluor conjugate injections into various hindlimb muscle groups. While we did observe retrogradely-labeled motor neurons (Clarification Figure 6a, representative images from iliopsoas injections shown), we did not see a preponderance of puncta-like structures near laminae V-VIII. We observed little-to-no CTB/vGLUT1 or CTB/synaptophysin co-localizations, even after enhancement of CTB using immunohistochemistry (Clarification Figure 6b, d-e). These experiments were repeated multiple times with various concentrations of CTB (1-3%), different target muscle groups (iliopsoas, gracilis, and the medial/lateral gastrocnemius), and various time points post-injection for euthanasia/tissue processing. Therefore, while we attempted to perform the experiments suggested by the reviewer, we have serious concerns moving forward with this approach.

In an attempt provide additional validation for the histological interrogation of afferent feedback onto L2-L5 interneurons, we also screened for parvalbumin-positive puncta onto L2-L5 interneurons. Parvalbumin has been shown to mark sensory fibers (Honda 1995, Neuroscience; Arber et al 2000, Cell; Patel et al 2003, Neuron; de Nooij et al 2013, Neuron; O'Toole et al 2017, J Neurophysiol) and presents with distinct patterns of co-localization with vGLUT1 (Siembab et al 2010, Journal of Comp Neurol). Unfortunately, we could not reliably detect parvalbumin⁺ immunoreactivity in our remaining tissue (screened with three different Swant primary antibodies). It seems as though we are left with our vGLUT1 immunoreactivity, which we now show co-localizes with synaptophysin (Clarification Figure 6). We have updated manuscript to state that the input markers co-localize with synaptophysin (lines 313-315, stated as “data not shown” in manuscript).

More importantly, we have modified our interpretation of these data. Instead of explicating stating that “...these neurons [L2-L5 interneurons] also appear to receive ...proprioceptive inputs”, we have adjusted our language to the following:

“...these neurons also appear to receive 5HT (Fig. 8v) and vGLUT1-positive inputs (Fig. 8w) (with synaptic co-localization, data not shown). These data suggest that L2-L5 interneurons could receive both descending and afferent input, although the direct source of vGLUT1 cannot be identified.” We believe this revised conclusion is supported by the data shown (lines 315-317).

Clarification Figure 6.

Clarification Figure 6. Immunohistological assessment of puncta-like structures in spinal gray matter derived from intramuscular injections of CTB. (a)

Representative image of the pattern of CTB-positive signal observed following intramuscular injections (results from iliopsoas shown). (b) Sparse, CTB-positive puncta-like structures were detected within the deep dorsal horn (bright white signal), an area known to be densely innervated by sensory fibers (Watson 2009). (c) For comparison, vGLUT1 shows dense immunoreactivity in the same region of interest. For visual aid images were converted to gray scale. (d-e) Confocal images show CTB-positive puncta do not exclusively co-localize with vGLUT1 nor synaptophysin. Alternatively, vGLUT1 consistently co-localized with synaptophysin. Images shown in a-c taken at 10x. Images shown in d-e taken at 100x. Scale bar=20 μ m.

2) I share reviewer #2's concern that the relatively low proportion of hindlimb steps affected (30%) could be related the efficacy of viral transduction, the degree of silencing, or restriction to a subset of neurons involved in this behavior (i.e. only L2 and not neighboring segments). The authors address this point in the discussion and improve their analysis by more fully characterizing the location of double-infected neurons. That said, some degree of correlation between the severity of phenotype in each individual mouse and the number of identified double-infected neurons could help. Do mice with higher rates of viral transduction have a higher proportion of affected steps? I recognize the legitimate limitations in generating absolute cell counts described by the authors. That said, they do rely on amplification protocols to detect infected neurons and synaptic terminals in multiple parts of the manuscript. If the technique is reliable enough to identify at least the rough preponderance of infected neurons in L2

and their synaptic terminals in more caudal regions of the spinal cord, then one would think it should be reliable enough to give at least a rough estimate of viral transduction efficiency in each mouse, which could then be correlated to the behavioral data on a mouse-to-mouse basis. At the very least, if no additional immunohistochemical analysis is performed, the authors could describe the level of variability between individual mice- do some mice display a severe phenotype (much greater than 30% of steps affected) while others show only minor disruption? The degree of inter-mouse variability in phenotype severity would be informative.

While we have emphasized the technical difficulties in generating absolute cell counts of affected L2-L5 interneurons, we also respect the reviewer's repeated requests for this information. Therefore, in an attempt to explore whether rats with a higher proportion of affected steps had higher rates of viral transduction (underscoring the inter-animal variability), we performed relative cell counts of eTeNT-positive DAB immunoreactive neurons throughout the L2 segment in two animals. Prior to the counts, we did not screen for how these animals performed functionally during silencing.

In animal #3, we observed an average of 123.43 ± 14.66 eTeNT.EGFP⁺, DAB-immunoreactive neurons per cross-section (n=7 cross-sections). In animal #6, we observed an average of 139.33 ± 6.66 eTeNT.EGFP⁺, DAB-immunoreactive neurons per cross-section. These counts were performed on sections sampled throughout the rostrocaudal extent of the L2 segment, excluding sections close to the injection sites due to the signal:noise technical difficulties we previously described. For reference, we observed an average of 153.71 ± 34.75 to 180 ± 34.48 CTB-labeled L2-L5 interneurons when sampling across the L2 segment (n=7 cross-sections; data from experiment shown in Figure 8f). Again, we would like to emphasize there are consequential technical considerations regarding direct comparisons between these experiments and numbers (please refer to our previous response letter).

Cross-referencing these data to the functional outcome measures revealed that we performed cell counts on "low" and "high" performing rats (table below). The data shown in the table reflects our assessment of the inter-animal variability, both the peak magnitude change in hindlimb coordination as well as when this event occurred. These data were generated by calculating the percent of total steps analyzed (at each time point, respectively) that fell beyond normal variability associated with stepping observed across all control time points.

Animal #3 (average of 123.43 ± 14.66 eTeNT.EGFP⁺ L2-L5 interneurons) showed modest silencing-induced changes (~20% steps affected) while animal #6 (139.33 ± 6.66 eTeNT.EGFP⁺ neurons) showed overt changes to hindlimb coordination (~69% steps affected). These preliminary data could suggest that there might not be a "clear" association between the number of L2-L5 interneurons silenced and the magnitude change in hindlimb alternation. However, we strongly state that these data are not conclusive evidence to support this notion. Absolute cell counts of all infected L2-L5 interneurons are unequivocally required to draw a firm conclusion to directly address the reviewers' questions. These data cannot be generated due to the inherent limitations of the TetOn system as well as the histological analyses (as described in our previous

response letter). Therefore, we strongly advise that these preliminary counts are not published with the manuscript. We provided them for the benefit of the reviewers.

Inter-animal variability in the effects of silencing L2-L5 interneurons		
Animal ID	% steps disrupted	Time point
1	60.0	DOX1 ^{ON} -D5
2	29.4	DOX1 ^{ON} -D8
3	21.4	DOX1^{ON}-D5
4	30.8	DOX2 ^{ON} -D5
5	25.0	DOX1 ^{ON} -D5
6	68.8	DOX2^{ON}-D5

Referring back to the inter-animal variability findings, we acknowledge the reviewer's statement regarding the relatively modest effects silencing L2-L5 interneurons had on left-right alternation (e.g. rat #3 and #5 with ~20% and 25% of the steps taken affected). However, when compared to the "quality" of hindlimb stepping observed at control time points, even these modest changes are still pronounced (e.g. at DOX^{OFF}, 0% of steps taken deviated beyond control variability for #3 and #5, respectively). Not only do we observe inter-animal variability in the "peak effects" of silencing L2-L5 interneurons, but also when these events occurred. These inter-animal variability findings are briefly summarized in the results section (lines 112-116).

Minor points:

- typo in line 172-;shed more light on the;

We kindly thank the reviewer for pointing out this mistake. We have corrected it accordingly (lines 176-178).

- line 313- descending efferent input would not be called feedback. Perhaps change to interneurons receive both descending efferent input and afferent feedback (assuming data are presented to support this conclusion- see above)

We have updated our language per the reviewer's suggestion to state the following: "These data suggest that L2-L5 interneurons receive both descending and afferent input, although the direct source of vGLUT1-positive terminals cannot be identified" Please refer to our previous commentary regarding the synaptic co-localization of various input markers onto retrogradely labeled L2-L5 interneurons.

- Figure 5e legend (line 1220)- what is*reference?

We apologize for the ambiguity of our language in the figure legend. Here, "*reference" refers to the reference limb that defines one complete stride cycle from initial contact to initial contact (as defined in the methods section, lines 546-548). Therefore, with respect (or reference) to one complete right hindlimb stride cycle, we are investigating when the

left hindlimb makes initial contact. We have clarified this further in the figure legend (lines 1231-1234).

- Figure 7 legend- g mistakenly used instead of (f) a few times, as mentioned above

We thank the reviewer for identifying this mistake. We have made the relevant corrections (lines 1266-1274).

REVIEWERS' COMMENTS:

Reviewer #1 (Remarks to the Author):

The authors had addressed my original questions and reservations already in the first round, and I do not like to raise any further questions now.

Reviewer #3 (Remarks to the Author):

Pocratsky et al. have submitted a second revision of their manuscript exploring the locomotor contributions of a subpopulation of neurons in the lumbar spinal cord that send descending projections from L2 to L5.

The authors have made significant attempts to address the concerns raised. While not all new experiments were successful, the main concern, that the conclusions made were not supported by the existing data, has been almost entirely addressed. I now believe this manuscript is suitable for publication, if one error that persists is corrected.

The problem, as mentioned in the last review, rests with a misstatement on lines 249-250: "We used anti-vesicular acetylcholine transferase (VACHT) as a presynaptic marker for inputs onto motor neurons." Labeling of cholinergic presynaptic terminals is not a definitive demonstration that the postsynaptic neuron is a motor neuron. While it is true that motor neurons receive abundant cholinergic inputs (c-boutons), there is evidence that spinal interneurons do as well (for example see Siembab et al, 2010). I see a few solutions here: the authors present the images from the rebuttal letter Clarification Figures 1 and 2 showing VACHT signal (albeit weak) in the motor neurons themselves (it doesn't really matter if the C-boutons are oversaturated); the authors state that the cells are VACHT-positive but data is not shown (not ideal); or the authors modify the text to say motor neurons and/or interneurons receive input (which provides little additional information).

As an added suggestion, given the nice demonstration that GFP colocalizes with VACHT in presynaptic terminals, and the additional data presented in Clarification Figure 4, the authors could mention that the data suggest a small proportion of L2-L5 neurons are cholinergic, as the findings support this conclusion.

Besides this one issue, I congratulate the authors on a nice manuscript (and I apologize for my purely habitual use of 'mouse' instead of 'rat'!)

Reviewer #3 (Remarks to the Author):

The problem, as mentioned in the last review, rests with a misstatement on lines 249-250: “We used anti-vesicular acetylcholine transferase (VACHT) as a presynaptic marker for inputs onto motor neurons.” Labeling of cholinergic presynaptic terminals is not a definitive demonstration that the postsynaptic neuron is a motor neuron. While it is true that motor neurons receive abundant cholinergic inputs (c-boutons), there is evidence that spinal interneurons do as well (for example see Siembab et al, 2010). I see a few solutions here: the authors present the images from the rebuttal letter Clarification Figures 1 and 2 showing VACHT signal (albeit weak) in the motor neurons themselves (it doesn't really matter if the C-boutons are oversaturated); the authors state that the cells are VACHT-positive but data is not shown (not ideal); or the authors modify the text to say motor neurons and/or interneurons receive input (which provides little additional information).

We kindly thank the reviewer for providing several options to address their concern. We have opted to present the images shown in the Clarification Figures 1 and 2 as a new Supplementary Figure (#6). We have updated our results section to guide the readers to refer to this figure regarding the somatic versus pre-synaptic VACHT immunoreactivity (lines 294-298).

As an added suggestion, given the nice demonstration that GFP colocalizes with VACHT in presynaptic terminals, and the additional data presented in Clarification Figure 4, the authors could mention that the data suggest a small proportion of L2-L5 neurons are cholinergic, as the findings support this conclusion.

We appreciate this additional suggestion and have since updated our results section to reflect that a small proportion of L2-L5 interneurons are cholinergic (lines 292-293).

Besides this one issue, I congratulate the authors on a nice manuscript (and I apologize for my purely habitual use of ‘mouse’ instead of ‘rat’!)